



# Atmospherically-produced beryllium-10 in annually laminated late-glacial sediments of the North American Varve Chronology

Greg Balco[1], Benjamin D. DeJong[2,3], John C. Ridge[4], Paul R. Bierman[3], and Dylan H. Rood[5,6]

[1]Berkeley Geochronology Center, 2455 Ridge Road, Berkeley CA USA
[2]Vanasse Hangen Brustlin, Inc., Montpelier VT USA
[3]Department of Geology, University of Vermont, Burlington VT USA
[4]Department of Earth and Ocean Sciences, Tufts University, Medford MA USA
[5]Department of Earth Science and Engineering, Royal School of Mines, Imperial College London, London UK
[6]Earth Research Institute, University of California, Santa Barbara, CA 93106, USA

**Correspondence:** Greg Balco (balcs@bgc.org)

**Abstract.**

We attempt to synchronize the North American Varve Chronology (NAVC) with the calendar year time scale by comparing records of atmospherically produced [10]Be fallout in the NAVC and in ice cores. The North American Varve Chronology (NAVC) is a sequence of 5659 varves deposited in a series of proglacial lakes adjacent to the southeast margin of the

retreating Laurentide Ice Sheet between approximately 18,200 - 12,500 years before present. Because properties of NAVC varves are related to climate, the NAVC is also a climate proxy record with annual resolution, and our overall goal is to place the NAVC and ice core records on the same time scale to facilitate high-resolution correlation of climate events. Total [10]Be concentrations in NAVC sediments are within the range of those observed in other lacustrine records of [10]Be fallout, but [9]Be and [10]Be concentrations considered together show that the majority of [10]Be is present in glacial

sediment when it enters the lake, and only a minority of total [10]Be derives from atmospheric fallout at the time of sediment deposition. Because of this, an initial strategy to determine whether or not [10]Be fallout variations were recorded in NAVC sediments by attempting to observe the characteristic 11-year solar cycle in high-resolution sections of varve sequences was inconclusive: observed short-period variations at the expected magnitude of this cycle were not distinguishable from measurement scatter. On the other hand, we did observe centennial-period variations in [10]Be fallout that are replicated

between separate varve sections and have similar magnitude and frequency as coeval [10]Be fallout variations recorded in ice core records. These are most prominent in glacial sections of the NAVC that were deposited in proglacial lakes, but are suppressed in paraglacial sections of the NAVC deposited in lakes lacking direct glacial sediment input, which leads us to conclude that proglacial lakes whose watershed likely includes a large portion of the ablation area of an ice sheet can effectively record [10]Be fallout. We matched observed centennial-scale [10]Be fallout variations in two segments

of the NAVC to ice core [10]Be fallout records. Although the calibration of the NAVC to the calendar year time scale implied by these matches is similar to that proposed previously in independent calibrations based on radiocarbon data and correlation of climate events, matches for the two different segments disagree with each other and with the independent calibrations by 50-200 years. One of these matches is not consistent with independent evidence and is probably not valid,





but the other is consistent with most, although not all, evidence and may be valid. This leaves several remaining ambiguities in whether or not [10]Be fallout variations can, in fact, be used for synchronizing NAVC and ice core timescales, but these could likely be resolved by higher-resolution and replicate [10]Be measurements on targeted sections of the NAVC.

## 1 Introduction

We describe measurements of atmospherically produced [10]Be in annually laminated sediments of the North American Varve Chronology (NAVC;  Ridge et al., 2012). These sediments were deposited in an initially proglacial and subsequently paraglacial lake adjacent to the southeast margin of the retreating Laurentide Ice Sheet (LIS) between 18,200 - 12,500 years BP. The NAVC records events taking place at the ice sheet margin during this time, including the position and retreat rate of the ice margin, meltwater and sediment production, and proglacial lake outburst floods (Ridge, 2004, 2012;

Ridge et al., 2012). Because some of these events are related to climate (Ridge et al., 2012; Thompson et al., 2017), the NAVC is also a climate proxy record with annual resolution.

  Our overall goal in investigating [10]Be deposition in NAVC sediments is to synchronize this record with Greenland ice core records that provide a primary template for our understanding of northern hemisphere climate change during the last deglaciation. The NAVC is a floating varve chronology that is anchored to the absolute time scale by radiocarbon dating

of plant macrofossils deposited in individual varves, and the uncertainty in this calibration is estimated to be no better than $\pm$ 200 years (Ridge et al., 2012; Thompson et al., 2017). When changes in ice-margin positions and meltwater flux recorded in the NAVC are compared with climate records in Greenland ice cores, it is evident that many events recorded by the NAVC appear to correlate with rapid temperature changes in Greenland occurring at boundaries for Greenland stadials and interstadials GS2a, GI-1e through 1a and GS-1 (Lowe et al., 2008). Improved precision in correlating these

two records, therefore, could potentially illuminate important aspects of ice sheet-climate feedbacks during rapid climate changes that took place during deglaciation, for example, the phasing relation between atmospheric temperature change in Greenland and the retreat rate of the southern margin of the Laurentide Ice Sheet in New England, or the effect of proglacial lake drainage into the North Atlantic Ocean on regional surface climate.

  In principle, it should be possible to synchronize these two climate records by comparing the deposition rates of

atmospherically-produced [10]Be. [10]Be is produced in the atmosphere by cosmic-ray bombardment of target nuclei, primarily N and O, and delivered to the surface by precipitation or dry fallout (Lal and Peters, 1967; Lal, 1987). The [10]Be production rate and, therefore, the deposition rate at the surface, vary significantly with changes in solar activity and geomagnetic field properties. Because these processes are global in nature, variations in the [10]Be fallout rate are, in general, globally synchronous, and records of the [10]Be deposition rate in ice cores and sediments have been extensively

used both to reconstruct past solar and geomagnetic changes (e.g., Heikkilä et al., 2013, and references therein) and to synchronize different records (e.g., Adolphi et al., 2018, and references therein). Beryllium-10 deposition rates have been measured in ice cores in Greenland and elsewhere in many studies (e.g., Beer et al., 1985; Heikkilä et al., 2013; Adolphi





et al., 2018, and references therein) and display significant centennial-scale variability during the time period represented by the NAVC.

Our goal here is to determine whether it is possible to generate a similar record of $^{10}$Be deposition from NAVC sediments that can be correlated with the Greenland records, thereby potentially improving the absolute dating of events

recorded in the NAVC and our ability to relate them to climate changes recorded in ice cores. To pursue this, we describe (i) basic observations relating to $^{10}$Be deposition and systematics in glacial and paraglacial lake sediments of the NAVC, (ii) experiments designed to determine whether or not variations in $^{10}$Be fallout due to solar variability are recorded in the NAVC, and (iii) a comparison between ice core $^{10}$Be fallout records and $^{10}$Be deposition records from portions of the NAVC.

**1.1  The North American Varve Chronology**

The NAVC consists of annually laminated sediments that were deposited in several lakes adjacent to the margin of the retreating Laurentide Ice Sheet (LIS) in New York and New England between approximately 18,500 and 12,500 years ago (Figures 1, 2). These sediments occur in numerous stratigraphic sections throughout the region that were originally correlated and assembled into several floating varve chronologies by Antevs (1922, 1928). These have subsequently

been consolidated, corrected in a few places, and extended into a single 5659-year sequence (Ridge, 2012; Ridge et al., 2012, and references therein).

The majority of the NAVC is composed of sediments deposited in proglacial lakes that occupied the north-south-trending Connecticut River Valley from central Connecticut to northern Vermont as the margin of the LIS receded northward. Although the extent of Connecticut Valley lakes varied with ice margin position, glacioisostatic rebound, and

changes in outlet position and lake level, some portion of the lake system was continuously present and in contact with the ice margin until approximately 13,700 years ago, at which point changes in the sedimentological characteristics of the varves show that the lake was no longer receiving direct glacial meltwater. The complete chronology includes 4183 glacial varves overlain by 1476 paraglacial varves, although the glacial-paraglacial transition is gradational over 100-200 years and also most likely not isochronous across the entire lake system.

Glacial varves are, in general, substantially thicker than paraglacial varves, and are derived predominantly from direct glacial meltwater and sediment production, although paraglacial sources including surface runoff from the recently deglaciated landscape adjacent to the lake may contribute. At a specific location in the lake, the contribution of paraglacial sediment to glacial varves increases upsection as the distance to the receding ice margin increases. Glacial varves, especially in areas proximal to the glacier, have melt season (summer) layers composed of a stack of micrograded beds

that often fine upwards. Summer layers grade into a non-melt season (winter) layer composed of nearly pure clay, of which 60% or more is typically less than 1 micron grain size. Glacial varve sections also occasionally have extremely thick varve couplets (over 50 cm) with single thick graded beds produced by the sudden release of water from proglacial lakes impounded in tributary valleys (see, for example, Thompson et al., 2017). Paraglacial varves are thinner, typically





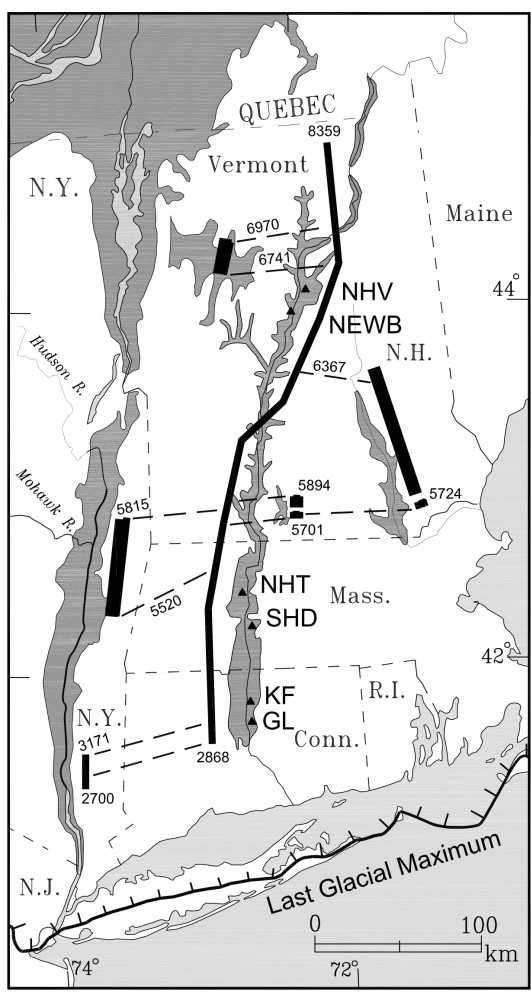

**Figure 1.** Map of New England and adjacent parts of the eastern United States and Canada showing the location and extent of the NAVC (Ridge et al., 2012). Labeled triangles show the location of sections sampled for this study (Table 1), with the exception of 930PN, which is located off the map to the west in central New York. Dark shaded areas show the maximum extent of major proglacial lakes present during deglaciation, although lake boundaries were time-transgressive throughout deglaciation and these do not represent exact lake boundaries at a specific time. Dark lines represent segments of the NAVC that are present in each lake, with bounding NAVC years for each segment. Although lake sediments at any particular point span a range of NAVC years, deglaciation of the area proceeded from south to north, so, in general, younger (higher-numbered) varves are found at more northerly sites (Figure 2).




millimeter-scale, and composed predominantly of glacial sediment deposited on the surrounding landscape, eroded following deglaciation and then subsequently transported to the lake by surface runoff.

Paraglacial varves in the NAVC are considered to be varves formed with zero or negligible direct glacial input, although the transition between glacial and paraglacial varves is typically gradational. They are composed of silty and fine sandy

5 summer layers, and may have occasional graded fine sand beds much thicker than a typical summer layer and that represent unusual runoff or flood events that washed sandy sediment onto the lake floor. Winter layers in paraglacial varves have diffuse contacts with summer layers below and sometimes show single, internal silt to fine sand laminations that may represent fall overturning.

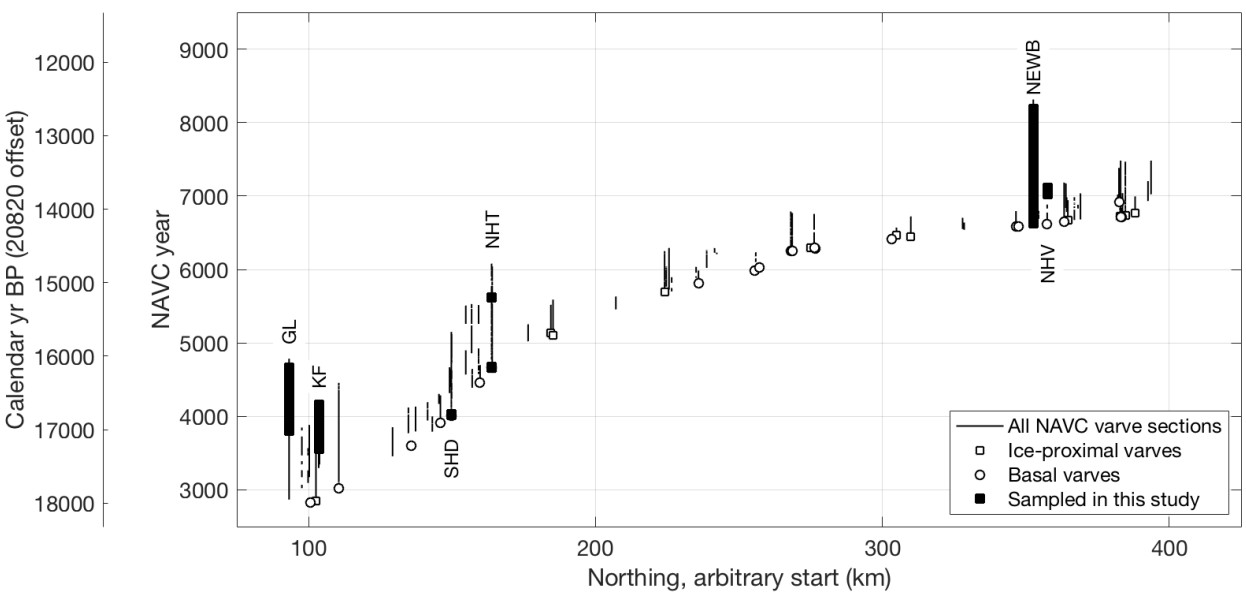

**Figure 2.** Time-distance diagram for NAVC varve sections, highlighting sections sampled in this paper. Thin lines show age range and location of varve sections in central New England that are correlated to the NAVC. Circles and squares highlight basal or ice-proximal varves, respectively, that indicate the location of the ice margin in a given varve year. Thick lines at labeled sections indicate portions of these sections analyzed in this study. As in Figure 1, the 930PN section in New York is not shown here. The calendar year BP time scale at left uses the value of 20820 yr for the NAVC yr - cal yr BP offset from Thompson et al. (2017); see discussion in section 1.2.

## 1.2 Existing calendar year calibration of the NAVC

10 Although the NAVC is continuous for its entire length, the lakes in which varves were deposited drained ∼12,000 yr BP, so the varve chronology does not have an absolute connection to the calendar year time scale. Given that the NAVC is linear with respect to true calendar years, that is, that it lacks gaps or spurious varves, the calibration of the NAVC to the



true calendar year time scale is represented by a single value for the offset between NAVC years and calendar years. The offset throughout this paper is defined as the age in years BP of NAVC year zero, where "years BP" means "years before 1950," as typically defined for purposes of radiocarbon age calibration. NAVC years are positive and increasing towards the present, and calendar years BP are positive and decreasing towards the present, so the offset is equal to

the sum of the NAVC year of any varve and the calendar year BP in which that varve was deposited. Although previous publications (e.g., Ridge et al., 2012; Thompson et al., 2017) variously report estimates for (NAVC-calendar year BP) and (NAVC-calendar years before 2000) offsets, in this paper we use only the NAVC-calendar year BP offset to minimize confusion. Later in this paper, we will also correlate the NAVC with ice core records on the GICC05 timescale (Andersen et al., 2006). Again, throughout this paper we apply the GICC05 timescale in years BP rather than years before 2000 as

sometimes used elsewhere. As the GICC05 timescale is not exactly linear with the true calendar year timescale because of inaccuracies in ice core layer counting (Adolphi et al., 2018, and references therein), NAVC-cal yr BP and NAVC-GICC05 yr BP offsets may not be exactly the same, and we will use both of these terms as appropriate throughout this paper.

The value of the NAVC-cal yr BP offset has been determined by fitting radiocarbon ages of plant macrofossils asso-

ciated with individual varves to the INTCAL09 (in Ridge et al., 2012)) and subsequently INTCAL13 (in Thompson et al., 2017) calibrated radiocarbon time scales. Thompson et al. (2017), using the INTCAL13 calibration, obtained a best estimate for the NAVC-cal yr BP offset of 20820 yr. The uncertainty in this estimate is not well quantified, because residuals between the measured $^{14}$C ages and those predicted by the best-fitting offset to INTCAL13 scatter significantly more than expected from measurement uncertainty (see discussion in Ridge et al. (2012) and below in section 4.2.4). Ridge et al.

(2012) estimated the uncertainty in the calibrated value of the offset to be approximately $\pm$ 200 yr, but also interpreted the distribution of the residuals to indicate that the most likely source of excess scatter was small amounts of sample contamination by postdepositional carbon. This, in turn, would imply that the true value of the offset is likely larger by ~100-200 years than the 20820 yr estimate.

Both Ridge et al. (2012) and Thompson et al. (2017) proposed that the estimate of the NAVC-cal yr BP offset could

be further refined by aligning prominent climate events evident in Greenland ice core records with prominent changes in varve thickness and/or ice margin position recorded in the NAVC. They matched climate events near 14,000 yr BP to yield a NAVC-GICC05 offset of 20820 yr, which is the same as the best-fitting offset inferred from matching $^{14}$C data to INTCAL13. This result implies a zero offset between INTCAL13 calendar years and GICC05 years at 14,000 yr BP, which is within the range of existing estimates (Adolphi et al., 2018). Ridge et al. (2012) also matched events in NAVC varve

thickness records with climate events in the range 16,000-18,000 BP to infer a NAVC-GICC05 offset of 20750 years at this time, which may be consistent with the observation that GICC05 likely undercounts ice layers in the 13,000 - 20,000 yr BP interval (Andersen et al., 2006; Rasmussen et al., 2008; Adolphi et al., 2018). However, as noted above, radiocarbon calibration of the NAVC still suggests that the NAVC-INTCAL13 offset may be ~100-200 years larger than 20820 yr. In addition, it is important to note that event correlation between the NAVC and ice core climate records relies on the

assumption that the responses of $\delta^{18}$O variations in Greenland ice cores and varve thickness records in New England to





climate changes have similar lag and relative amplitude, which has not been independently established. Also, matches based on event correlations were derived by looking for matches that were close to the offset already estimated from the $^{14}$C data. Both ice core and varve records display centennial-scale periodicity, so multiple matches could be possible in some time ranges.

## 1.3 Beryllium-10 in lacustrine sediments

Beryllium-10 fluxes to the Earth's surface in polar regions reconstructed from $^{10}$Be concentrations in ice cores have been widely used as a proxy for solar variability on interannual to centennial timescales (e.g., Beer et al., 1990; Yiou et al., 1997; Finkel and Nishiizumi, 1997; Steig et al., 1998), and $^{10}$Be fluxes to marine sediments have been used to infer paleomagnetic field strength changes on timescales from thousands to millions of years (e.g., Simon et al., 2016, and references therein). By comparison, relatively few studies have attempted to identify solar variability on decadal to centennial timescales using $^{10}$Be measurements on lacustrine sediments. Ljung et al. (2007) found that $^{10}$Be fluxes recorded in lacustrine sediments deposited between AD 900-1450 were similar to those recorded in Greenland ice cores for the same time period. More relevant to this study, Mann et al. (2012) and Czymzik et al. (2016, 2018) measured $^{10}$Be concentrations in annually laminated modern and late-glacial sediments, respectively, from European sites and concluded that variability in $^{10}$Be fluxes to the lake was attributable to both direct variations in fallout to the lake due to solar variability, and variability in environmental factors affecting transport and scavenging of fallout $^{10}$Be in the lake. These authors suggested several approaches to accounting for these factors in reconstructing the $^{10}$Be fallout flux, mainly focused on multiproxy analysis of lake sediment and the use of multivariate correlation analysis to isolate variability in $^{10}$Be flux that could not be explained by measured environmental proxies. In this work, we attempt to simplify approaches used in those studies by measuring both $^9$Be and $^{10}$Be in lake sediments and applying the following interpretive framework.

Our basic approach is to (i) use $^{10}$Be concentrations in sediment as a measure of both the delivery of background Be to the lake and any environmental factors affecting Be scavenging efficiency in the lake, and then (ii) identify additional variation in the $^{10}$Be flux that cannot be accounted for by these effects and must therefore represent variability in the atmospheric fallout flux. We quantify this with a simple model for Be fluxes to glacial lake sediments. Because we are working with annually laminated sediments, we can directly measure Be fluxes to the sediment (throughout this paper represented in units of atoms cm$^{-2}$ yr$^{-1}$) as the product of the Be concentration in sediment (atoms g$^{-1}$) and the mass accumulation rate (g cm$^{-2}$ yr$^{-1}$) computed from annual layer thickness (cm yr$^{-1}$) and density (g cm$^{-3}$).

We assume that measured Be fluxes to lake sediments derive from one source of $^9$Be and two sources of $^{10}$Be. First, sediment delivered to the lake either from glacial discharge or landscape runoff contains "inherited" $^9$Be and $^{10}$Be. Inherited $^9$Be includes adsorbed $^9$Be resulting from subglacial and subaerial mineral weathering. Inherited $^{10}$Be could arise from a number of processes, presumably mainly including subglacial recycling of sediment with adsorbed $^{10}$Be from a past history of surface exposure as well as interaction between subglacial sediment and fallout $^{10}$Be already present in ice sheet ice. We refer to all Be arising from all these sources collectively as "inherited" Be. Although our analytical procedure might extract in-situ-produced cosmogenic $^{10}$Be in mineral constituents of sediment, this can be assumed





negligible because on a per-area basis the in-situ production rate is three orders of magnitude less than the atmospheric deposition rate. Structural $^9$Be in Be-bearing minerals (e.g., beryl) is not extracted by our analytical procedure and is disregarded.

Because Be is insoluble and effectively adsorbed to sediment at neutral pH, we expect that sediment delivered to the lake by glacial meltwater discharge will include inherited $^9$Be and $^{10}$Be with a characteristic $^{10}$Be/$^9$Be ratio. The factors controlling the $^{10}$Be/$^9$Be ratio of glacial sediment are, presumably, complex, and include the Be concentration of sediment source material, water-rock interactions in the subglacial environment, the concentration of $^{10}$Be in ice, and the rate and extent of basal and surface melting. However, we assume that whatever these processes, subglacial sediment mixing is effective across a significant area of the ice sheet and results in a characteristic $^{10}$Be/$^9$Be ratio in subglacially

discharged sediment that can be assumed to be constant, or at least slowly varying, over the 100- to 1000-year timescales represented by our sample sets.

Second, once glacial sediment with a characteristic $^{10}$Be/$^9$Be ratio is delivered to a proglacial lake, additional $^{10}$Be can be supplied to the lake from atmospheric fallout and adsorbed to the sediment during transport or deposition, but $^9$Be cannot. Thus, lake sediment contains inherited $^9$Be, inherited $^{10}$Be, and fallout $^{10}$Be.

With these assumptions, Be concentrations in lacustrine sediment (which we can measure) can be related to the flux of atmospheric fallout $^{10}$Be (which we seek to reconstruct) by:

$$Q_{10,T} = Q_{9,T} R_S + Q_{10,A} f \qquad (1)$$

where: $Q_{10,T}$ is the total $^{10}$Be flux (atoms cm$^{-2}$ yr$^{-1}$) to the sediment, which is the product of the measured $^{10}$Be concentration (atoms g$^{-1}$) and the mass accumulation rate (g cm$^{-2}$ yr$^{-1}$); $Q_{9,T}$ is the corresponding total $^9$Be flux

(atoms cm$^{-2}$ yr$^{-1}$; calculated in the same way as $Q_{10,T}$); $Q_{10,A}$ is the flux of atmospheric fallout $^{10}$Be to the surface (atoms cm$^{-2}$ yr$^{-1}$), $R_S$ (nondimensional) is the characteristic $^{10}$Be/$^9$Be ratio of sediment supplied to the lake, and $f$ (nondimensional) is a focusing factor. The two terms on the right-hand side of the equation represent the two possible sources of $^{10}$Be described above: inherited $^{10}$Be ($Q_{9,T} R_S$) and fallout $^{10}$Be ($Q_{10,A} f$).

The focusing factor $f$ reflects the fact that $^{10}$Be deposition occurs throughout the lake and, perhaps, an adjoining

watershed on the ice sheet and/or the surrounding ice-free landscape, whereas sediment deposition only occurs in a fraction of the lake basin. Thus, in area-normalized units (atoms cm$^{-2}$ yr$^{-1}$), the deposition rate of fallout $^{10}$Be to lake sediment ($Q_{10,A} f$) is greater than the fallout rate from the atmosphere ($Q_{10,A}$) to the watershed. The focusing factor depends on the geometry of the lake and its watershed; for example, Ljung et al. (2007) estimated f $\simeq$ 20 for a small lake, Belmaker et al. (2008) estimated f $\simeq$ 15 for the much larger Lake Lisan, and the results of Aldahan et al. (1999) implied

f $\simeq$ 3 for the still larger Lake Baikal. Absent significant changes in the geometry of the lake and watershed, however, $f$ is expected to be constant or changing slowly, so because we are interested in the short-term variability of the atmospheric fallout flux rather than its absolute value, it is sufficient for our purposes to estimate the quantity $Q_{10,A} f$. Therefore, given the framework of Equation 1 and measured total $^9$Be and $^{10}$Be fluxes to the sediment, the primary obstacle to





reconstructing atmospheric fallout fluxes of $^{10}$Be is the need for an estimate of $R_S$, which we discuss in detail later from the perspective of analyzing each of our data sets.

Equation 1 represents a very simple model for $^{10}$Be supply to lake sediments, and several complications may be relevant for our data. One is that $R_S$ and $f$ will likely change over time, both as the fraction of sediment supplied by direct
subglacial discharge decreases and that supplied by runoff from the deglaciated landscape increases, and also during the transition from a glacial to paraglacial lake environment. As noted above, however, it is likely that these changes will either be (i) steady, gradual changes associated with ice margin retreat from the lake basin, or (ii) large instantaneous changes associated with lake inlet or outlet switching or reconfiguration of ice-marginal drainage. It appears unlikely that these would be manifested as periodic, decadal to centennial-scale, variations that might mimic the solar variability we seek to
identify. A second complication is the possible role of watershed processes; if fallout $^{10}$Be were sourced from a significant ice-free watershed surrounding the lake, variations in $^{10}$Be fallout rates might be buffered by water-soil interactions in the watershed and reduce observed fallout variability in the lake (e.g., Czymzik et al., 2016). Finally, the potential importance of $^{10}$Be already present in ice sheet ice that could be released by surface melt is unknown. The expected contribution from typical $^{10}$Be concentrations in ice sheets (order $10^4$ atoms g$^{-1}$) at moderate melt rates averaged over the ablation
zone (order 10 cm yr$^{-1}$) would be small (order $10^5$ atoms cm$^{-2}$ yr$^{-1}$) compared to the atmospheric fallout flux ($\sim$1.5 x $10^6$ atoms cm$^{-2}$ yr$^{-1}$). On the other hand, if melt rates averaged over the contributing ablation area reached meters per year, which could be possible during periods of rapid retreat, this contribution might be significant.

## 2 Methods

### 2.1 Sample collection

We collected several sample sets from a total of 7 varve sections, with various sampling strategies designed to address different aspects of the study (Table 1 and discussion below). Samples were extracted from (i) short cores collected from surface outcrops by cleaning outcrops and driving in short (typically 0.6-m) sections of PVC pipe with a steel cap and sledge hammer, and (ii) deep (up to 50 m), continuously sampled hollow-stem auger cores collected in 2007-2009. Cores are archived at Tufts University. All cores were split after collection and partially dried to improve visual identification
of layers and facilitate sampling of individual varves. Only cores with minimal coring deformation were sampled. We sampled seasonal layers of single varves, complete single varves, and continuous sets of contiguous varves by scraping core surfaces and/or cutting interior sections of the cores to remove any possible surface contamination, then separating varves along partings between layers and scraping top and bottom surfaces to isolate the varve(s) of interest. Each sample was a column or wedge cut such that the cross sectional area was constant for all varves incorporated in the
sample. As in previous work, we consider a single varve to extend from the bottom of the coarse-grained summer layer to the top of the fine-grained winter layer.





### 2.1.1 Short biennial records

At two sites, one with glacial varves (the Kelsey-Ferguson, or KF, section; Figs 1, 2; Table 1) and one with paraglacial varves (the North Haverhill, or NHV, section; Figs 1, 2; Table 1), we collected continuous sets of samples spanning 80-year periods at 2-year resolution. The purpose of these sample sets is to determine whether decadal-scale solar variability, specifically the ∼11-year Schwabe cycle, can be identified in $^{10}$Be fluxes to NAVC sediments (see section 4.1 below).

### 2.1.2 Long decadal records

At two sites, we collected sets of samples designed to produce records of $^{10}$Be deposition over 1000-1500-year periods with decadal (ca. 25-yr) resolution, that could potentially be correlated with $^{10}$Be records from ice cores (see section 4.2 below). One set of samples is from glacial varves at the Kelsey-Ferguson (KF) and nearby Glastonbury (GL) sites in central Connecticut (Figs 1, 2; Table 1) that overlap and together span NAVC years 3658-4675. The second set is a sequence of glacial grading into paraglacial varves from Newbury, Vermont (NEWB; Figs 1, 2; Table 1) that spans NAVC years 6631-8193.

### 2.1.3 Winter/summer pairs

To investigate the seasonal distribution of Be deposition, we separately analyzed winter and summer layers from individual varves at four sites (Figs 1, 2; Table 1): (i) glacial varves from the Kelsey-Ferguson (KF) site; (ii) paraglacial varves from North Haverhill (NHV); (iii) glacial varves at a site in North Hatfield, MA (NHT); and (iv) glacial varves from a section in the Mohawk Valley of central New York State (930PN) that is not correlated with the NAVC, but was deposited approximately 17,000 yr BP. Varves at 930PN are unusual among NAVC sites in that their parent material is limestone, so they are mostly carbonate.

### 2.1.4 Ice-proximal sediment and underlying till

To investigate inherited Be concentrations in subglacial sediment that serves as the parent material for varved lake sediments, we analysed basal ice-proximal varves and underlying diamicton at South Hadley (SHD) and North Hatfield, MA (NHT) from hollow-stem auger cores collected in 2009 that penetrated the base of these varve sections. The basal varves are relatively coarse, 2-6-cm-thick varves deposited at the bottom of these sections, presumably close to the ice margin. The diamicton at SHD is a stony subglacial till derived from Mesozoic red beds, whereas that at NHT mixes till and glaciolacustrine sediment and likely represents a subaqueous debris flow near the ice sheet grounding line.

## 2.2 Grain size measurements

We measured grain size distributions for all samples using a combination of mechanical sieving (for grain sizes larger than 4 Φ)and pipette analysis for smaller grain sizes (e.g., Lewis and McConchie, 2012). Throughout this paper we represent grain size using the logarithmic phi scale of Krumbein (1934), which is commonly used in sedimentology: Φ is the negative





base-2 logarithm of the grain diameter in millimeters, so larger values of Φ represent smaller grain sizes. We halted the pipette analysis at 11 Φ (~0.5 $\mu$m), but significant fractions (up to 50%) of all samples were finer than this (Figure 3). Thus, to calculate mean grain size and sorting parameters (see, e.g., Boggs, 2014) that require the complete distribution, we fit a log-normal distribution to the observed data for each sample and used the fitted log-normal distribution to compute mean grain size and sorting (Fig. 3).

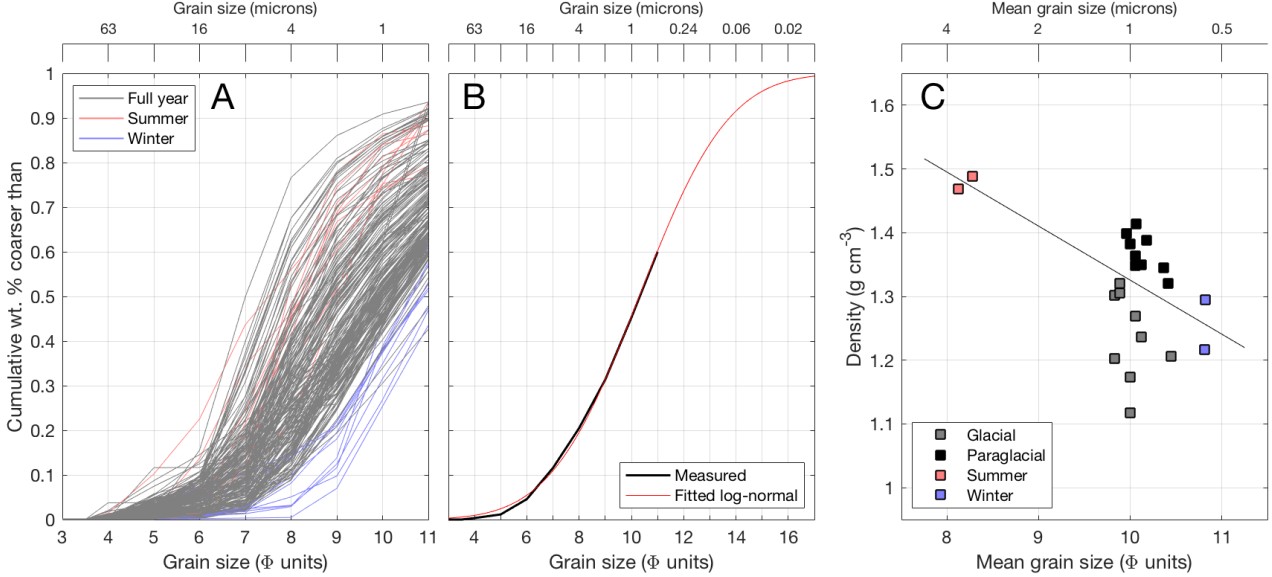

**Figure 3.** Grain size and sediment density measurements. **A**, measured cumulative grain-size distributions for summer layers (red), winter layers (blue), and complete single or multiple varves (gray). Pipette analysis stopped at 11 Φ. **B**, example of extrapolating measured grain-size data for a representative sample (NAVC 7068-69 from the NHV section) using a fitted log-normal distribution. **C**, sediment density-grain size relationship. The linear approximation shown by the black line (which is fit to all data excluding three low outliers that likely result from failure to completely fill the reference volume with stiff glacial sediment) is used in subsequent flux calculations.

## 2.3 Sediment density

Measurements of sediment density are required to interpret a Be concentration by weight (atoms g$^{-1}$) as a Be flux to the sediment (atoms cm$^{-2}$ yr$^{-1}$). We measured the dry density of representative samples of glacial varves at the Kelsey-Ferguson (KF) site, paraglacial varves from the North Haverhill (NHV) site, and separate summer and winter layers from North Hatfield (NHT) by pressing a cylinder of known volume into wet sediment and drying and weighing the resulting sample (Fig. 3; Supplementary Table S2). This procedure has several potential sources of inaccuracy, primarily due to the stiff and friable nature of the sediment, which causes difficulty in completely filling the standard volume without deforming





or compressing the sediment or introducing voids. It is important to note that the purpose of this study is to identify variations in Be flux to the sediment, not to make an accurate estimate of the absolute flux, so from this perspective inaccuracies in density measurements are not necessarily significant. On the other hand, if we assumed a constant density when, in fact, periodic variations in density were present, we could infer spurious Be flux variations. Density

variations, in turn, are most likely to be the result of variations in grain size and therefore in porosity and the relative proportion of silt and clay, and our measurements show this effect (Figure 3). Thus, for subsequent Be flux calculations we compute the density of all samples from mean grain size measurements using the linear relationship shown in Figure 3, with an uncertainty derived from the standard deviation of the residuals ($\pm$ 0.05 g cm$^{-3}$).

### 2.4   Beryllium-9 by ICP-OES

We measured concentrations of adsorbed $^9$Be using the procedure described in Greene (2016), which involves drying and powdering sediment samples, leaching in 6M $HCl$ for 24 hr, then diluting to a suitable concentration for analysis and measuring the Be concentration with a JY Horiba Optima inductively coupled plasma optical emission spectrometer (ICP-OES). Although the nominal internal precision of the ICP-OES measurement inferred from replicate analysis of liquid standards was typically 1-2%, replicate analyses of splits of the powdered sediment samples showed a total uncertainty

of 3.5%. Thus, we assume that all $^9$Be measurements have 3.5% uncertainty.

### 2.5   Beryllium-10 by accelerator mass spectrometry

We measured $^{10}$Be concentrations in sediment samples at the University of Vermont (UVM) using an adaptation of the potassium bifluoride fusion method of Stone (1998). We dried sediment samples, homogenized them by powdering in a Spex shatterbox mill, and subsampled 0.5 g of the homogenized sediment. We mixed the sample with $KHF_2$ and

$Na_2SO_4$, and spiked the mixture with 0.4 mg of Be carrier prepared from deep-mined beryl and having $^{10}$Be/$^9$Be between 2-8 x 10$^{-16}$. We then extracted Be by fusion of the sample, leaching of the fused sample in water, precipitation of excess K as $KClO4$, and precipitation of Be hydroxide. We modified the published method by adding an ion exchange step in which we redissolved Be hydroxide in dilute $HCl$, loaded it onto cation exchange resin (Bio-Rad AG50W X8), eluted B and other cations in dilute $HCl$, recovered Be in 6M $HCl$, evaporated to dryness, redissolved in dilute $HCl$, and precip-

itated Be hydroxide. Finally, we converted Be hydroxide to $BeO$ for Be isotope ratio measurement by accelerator mass spectrometry (AMS) at the Scottish Universities Environmental Research Centre (SUERC) in East Kilbride, Scotland. For these analyses, we sputtered samples and isotope ratio standards in the ion source for the same amount of time in order to suppress any effect of variations in emittance during sputtering. All $^{10}$Be measurements are normalized to the NIST SRM4325 standard material with an assumed $^{10}$Be/$^9$Be ratio of 2.79 x 10$^{-11}$, which is equivalent to the 07KNSTD

standardization of Nishiizumi et al. (2007). We processed samples in batches of 16, each containing one process blank and at least one replicate of a sample that was also analyzed in a different batch (see below). The process blank for Be fusion extraction at UVM utilizes a sample of fine-grained fluvial sediment chosen because of its naturally low bulk $^{10}$Be concentration, and subsequently powdered, homogenized, and etched in dilute $HNO_3$ in a heated sonic bath for an





extended period to remove any adsorbed Be. We processed 0.5 g aliquots of this material in the same way as the samples. Complete carrier and process blanks contained between 120,000 and 310,000 atoms $^{10}$Be, representing 0.1-0.7% of total atoms measured in samples.

To quantify total measurement uncertainty and as a general quality control measure, we prepared large quantities
of two representative samples to be analyzed repeatedly throughout the project. The first of these consisted of varves NAVC 3569-3570 from the Kelsey-Ferguson site, collected as part of the 80-year biennial record described above, which we refer to here by the internal UVM sample name BD3. When the BD3 sample was exhausted after 15 measurements, we prepared a second sample for replicate analysis (BD-STAN) by mixing and homogenizing aliquots of several other glaciolacustrine sediment samples from this study. The BD-STAN sample was subsequently analysed 6 times.

The purpose of chemical purification of Be for AMS analysis is to produce a sample of $BeO$ that is sufficiently pure to produce a Be ion beam of the same intensity as produced by pure reagent $BeO$ used as an isotope ratio standard. Poor or inconsistent Be yield from chemical purification, or failure to consistently remove contaminants from the $BeO$ target, can result in Be ion beam currents that are low and/or inconsistent compared to those from standards (see  Hunt et al., 2008). This is undesirable because (i) low beam currents reduce measurement precision by reducing $^{10}$Be count
rates, and (ii) inconsistent beam currents can introduce systematic errors in isotope ratio measurements due to current-dependent variations in beam emittance, ion beam transport, or detector efficiency. Although the goal of AMS beamline and detector setup and adjustment is to remove or minimize any beam current dependence of ratio measurements, it is difficult to verify whether this has been achieved for an arbitrarily large range in beam current, so instrument tuning is not a complete substitute for consistent sample preparation.

AMS analysis of Be samples prepared in the first eight chemistry batches showed low and inconsistent beam currents (Figure 4; mean and standard deviation of sample beam currents relative to standards for these batches were 0.63 $\pm$ 0.25). We found that this effect was related to trace Cl remaining in samples after hydroxide precipitation, so we further modified the Be extraction procedure by redissolving Be hydroxide in dilute $HNO_3$ and reprecipitating. Be beam currents from samples in 17 chemistry batches after this modification were higher and less variable (Fig. 4; mean and standard
deviation of relative sample beam currents after the modification were 0.87 $\pm$ 0.16).

Although this change to the sample preparation method significantly improved beam current performance, the complete data set displays high variability in Be currents. Thus, we used the results of replicate analyses, which spanned a range of Be currents, to look for and quantify any beam current dependence in the AMS results. In both replicate analyses and in analyses of groups of samples with similar $^{10}$Be concentrations, we observed a relationship between Be beam current and
apparent $^{10}$Be concentrations derived from AMS-measured $^{10}$Be/$^9$Be ratios (Figure 5). Although apparent concentrations do not display a significant correlation with beam current for samples with beam currents similar to standards (between ca. 80-110% of standard currents), our data include a much wider range of beam currents, and when all data are considered, significant correlations are present (Figure 5). We observed this effect to be persistent through many AMS measurement periods during a 3-year period, and similar to the current dependence observed for many replicate measurements of
in-situ-produced $^{10}$Be in the CRONUS-N quartz standard measured at the SUERC accelerator by Bierman et al. (2017)





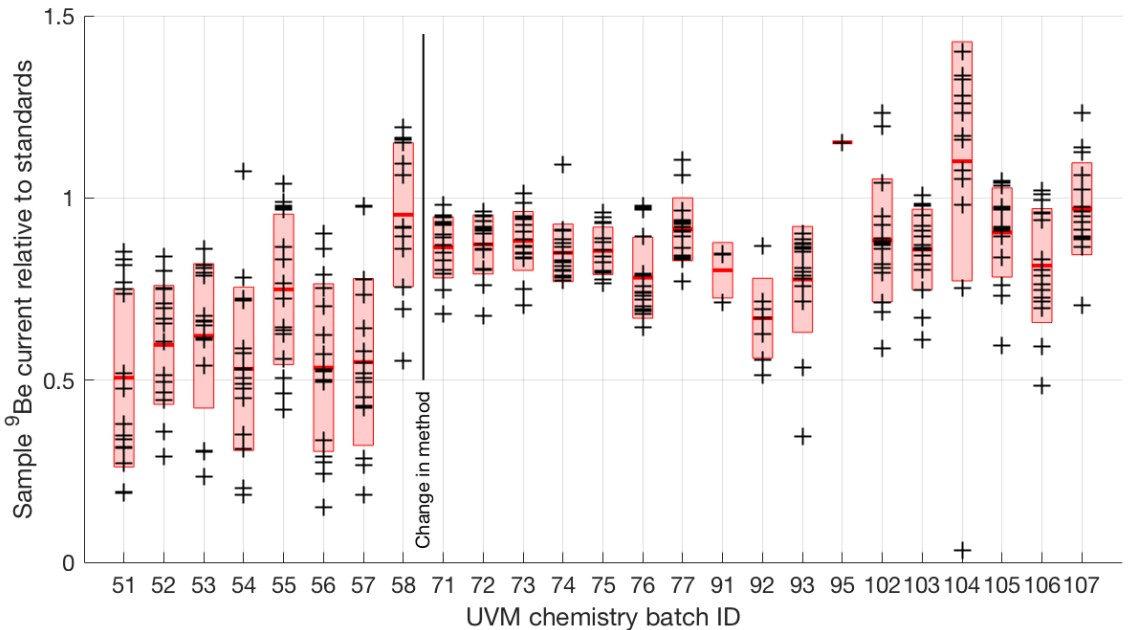

**Figure 4.** Be beam currents during AMS analysis at SUERC relative to Be isotope ratio standards. Each symbol shows the average beam current during analysis of a single sample, normalized to the average current of all primary standards analyzed during the same measurement period. Red lines and pink bars show means and standard deviations for samples processed in each chemistry batch. UVM batch ID numbers are assigned consecutively, so batches are shown in the order they were processed. Only samples of NAVC sediments that are part of this study are included in this plot; some batches included unrelated samples, which are not shown.

as well as for analysis of the CoQtz-N quartz standard at an anonymous AMS facility described by Binnie et al. (2019, ; their Fig. 3).

We conclude from these observations that apparent measured $^{10}$Be concentrations for our samples that had anoma-
5  lously low or high AMS beam currents are inaccurate due to beam current dependence of the measured isotope ratios. Lacking a physical model for the observed beam current dependence, we devised the following procedure to correct for this bias. If $(C/C_S)$ is the beam current for a sample relative to the average current for standards measured si-multaneously, $R_M$ is the apparent $^{10}$Be/$^9$Be ratio observed for the sample, and $R_C$ is a nominally correct $^{10}$Be/$^9$Be ratio that would be observed for $(C/C_S) = 0.9$, then we can define an empirical linear correction factor $S$ such that $R_M/R_C = S(0.9 - C/C_S)$. In the case where blank corrections are small to negligible (as is the case for these data; see
10  above) the ratio of apparent and corrected nuclide concentrations $N_M/N_C$, where $N_M$ and $N_C$ are nuclide concentrations in atoms g$^{-1}$, is equal to $R_M/R_C$. Thus, to account for small variations in sample and $^9$Be carrier masses among repli-cate samples, we correct apparent concentrations instead of ratios using $N_M/N_C = S(0.9 - C/C_S)$. 0.9 is chosen as the normalizing value because it is the modal value of $(C/C_S)$ for our complete data set, so minimizes required corrections.





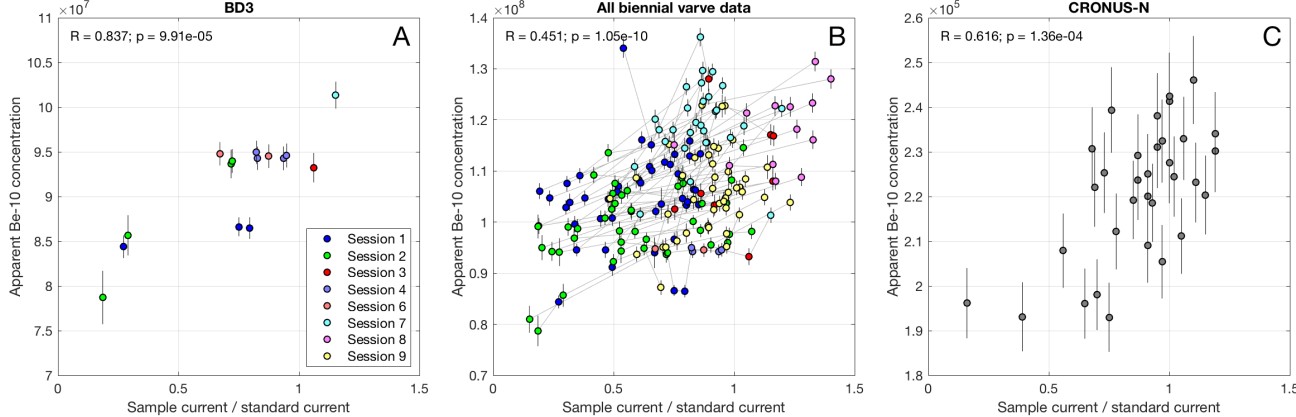

**Figure 5.** Observed relation between $^9$Be beam current relative to standards and measured $^{10}$Be/$^9$Be ratio, here expressed as nuclide concentration instead of ratio to account for small differences (ca. 5%) in sample and Be carrier masses between replicate analyses (see text). **A**, 15 replicate analyses of the BD3 sample (see text). **B**, replicate analyses of samples from 80-year biennial records at the KF and NHV sites (see text above); each of these samples was measured at least twice, and light-colored lines connect replicate analyses. Note that true $^{10}$Be concentrations vary among samples in this data set, but a significant correlation with beam current is present regardless, and replicates of individual samples display a positive slope in this diagram. In the left-hand two panels, symbol color-coding indicates the AMS measurement session in which each analysis took place. Sessions 1-2 were measured in 2012, 3-5 in 2013, and 6-9 in 2014. **C**, equivalent data for in-situ-produced $^{10}$Be in the CRONUS-N quartz standard (Jull et al., 2015) prepared in the UVM chemistry lab and measured on the SUERC accelerator between 2013-2017, as reported by Bierman et al. (2017).

Choosing a normalizing value not equal to one might introduce a small systematic bias in measured $^{10}$Be concentrations relative to the primary measurement standards, but that would not affect any of the results in this study because we are concerned with variability in $^{10}$Be flux and not with a precise measurement of the absolute flux.

   We estimate $S$ by choosing the value that minimizes scatter among replicate analyses of the same sample. The sample
5  with the largest number of replicates (BD3; see discussion above) yields $S$ = 0.164. The CRONUS-N data of Bierman et al. (2017) yield $S$ = 0.185, which highlights the apparent consistency of this effect across multiple samples, chemical preparation methods, and AMS measurement periods. Figure 6 shows the effect of applying this correction procedure with $S$ = 0.174 (the average of the above two values) for the data in Figure 5. Corrected data are uncorrelated with relative beam currents. Note that this correction scheme has the property that corrections for measurements with beam
10 currents comparable to standards ($C/C_S$ between ca. 0.8-1.1, where no significant correlation between current and ratio is observed) are minimal and comparable to individual measurement uncertainties; corrections only become significant for samples with extreme relative currents. Figure 7 further highlights this point for the 80-year biennial record from North Haverhill (NHV), which was entirely measured in duplicate. The correction procedure improves agreement between the two sets of measurements primarily by correcting a minority of data that were biased low due to anomalously low beam





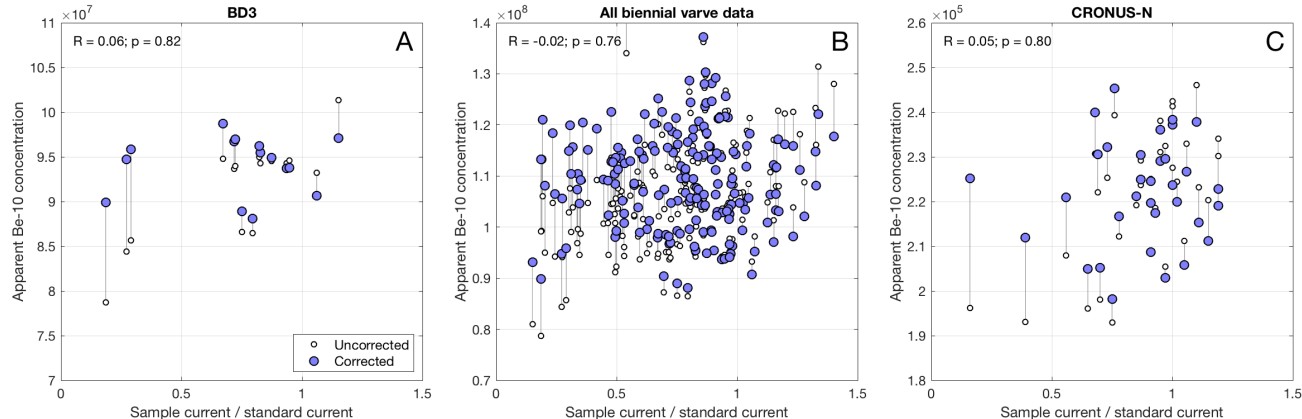

**Figure 6.** Correction for current dependence bias. Uncorrected data shown by open circles are the same data shown in Figure 5. Corrected concentrations shown by filled circles are connected by light lines to corresponding uncorrected concentrations; error bars are omitted for clarity. For measurements with beam currents comparable to standards, corrections are minimal and similar in size to estimated measurement uncertainties; corrections are most significant for measurements with extreme beam current values. The correlation coefficients and probabilities shown at upper left in all panels are for the corrected concentrations.

currents, without having a significant effect on the majority of the measurements. For the remainder of this paper, we correct all $^{10}$Be concentrations for beam current bias using $S = 0.174$.

Nominal measurement uncertainties on $^{10}$Be concentrations include: (i) AMS measurement uncertainties derived from machine stability and counting statistics ($\sim$1%) and replicate analyses of secondary isotope ratio standards during

AMS measurements periods ($\sim$1%); (ii) uncertainty in $^{9}$Be carrier concentrations ($<$1%); and (iii) and blank uncertainty ($\sim$0.2%). These range from 1.5-2% for most samples. However, as we found for ICP-OES $^{9}$Be measurements, reproducibility of $^{10}$Be concentrations in replicate aliquots of homogenized sediment samples (after correction for current dependence bias as described above) is poorer than expected from the estimated internal measurement uncertainties.

Figure 8 shows the reproducibility of replicate analyses of a total of 84 samples. The mean relative standard deviation

of all sets of replicate analyses is 4%, that for BD3 is 3.5% (for 15 measurements), and that for BD-STAN is 4.1% (for 6 measurements). We conclude that the true uncertainty in a given measurement from our sample set is 4%, which is significantly greater than nominal measurement precision. This likely reflects imperfect homogenization of sediment samples. Thus, we compute total uncertainties for all $^{10}$Be concentration measurements as follows. First, we assume that each individual measurement has 4% precision. Second, to incorporate the principle that repeated measurements

of the same sample should improve total measurement precision, for samples analysed more than once, we calculate a weighted mean and standard error (although, as all individual measurements are assigned 4% uncertainty, they are weighted equally). Summary concentrations and uncertainties calculated via this approach appear in the supplementary tables and are used henceforth throughout the paper.

**Figure 7.** Effect of current dependence bias correction on agreement between replicate analyses of NHV biennial samples. All data are assigned 4% uncertainties as discussed below.

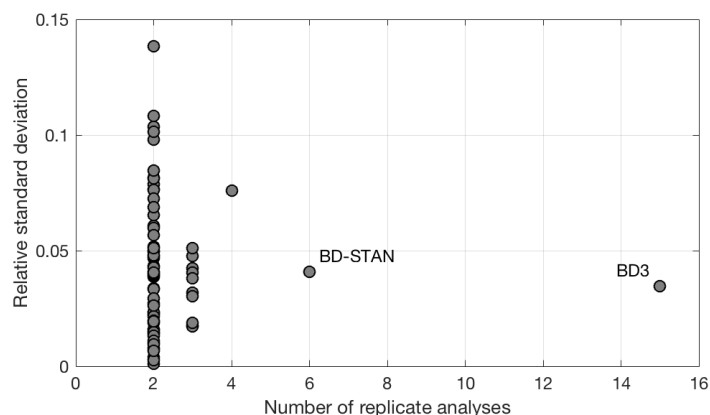

**Figure 8.** Relative standard deviation for analyses of replicate splits of 84 sediment samples that were analyzed more than once. Data have been corrected for current dependence bias.





## 3   Results: general framework and systematics

Beryllium-10 concentrations for all NAVC varves (Fig. 9; Supplementary Table S1) are in the range 0.5-1.5 x $10^8$ atoms $g^{-1}$, which is typical of concentrations observed in lacustrine sediments elsewhere (typically 1-10 x $10^8$ atoms $g^{-1}$; e.g., Ljung et al., 2007; Czymzik et al., 2016). Total leachable Be concentrations in sediments are 0.6-1.7 ppm, so $^{10}$Be/$^9$Be

5    ratios are in the range 1-2 x $10^{-9}$. As expected from the inverse relationship between grain size and surface area (e.g., Pavich et al., 1984), the dominant control on adsorbed Be concentrations is grain size: both $^9$Be and $^{10}$Be concentrations are inversely correlated with mean grain size at all sites (Figure 10). However, $^{10}$Be/$^9$Be ratios are not correlated with grain size, which is important because it supports the hypothesis used in formulating Equation 1 that inherited Be adsorbed to sediment has a characteristic $^{10}$Be/$^9$Be ratio.

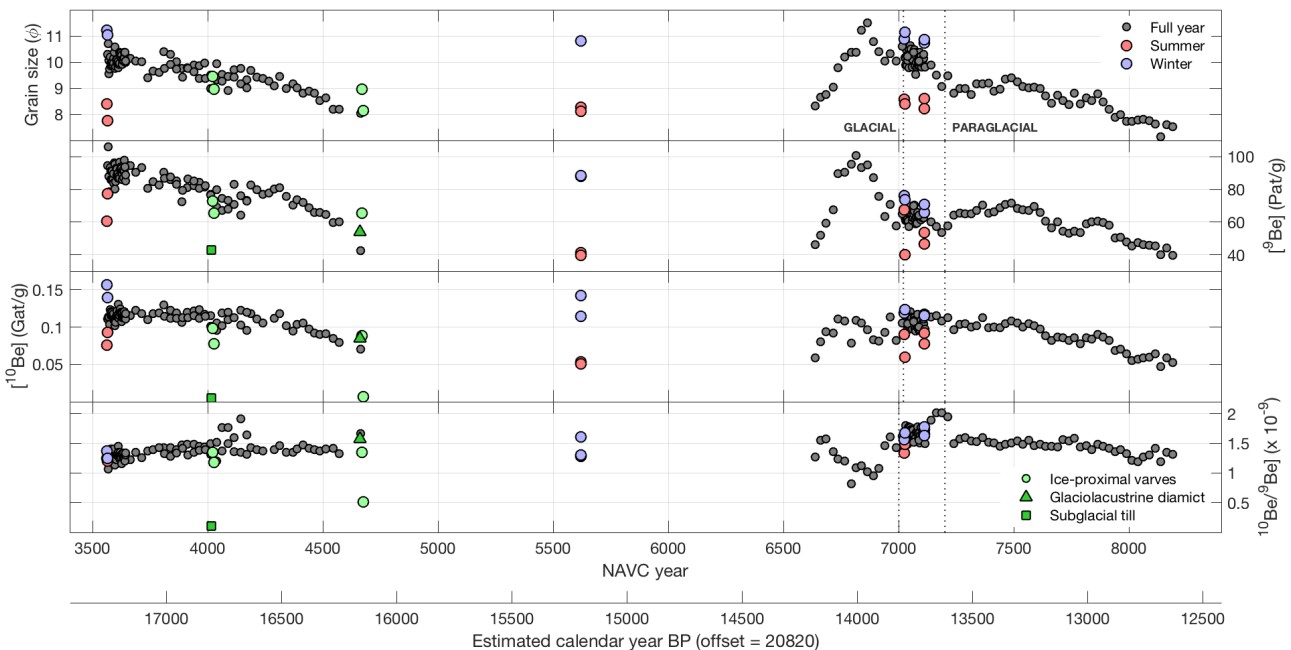

**Figure 9.** Mean grain-size and $^{10}$Be - $^9$Be concentrations for all samples, shown in stratigraphic order by NAVC year. Seasonal samples from site 930PN, which is not correlated with the NAVC, are not shown. The boundary between glacial and paraglacial varves is between NAVC 7020-7200, but is transitional at many sites and also not isochronous between sections. Error bars are not shown for clarity.

10    Variations in mean grain size, and thus in Be concentrations, at our sample sites reflect a balance of two primary effects. Initially, retreat of the ice margin from a site and therefore an increase in distance from the glacial sediment source will cause an upsection decrease in mean grain size. Subsequently, water depth decreases at the site due to sediment infilling of the lake and glacioisostatic rebound relative to an ice-distal outlet. Shallowing increases the energy of the depositional





environment, both because of increased exposure to surface waves and because the decreasing cross-sectional area of the lake requires that current velocities increase to maintain constant discharge, and leads to an upsection increase in grain size. An additional possible effect that may be relevant at some sites is that as the ice margin retreats, sediment is increasingly sourced from the surrounding deglaciated landscape rather than direct ice sheet discharge, and paraglacial runoff sediment is likely coarser than subglacial sediment.

**Figure 10.** Correlation analysis for grain size and Be isotope concentrations and ratio. The correlation coefficient and p-value are shown in red when data are correlated at 95% or better confidence. [9]Be and [10]Be concentrations are always significantly correlated with mean grain size. However, the [10]Be/[9]Be ratio is not. Note that $\Phi$ size is inverse to dimensional grain size, so a positive correlation between Be concentration and $\Phi$ is a negative correlation between Be concentration and grain size.



The long record at Newbury that we sampled with decadal resolution (NAVC 6631-8193; Fig. 9) shows these effects. Grain size decreases and Be concentrations correspondingly increase in the initial part of the record (NAVC 6600-6900) as the ice margin recedes from the site, but subsequently lake shallowing becomes more important and these trends reverse (NAVC 6900-8100). The long records at the Kelsey-Ferguson and Glastonbury sites (NAVC 3658-4669; Fig. 9),

on the other hand, begin in relatively ice-distal varves, because these sites were deglaciated at about NAVC 2800. Thus, grain size increases and Be concentrations decrease throughout, following the pattern in the upper half of the Newbury section.

Measurements of Be concentrations in separate summer layer - winter layer pairs from individual varves (Figs. 9,10) highlight the grain-size dependence of Be concentrations: finer-grained winter layers have much higher Be concentrations

than summer layers, but $^{10}Be/^9Be$ ratios are not significantly different. In addition, summer/winter data display the same grain size-concentration relationship as full-year data (Fig. 10). Although the winter and summer samples define the end members of the distribution in grain size-concentration space, seasonal and the full-year data lie on the same trend.

Samples of ice-proximal glaciolacustrine sediment and underlying diamict (see Section 2.1.4) have similar Be concentrations and isotope ratios to more ice-distal glaciolacustrine sediment in similar stratigraphic positions (compare the

green symbols in Fig. 9 to other data), with one significant exception. The exception is a subglacial till (from the SHD section; green square in Fig. 9). In contrast to all other samples, the properties and stratigraphic position of this sample indicate that it has no glaciolacustrine input and has most likely not interacted with lake water, and it has much lower $^9Be$ and $^{10}Be$ concentrations. Lower Be concentrations are consistent with the relatively large mean grain size (1.1 $\Phi$ compared to 8-11 $\Phi$ for glaciolacustrine samples, although with much poorer sorting). However, the much lower $^{10}Be/^9Be$

ratio ($1 \times 10^{-10}$) in this sample than in glaciolacustrine sediments ($\sim$1-2 x $10^{-9}$; see Fig. 9) is not necessarily expected, and may indicate that the majority of inherited $^{10}Be$ in glaciolacustrine sediments is not reflective of recycling of previously exposed sediment but instead may derive from interaction within the subglacial water system between till and $^{10}Be$ that is present in ice and released by melting.

## 4   Results and discussion: biennial and decadal data series

### 4.1   Biennial series

The purpose of sampling two 80-year varve sequences at 2-year resolution is to determine whether or not the 11-year Schwabe solar cycle is present in $^{10}Be$ flux records. This cycle is evident in ice cores (e.g., Steig et al., 1998) and some lake sediments (e.g., Mann et al., 2012). Recognizing this cycle in the $^{10}Be$ flux to NAVC sediments would be strong evidence that the NAVC does, in fact, contain a record of global variations in $^{10}Be$ production and fallout.

An 80-year segment of the paraglacial varve sequence at North Haverhill (NHV; Figure 11) has a mean varve thickness of 0.3 cm, corresponding to mass accumulation rates in the range 0.25-0.75 g cm$^{-2}$ yr$^{-1}$. As discussed above, $^{10}Be$ and $^9Be$ concentrations are correlated with mean grain size (Figs. 10, 11). Multitaper spectral analysis (Fig. 11;  Dettinger



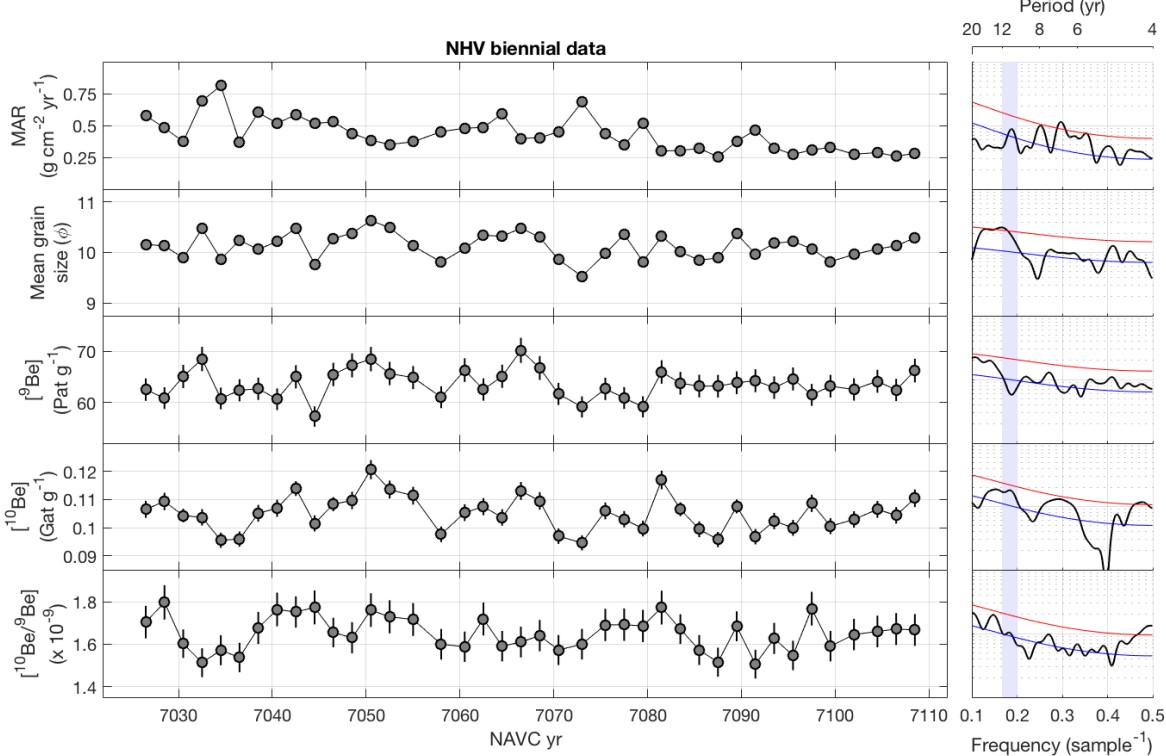

**Figure 11.** Sedimentological data and Be concentrations for 80-year record with 2-year resolution from paraglacial varves at North Haverhill (NHV) site. Mass accumulation rate (MAR) is calculated from varve thickness and the density-grain size relationship shown in Fig. 3. Plots at right show results of multitaper method (MTM) spectral analysis of the observational data series using the SSA-MTM Toolkit (Dettinger et al., 1995; Ghil et al., 2002), highlighting frequencies in the vicinity of the 11-year Schwabe solar cycle. Blue and red lines show 50% and 90%, respectively, significance thresholds relative to an AR(1) red noise estimate, and the shaded area highlights the 11-year solar cycle band.

et al., 1995; Ghil et al., 2002) does not show significant spectral power in the 11-year band for any of the observed data series except mean grain size, for which a 12-year peak is significant at 90% confidence.

The second 80-year sequence is from glacial varves at the Kelsey-Ferguson (KF) site (Fig. 12) and has a mean varve thickness of 0.85 cm and mass accumulation rates of 0.5-2.5 g cm$^{-2}$ yr$^{-1}$, more than twice that for the paraglacial varves at NHV. Although all the data from KF (shown in Figure 10) display significant correlations between grain size and both $^9$Be and $^{10}$Be concentrations, in this subset of the data, $^9$Be concentrations are significantly correlated with mean grain size (r = 0.87; p < 0.01) but $^{10}$Be concentrations are not (r = 0.18; p = 0.28). None of the observational data series show significant spectral power in the 11-year band.



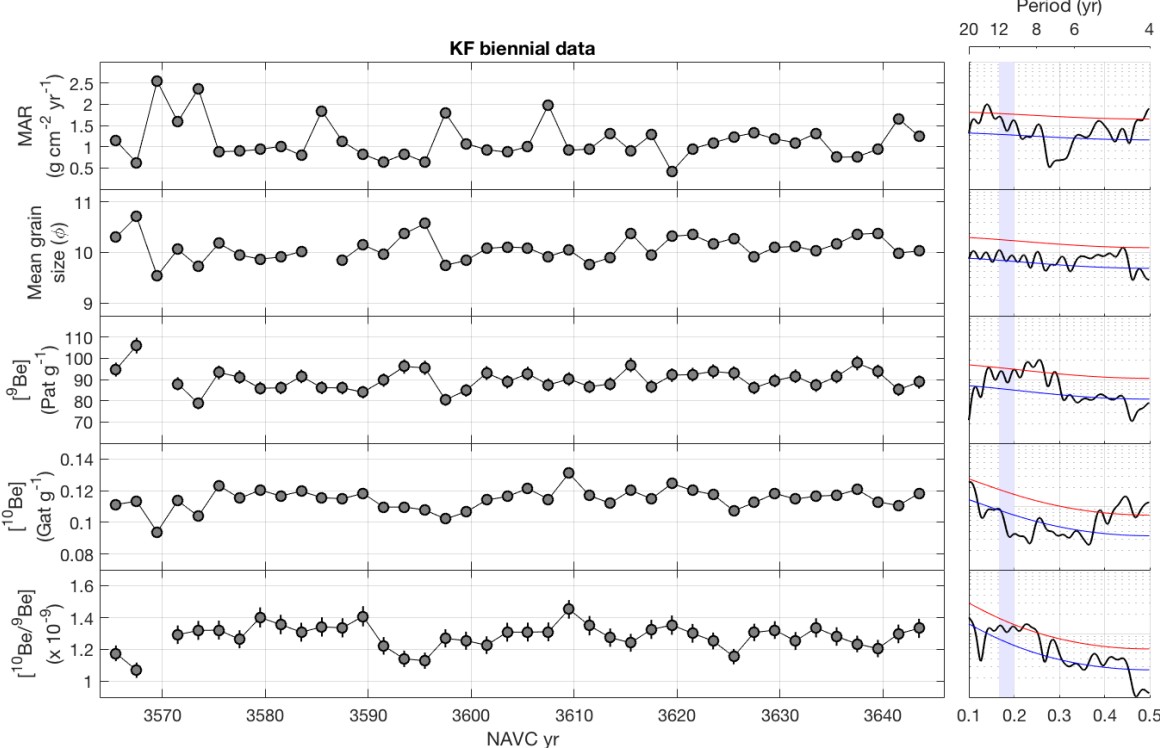

**Figure 12.** Sedimentological data and Be concentrations for 80-year record with 2-year resolution from glacial varves at Kelsey-Ferguson (KF) site, with MTM spectra. Plot elements are the same as described in Figure 11. For purposes of spectral analysis, missing grain-size and $^{10}$Be data for two samples were filled by linear interpolation between adjacent measurements.

Be isotope fluxes to the sediment can be derived from the data in Figs. 11 and 12 and are shown in Figs. 13, 14, and 15 for the KF and NHV biennial data. As discussed above, Be flux has units of atoms cm$^{-2}$ yr$^{-1}$ and is computed by multiplying Be isotope concentrations (atoms g$^{-1}$) and mass accumulation rates (g cm$^{-2}$ yr$^{-1}$). Following Equation 1, the $^9$Be flux is inherited $^9$Be adsorbed to sediment entering the lake, whereas the total $^{10}$Be flux includes both inherited

5    and fallout components.

We are interested in reconstructing the fallout $^{10}$Be flux ($Q_{10,A}f$ in Equation 1), and here we attempt to do this by applying Equation 1 with the following assumptions. First, we assume that the inherited $^{10}$Be/$^9$Be ratio $R_S$ is constant throughout the time series, which, as discussed above, appears likely over short periods of time. Second, we assume that the fallout flux $Q_{10,A}f$ is either (i) constant, or (ii) variable with a symmetrical distribution around the mean and

10   and also uncorrelated with $Q_{9,T}$. Assumption (ii) is likely true if short-term variability in the fallout flux is periodic, and processes affecting sediment supply to the lake are otherwise unaffected by solar cycle variations. The latter assumption




is supported by the fact that Rittenour et al. (2000) did not observe 11-year-band spectral power in NAVC varve thickness records, although spectral analysis of grain size variations in the NHV series (Fig. 11) may suggest otherwise.

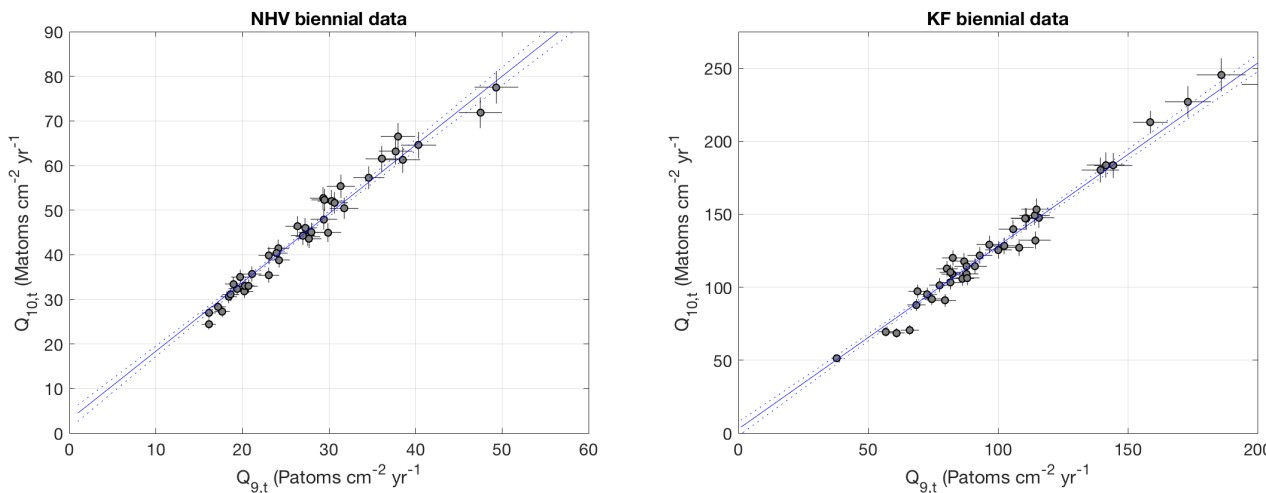

**Figure 13.** Plots showing regression of $^{10}$Be against $^9$Be fluxes for NHV and KF biennial data. Following Equation 1, the y-intercept should give the mean $^{10}$Be fallout flux. It is evident that, although this is above zero (as it should be), it is small compared to total $^{10}$Be fluxes and comparable in magnitude to measurement uncertainty and scatter around the line. The dotted lines show a 68% confidence interval for the regression derived from a Monte Carlo simulation.

With these assumptions, Equation 1 predicts that linear regression of $(Q_{9,T}, Q_{10,T})$ pairs will yield a line having slope equal to $R_S$ and y-intercept equal to the mean fallout flux. Sample-to sample variation in the fallout flux is expressed as

scatter around the line. $Q_{9,T}$ and $Q_{10,T}$ are, in fact, linearly related for both biennial records (Figure 13). Linear regression for NHV yields $R_S$ = 1.539 ± 0.077 x 10$^{-9}$ and mean($Q_{10,A}f$) = 3.0 ± 1.9 x 10$^6$ atoms cm$^{-2}$ yr$^{-1}$. The regression for KF yields yields $R_S$ = 1.252 ± 0.054 x 10$^{-9}$ and mean($Q_{10,A}f$) = 3.0 ± 4.9 x 10$^6$ atoms cm$^{-2}$ yr$^{-1}$. However, these results imply several complications. First, although intercepts are positive for both data sets, neither are distinguishable from zero at high confidence, and both imply that the fallout $^{10}$Be flux is a small fraction ($< 10\%$) of the total $^{10}$Be flux

at these sites. This is not favorable from the perspective of reconstructing solar-cycle variability in $^{10}$Be fallout fluxes, because the expected magnitude of variability in $^{10}$Be fallout associated with the 11-year cycle is ∼10-20% (Lal and Peters, 1967; Steig et al., 1998). This would correspond to variability in total measured $^{10}$Be fluxes no greater than 1-2%, which is smaller than typical measurement uncertainties of ∼3%. Second, in agreement with this prediction, Figure 13 shows that measured $^{10}$Be fluxes at both sites are indistinguishable from the regression line at the level of measurement

uncertainties; in other words, from this analysis alone we cannot reject the hypothesis that there is zero variation in the $^{10}$Be fallout flux and deviations from the linear relationship are entirely due to measurement errors. Both of these observations indicate that relatively high concentrations of inherited $^{10}$Be and relatively low concentrations of fallout $^{10}$Be





result in a poor signal-to-noise ratio for 11-year period variations in fallout [10]Be, and a low likelihood of detecting these variations.

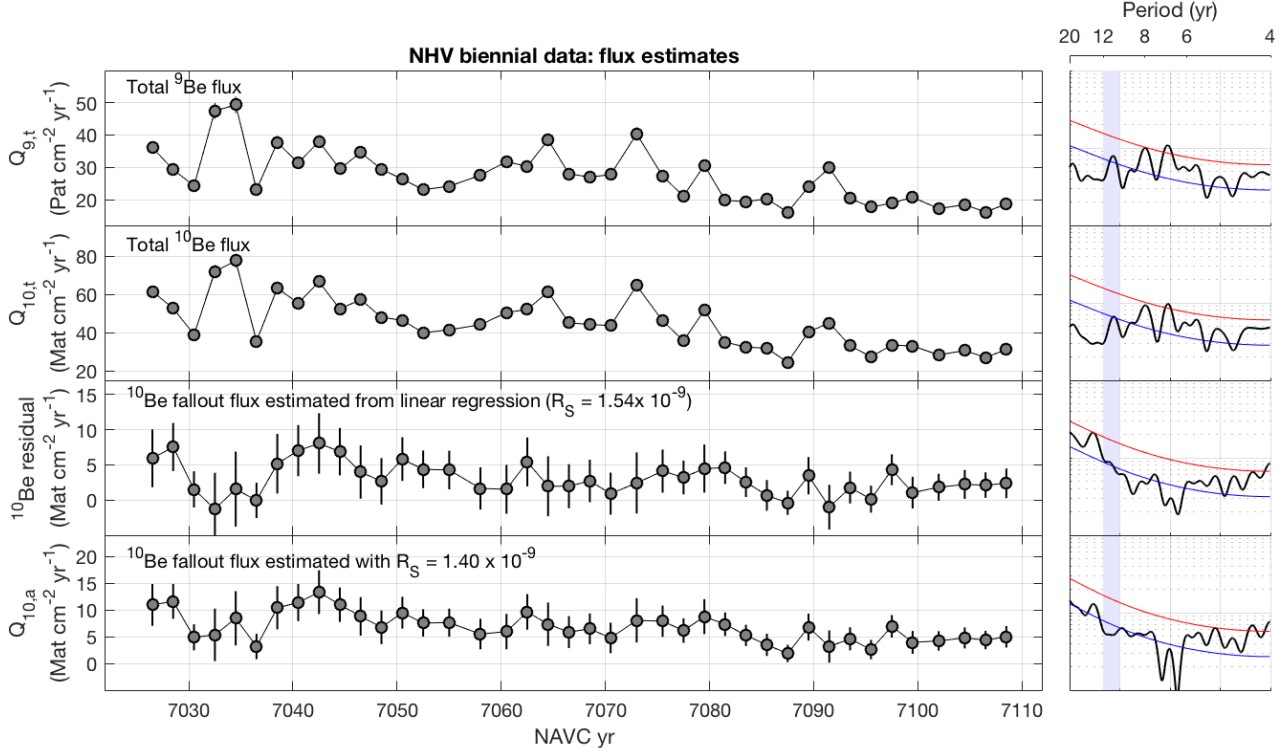

**Figure 14.** Be flux estimates for 80-year biennial record at NHV. Total fluxes are calculated using the grain size - sediment density relationship shown in Figure 3. [10]Be tallout fluxes are estimated from Equation 1 using two different values of $R_S$: the value inferred from the linear regression in Fig. 13, and a second value chosen to yield higher apparent fallout fluxes and highlight that apparent time variations in the flux are not strongly dependent on the choice of $R_S$. Error bars reflect propagation of uncertainty on Be isotope concentrations, but not on mass accumulation rates, so may be underestimated. Plot elements for MTM spectra are the same as in Figure 11.

Figures 14 and 15 show estimates of the [10]Be fallout flux for the biennial data sets using the estimates of $R_S$ derived from the linear regressions in Figure 13. Note that because the scatter of data around the regression lines is larger than

5   the mean fallout flux estimated from the y-intercept, this calculation results in some values for the fallout flux that are less than or indistinguishable from zero. In addition, this calculation yields fallout fluxes that scatter by ∼60% around the mean, which is more than expected for the 11-year solar cycle. These observations suggest that the regression method may be underestimating the [10]Be fallout flux. A systematic underestimate might not matter if we are only concerned with whether or not periodic variability exists, but to account for this possibility, we repeated the calculation with values for $R_S$ that are

10   lower than inferred from the regressions in Figure 13 and chosen to yield mean fallout flux estimates that all exceed zero



(Figs. 14, 15). Although there is no strong basis for choosing these values, they yield estimates of the [10]Be fallout flux that have less relative variability, and highlight the property of Equation 1 that although the choice of $R_S$ affects the mean estimated [10]Be fallout flux, it has minimal effect on its pattern of variability. This, in turn, is important because it implies that an inaccurate estimate of $R_S$ is most likely not an obstacle to spectral analysis of short-term variability or correlation

5 with ice core records. On the other hand, Figures 14 and 15 also highlight that propagated measurement uncertainties in the estimates of the [10]Be fallout flux, especially for the paraglacial NHV data, have the same scale as the short-term variability in the estimates, which may indicate that the unexpectedly large magnitude of short-term variation in the [10]Be fallout flux is not the result of underestimating $R_S$, but rather reflects measurement uncertainty.

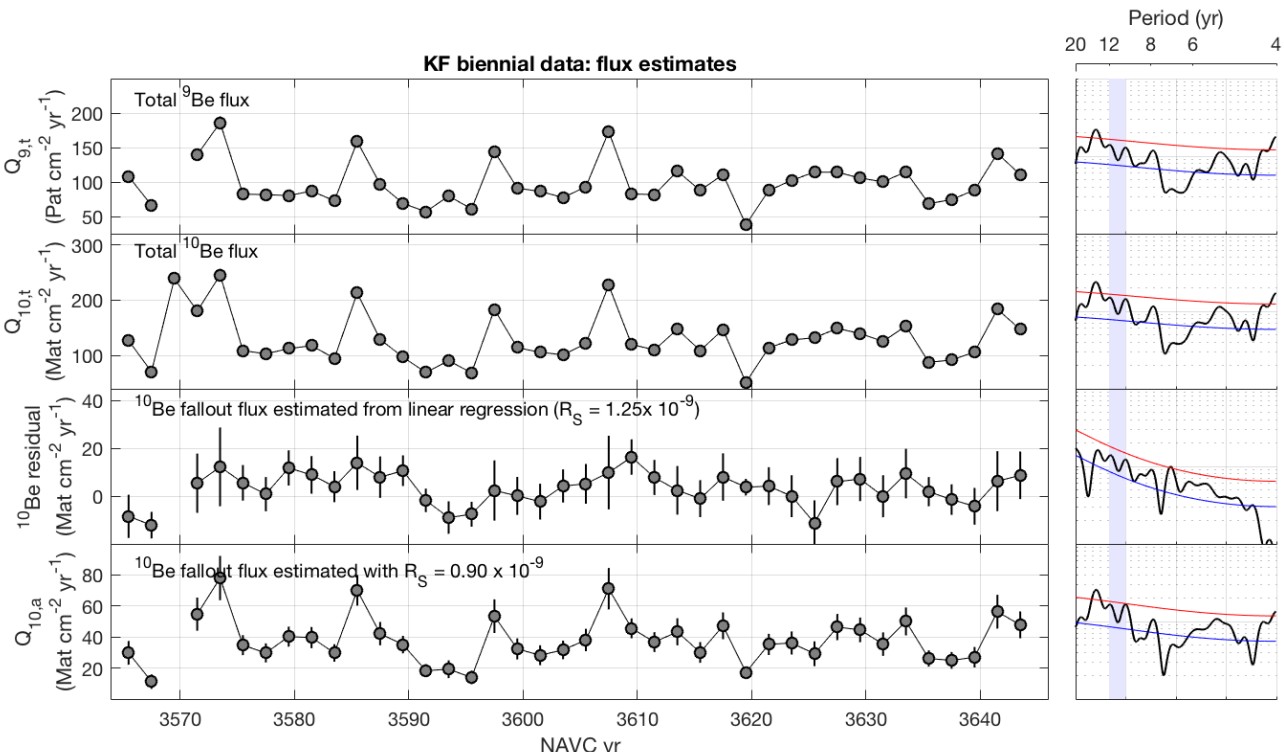

**Figure 15.** Be flux estimates for 80-year biennial record at KF. Total fluxes are calculated using the grain size - sediment density relationship shown in Figure 3. Fallout fluxes are estimated from two different assumed values of $R_S$; see text for details. Error bars reflect propagation of uncertainty on Be isotope concentrations, but not on mass accumulation rates. Plot elements for MTM spectra are the same as in Figure 11.

Spectral analysis of total [10]Be fluxes for both biennial records does not display significant spectral power in the 11-year
10 band (Figs. 14, 15). [10]Be fallout flux estimates display spectral peaks near this band in both records, but the significance level of these peaks varies with the value of $R_S$ used to compute the fallout fluxes. That is, $R_S$ derived from the regression





analysis at NHV (Fig. 14) yields a 14-year peak that is significant at 90% confidence; a lower value of $R_S$ yields the same peak, but at lower significance. At KF (Fig. 15), on the other hand, a lower value of $R_S$ increases the significance level of peaks in the 10-14 year band. Overall, spectral analysis of these records does not provide strong evidence for significant variability in $^{10}$Be fallout on the 11-year Schwabe cycle band.

We conclude from the 80-year biennial records that because of the relative magnitudes of inherited and fallout $^{10}$Be fluxes at these sites, the 11-year solar cycle can neither be confidently identified or excluded in these data. At these sites, the inherited $^{10}$Be flux represents the majority of the total $^{10}$Be flux and is much higher than the $^{10}$Be fallout flux. Expected variability in $^{10}$Be concentrations associated with 11-year band solar variability is small in relation to measurement uncertainties, which in turn makes it difficult to recognize such variability in $^{10}$Be fallout flux estimates
derived from our measurements. Therefore, our strategy of using the 11-year solar cycle to test the hypothesis that $^{10}$Be fallout variability is recorded in NAVC sediment did not yield a significant result: we have not proven conclusively that 11-year variability is present, but we also showed that, because of the poor signal-to-noise ratio induced by relatively high concentrations of inherited $^{10}$Be, we cannot prove that it is absent.

## 4.2   Decadal series

We now describe results from sampling two ~1000-year varve sequences at decadal resolution. The purpose of these sample sets is to determine whether or not centennial-scale variability in $^{10}$Be fallout fluxes evident in ice core records can be identified in NAVC sediments and, if identified, matched to ice core $^{10}$Be records. During the time periods represented by these sequences, ice core $^{10}$Be records show centennial variations in $^{10}$Be fluxes of up to a factor of two, substantially greater than expected from the 11-year solar cycle. Thus, it is more likely that these longer-period variations could be
distinguished from measurement noise in NAVC sediments.

### 4.2.1   Decadal series of glacial varves from Glastonbury and Kelsey-Ferguson sites

Overlapping sequences of glacial varves from the Glastonbury (GL) and Kelsey-Ferguson (KF) sites in north-central Connecticut (Figures 1, 2; Table 1; Figure 16) span NAVC years 3658-4675, approximately corresponding to 17,200-16,200 years BP. This set of samples has 20-30 year spacing, each individual sample includes 10-15 contiguous varves,
and samples collected in the range of overlap of the two sections sample nearly exactly the same varves, although some pairs of samples are offset by 1-2 years. KF is 8 km north of GL and therefore more ice-proximal, but varves at KF are also slightly thinner because the site is shallower and therefore subject to a lower sedimentation rate. The interval sampled was deposited while the glacier was at least 30 km to the north of KF. The mean grain size of matching samples from the same set of varves is coarser at KF, which is consistent with the site being shallower and more ice-proximal (Figure
16). $^{9}$Be concentrations are not significantly different between paired samples, but $^{10}$Be concentrations are systematically slightly higher at GL, resulting in $^{10}$Be/$^{9}$Be ratios that are systematically higher at GL than KF in the period of overlap between the two sample sets. Although most of the observed variability in the $^{10}$Be/$^{9}$Be ratio in both records is similar



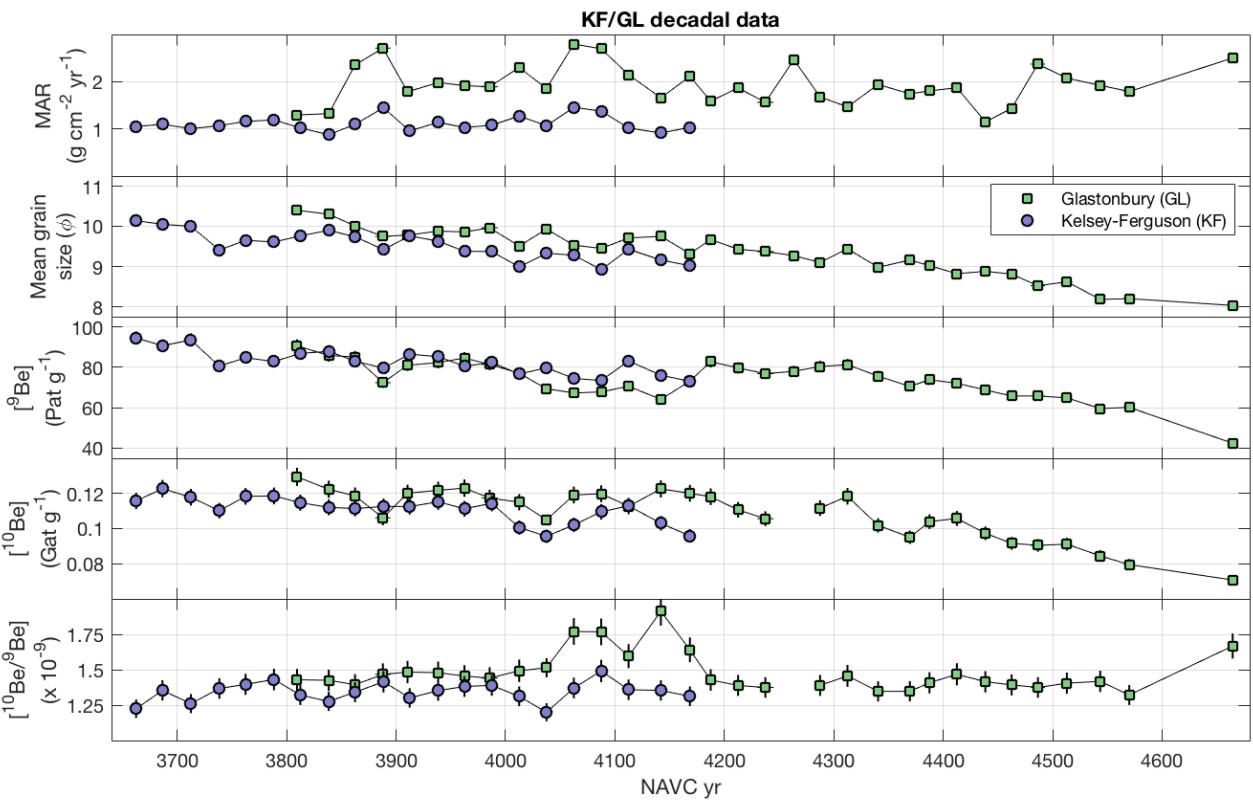

**Figure 16.** Sedimentological data and Be concentrations for decadal-resolution records from adjacent, overlapping sequences of glacial varves at Glastonbury (GL) and Kelsey-Ferguson (KF).

in magnitude to measurement uncertainties, both sites show a perturbation in $^{10}$Be/$^{9}$Be ratio that exceeds measurement uncertainty at NAVC 4000-4100.

Figure 17 shows estimates of $^{10}$Be fallout flux for these sites. To calculate $^{10}$Be fallout flux for the 80-year records discussed in the previous section, we estimated $R_S$ from linear regression of $^{9}$Be and $^{10}$Be fluxes by assuming that $R_S$ is

5     constant and that variations in the $^{10}$Be fallout flux are uniformly distributed around some mean as well as independent of variations in the total Be flux. However, for a 1000-year record, these assumptions are not likely to be true. In particular, changes in $R_S$, $Q_{9,T}$, and $f$ are likely to occur on time scales of hundreds of years, as the ice margin recedes from a site and the sediment grain size, the size of the contributing watershed, and the balance of glacial and paraglacial sediment all change. In fact, regression of $^{9}$Be and $^{10}$Be fluxes for the long records at both KF and GL yield unphysical negative

10    intercepts, which is consistent with the prediction that $R_S$ is not constant throughout this record.



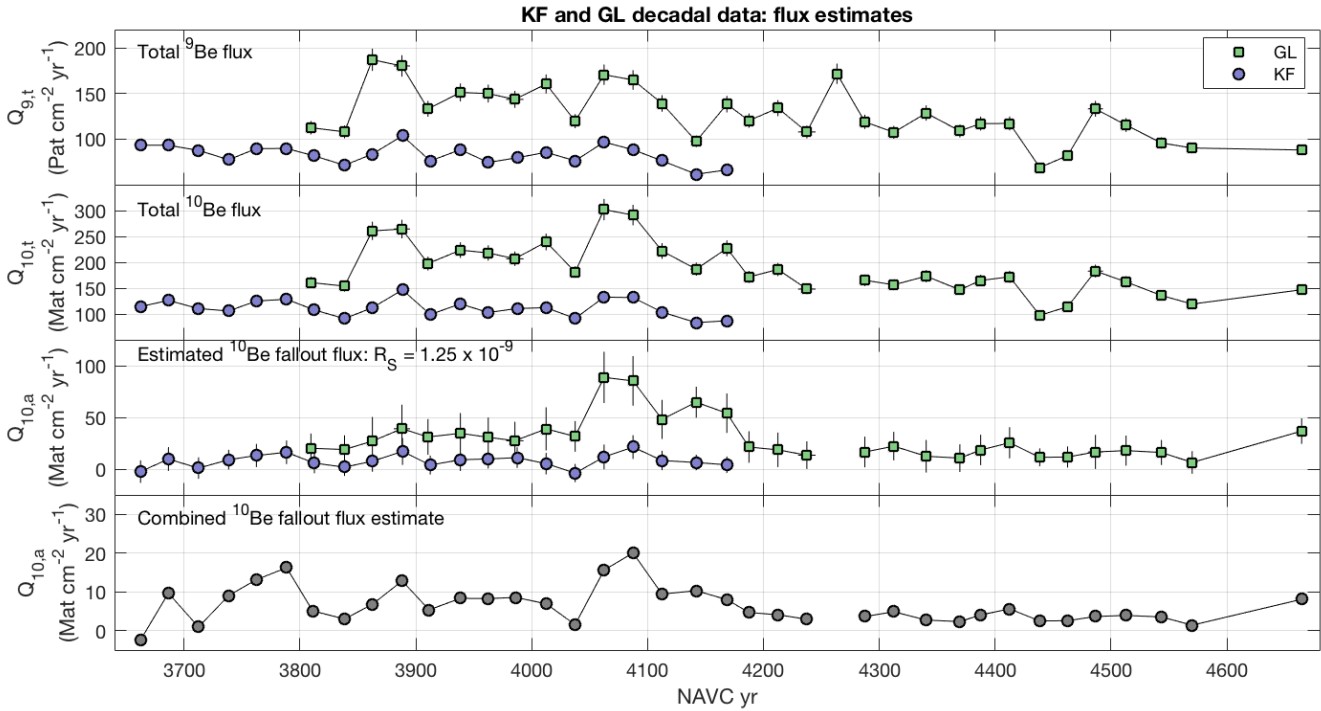

**Figure 17.** Be fluxes calculated for KF and GL decadal records. Upper panels, total $^9$Be and $^{10}$Be fluxes to sediment. Third panel, $^{10}$Be fallout flux estimates using Equation 1 for constant $R_S$. Lower panel, combined estimate computed by linearly scaling estimate from KF to force agreement in the period of overlap of the two records.

Thus, lacking any other constraints on $R_S$, we estimated $^{10}$Be fallout fluxes using the value of $R_S$ inferred from linear regression of $^9$Be and $^{10}$Be fluxes for the biennial data at KF (Fig. 17; also see section 4.1 and Fig. 13). The resulting estimates (Fig. 17) show similar short-term variability, although different magnitudes: for the period of overlap of the two records during which flux estimates are relatively constant between NAVC 3850-4000, the mean estimated flux at GL

5   is near 1 x $10^7$ atoms cm$^{-2}$ yr$^{-1}$, whereas at KF it is near 3.5 x $10^7$ atoms cm$^{-2}$ yr$^{-1}$. Given an approximate average fallout flux for temperate mid-latitudes of 1.5 x $10^6$ atoms cm$^{-2}$ yr$^{-1}$(Graly et al., 2011,  and references therein), these imply f $\simeq$ 6 for GL and f $\simeq$ 20 for KF, which are within the range observed for lakes elsewhere. This difference is simply a consequence of the fact that $^{10}$Be concentrations are systematically higher at GL. It could be an artifact of the calculation if $R_S$ is not, in fact, the same at both sites, although the fact that the sediment at both sites was sourced from the same

10  ice margin makes this unlikely. Alternatively, it could represent a real difference in the delivery rate of fallout $^{10}$Be to the sediment. For example, as GL is farther from the ice margin, the sediment deposited there could have a longer residence time in the lake, which may allow for more complete scavenging of fallout $^{10}$Be. If this hypothesis is correct, it could imply



that (i) the $^{10}$Be fallout flux to the sediment $Q_{10,A}f$ may be quite variable throughout the lake at any given time, and also (ii) variations in sediment transport within the lake could cause variations in the fallout flux at a given site that were unrelated to the actual atmospheric fallout rate. On the other hand, the similarity in fallout flux estimates between the two sites, including reproducibility of perturbations near NAVC 3880 and 4050-4100, is evidence that true variations in

atmospheric fallout are likely being recorded.

For purposes of comparing $^{10}$Be fallout flux estimates with variations in global fallout observed in ice cores, we assume that the difference between the two records is most likely explained by differences in $f$ between the two sites, and therefore obtain an estimate of centennial-scale variability represented in both records by applying a constant scaling factor to the GL data so as to align the two records where they overlap between NAVC 3810-4040, then averaging the

two records within the period of overlap. Multitaper spectral analysis of this combined record shows a significant (at 95% confidence) spectral peak at 91 years, which matches a characteristic period observed in solar variability records (the 88-yr 'Gleissberg cycle,' e.g., Feynman and Fougere, 1984). This, like the similarity of the two records, suggest that variability in our flux estimate is not solely the result of measurement noise, and that true centennial-scale variability in $^{10}$Be fallout may be recorded. Our combined estimate has higher internal variability ($\sim$50%; see lowest panel of Figure 17) than is

observed in ice core $^{10}$Be flux records for this time period ($\sim$15%). This could be an artifact of the flux estimate and scaling procedure, but it could also reflect focusing effects within the lake.

Figure 18 compares our combined $^{10}$Be fallout flux record to ice core $^{10}$Be fallout flux records compiled by Adolphi et al. (2018). The overall structure of the ice core records in this time period displays periodic centennial-band variability as well as a broader low between $\sim$16500-16800 GICC05 yr BP. The ice core records are shown in years BP (that is, before

1950) on the GICC05 timescale. If we start with the NAVC-calendar year BP offset of 20820 yr inferred from $^{14}$C data (Ridge et al., 2012; Thompson et al., 2017) and accept the estimate of Adolphi et al. (2018) that the GICC05 chronology is 175 yr too young in this time interval, then we expect a NAVC-GICC05 yr BP offset of (20820-175 = 20645). Aligning the composite $^{10}$Be fallout estimate from KF and GL with the ice core $^{10}$Be stack (Figure 18, panel B) with a NAVC-GICC05 yr BP offset of either 20820 or 20645 does not yield a good match: a peak in $^{10}$Be fallout flux near NAVC 4100 aligns

with a period of relatively low flux in the ice core stack, resulting in an inverse correlation between the two records. Ridge et al. (2012), based on event correlation between NAVC varve thickness records and Greenland ice core temperatures, suggested NAVC-GICC05 yr BP offsets of 20825 for events near 16100 yr BP and 20750 for events near 17400 yr BP, and neither of these yields a good match either.

We searched for a better match by looking for the closest alternative value of the offset for which the KF/GL combined

estimate is positively correlated with the ice core stack at high confidence (Fig. 18, panels C and D). This occurs for NAVC-GICC05 yr BP offsets in the range 20965-21165 yr, with the best correlation at 21110 years. This aligns the fallout peak at NAVC 4100 in the KF/GL record with a broad peak at 17,000 GICC05 yr BP in the ice core records. However, significant correlations occur over a fairly wide range of offsets, and in addition the results of this correlation are quite dependent on which ice core record is used. For example, a NAVC-GICC05 offset is better correlated with the GRIP $^{10}$Be



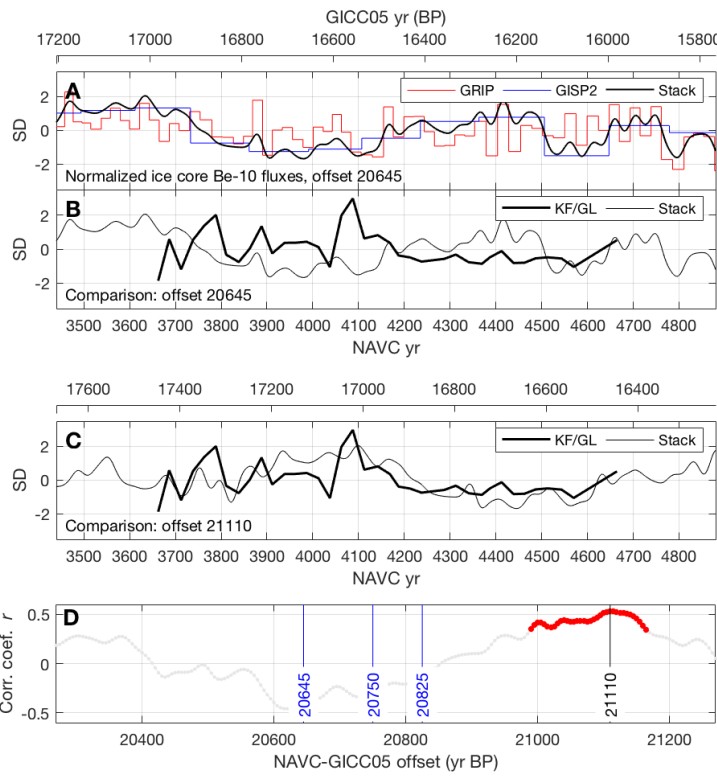

**Figure 18.** Comparison of the combined $^{10}$Be fallout flux estimate from the KF/GL sites (Fig. 16) with ice core records of $^{10}$Be fallout. **A**, scaled and centered ice core $^{10}$Be flux records from (Adolphi et al., 2018, and references therein), offset to NAVC years with a NAVC-GICC05 yr BP offset of 20645 yr (see text). "Stack" is the global ice core stack from Adolphi et al. (2018). **B**, scaled and centered combined KF/GL fallout flux estimate compared to the ice core stack with the same 20645 yr offset. **C**, same as B, only for an offset of 21110 yr. **D**, results of correlation analysis comparing the scaled and centered KF/GL fallout flux estimate with the ice core stack. Red highlights values of the offset where the two are positively correlated at 95% confidence. Vertical lines highlight offsets inferred from radiocarbon dating of NAVC varves (20820, or 20645 with the GICC05 - calendar year offset of Adolphi et al., 2018 for this time period) and event correlation of varve thickness with Greenland ice core temperature (20825 and 20750; Ridge et al., 2012). All of these yield an inverse correlation between the $^{10}$Be records.





record than the stack due to a prominent peak in the GRIP record at 16750 GICC05 yr BP that is not well expressed in the stack.

Overall, our [10]Be fallout flux estimate from the KF/GL decadal series (i) shows variations that are reproducible between two sites, (ii) shows evidence for periodic variability in a band characteristic of solar variability, and (iii) can be matched with the global ice core [10]Be fallout stack using an NAVC-GICC05 offset that is close to the expected value. These observations suggest that our fallout flux estimate is, in fact, recording global atmospheric [10]Be production and fallout variations. On the other hand, the apparent correlation between the two records mostly reflects alignment of long-period variability, specifically (i) the difference between higher average flux before NAVC 4150 and lower after and (ii) millenial-scale variability in the ice core stack, in this case a broad high near 16200 GICC05 yr BP and a broad low near 16700. Apparent correlation of ~1000-year period variations in a record that is only 1000 years long could easily be spurious if the assumption of constant $R_S$ or the scaling and centering procedure created a spurious systematic drift. Thus, although this match between NAVC and ice core fallout records is possible, it is not highly convincing.

### 4.2.2 Decadal series of glacial and paraglacial varves from Newbury

Our decadal-resolution record from Newbury comprises a sequence of glacial varves transitioning to paraglacial varves between NAVC 6630-8190 (Figure 19). This set of samples has 20-30 year spacing, and each individual sample includes 10-20 contiguous varves. Ridge et al. (2012) placed the boundary between glacial and paraglacial varves at NAVC 7100, where relatively thick varves with relatively high interannual variability (high MAR) are overlain by thinner varves (lower MAR) with reduced interannual variability (Fig. 19). However, grain size and Be concentration records (Fig. 19) do not show the same sharp transition, indicating that the glacial-paraglacial transition was more likely characterized by a gradual reduction in glacial meltwater and sediment delivery beginning near NAVC 7100 and continuing for ca. 200 years. Regardless, sediments in the lower part of this section before ca. NAVC 7000 are predominantly derived from direct glacial discharge, and sediments in the upper part after ca. NAVC 7225 are predominantly derived from runoff from the deglaciated landscape. A period of low MAR beginning at NAVC 6800 is associated with the Littleton-Bethlehem Readvance, a period of ice margin expansion into the lake recognized from a moraine complex and related stratigraphic sections ~40 km to the north of this site (Ridge and Toll, 1999; Ridge et al., 2012; Thompson et al., 2017). Although the distance to the ice margin presumably decreased at this time, this effect was offset by reduced meltwater production in a cooler climate that resulted in thinner, finer-grained varves.

Be isotope systematics are very different in glacial and paraglacial parts of the Newbury section (Figures 20, 21). Although, as noted above, the assumptions necessary to estimate $R_S$ from linear regression of [9]Be and [10]Be total fluxes are probably not valid for long records, regressions for glacial and paraglacial parts of the section in Figure 20 differ in a way that is similar to that observed for the paraglacial and glacial biennial records at NHV and KF, and are consistent with expected changes from a glacial to paraglacial environment. Glacial varves imply $R_S = 1.03 \times 10^{-9}$ and mean($Q_{10,A}f$) = $2.3 \times 10^7$ atoms cm$^{-2}$ yr$^{-1}$(e.g. f $\simeq$ 15), and [10]Be fluxes display high variability around the regression line, which implies that fallout [10]Be represents a significant fraction of the total [10]Be flux. Paraglacial varves imply $R_S =$

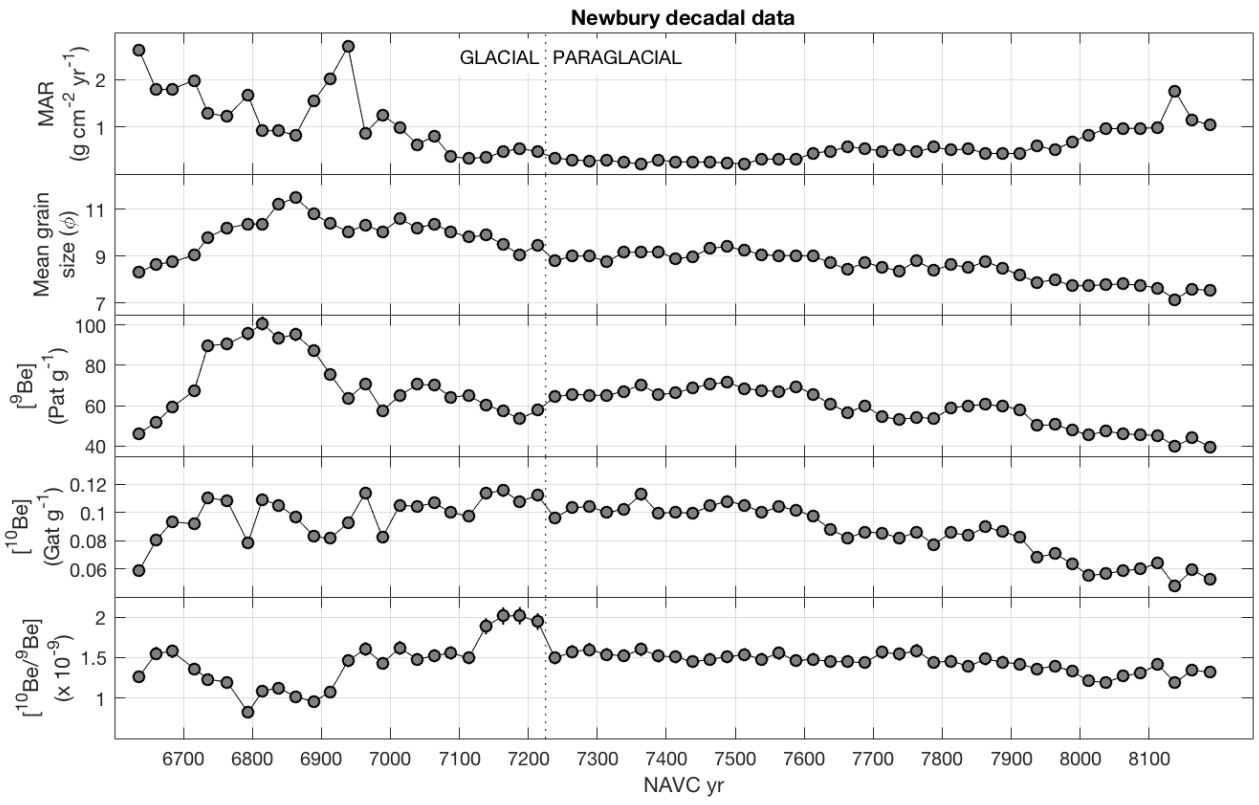

**Figure 19.** Sedimentological data and Be concentrations for decadal-resolution record at Newbury (NEWB; see Figs. 1 and 2).

1.2 x $10^9$ and mean($Q_{10,A}f$) = 8.4 x $10^6$ atoms cm$^{-2}$ yr$^{-1}$(e.g., f $\simeq$ 5), with minimal variability. This is consistent with our hypothesis, discussed above, that $R_S$ is expected to be higher for a deglaciated landscape, that may have been accumulating $^{10}$Be in soils for thousands of years, than for direct glacial sediment discharge. The decrease in $f$ between glacial and paraglacial varves at the same site is also consistent with the prediction that $f$ should be larger if the lake

5    watershed includes a significant area of the adjacent ice sheet, so the glacial-paraglacial transition is associated with a significant decrease in contributing area.

Although this analysis highlights differences in $^{10}$Be systematics between glacial and paraglacial varves, application of the estimates of $R_S$ from the regression slopes to Equation 1 would imply unphysical negative $^{10}$Be fallout fluxes for several samples. Thus, the estimate of $^{10}$Be fallout flux in Figure 21 uses $R_S$ = 0.9 x $10^{-9}$. Again, the assumed value for

10    $R_S$ affects the mean of calculated fallout fluxes, but does not affect reconstructed short-term variability in the flux as long as $R_S$ is assumed to be slowly varying.



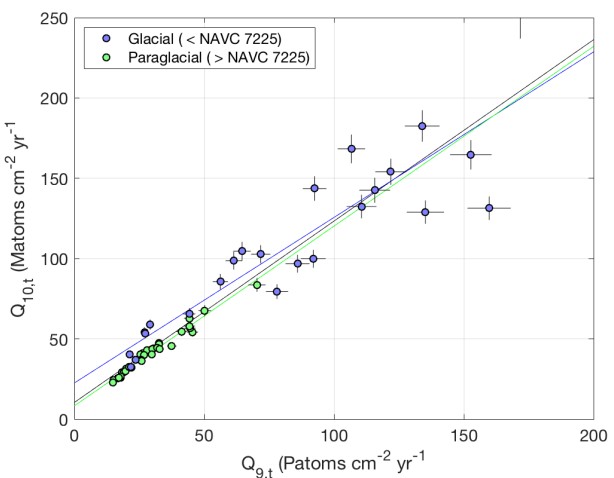

**Figure 20.** Regression of $^9$Be and $^{10}$Be total fluxes for glacial and paraglacial varves at Newbury.

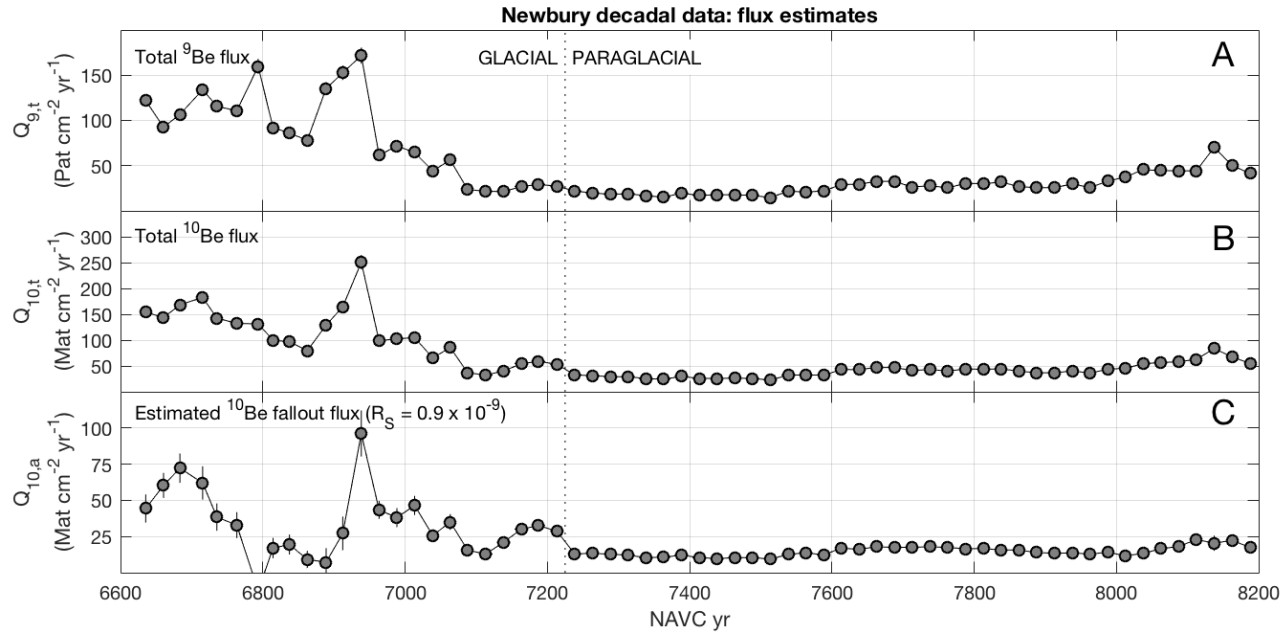

**Figure 21.** Be fluxes calculated for Newbury decadal record. **A** and **B**, total $^9$Be and $^{10}$Be fluxes to sediment. **C**, $^{10}$Be fallout flux estimate using Equation 1 for constant $R_S$.





The most striking aspect of $^{10}$Be fallout flux estimates for Newbury is that variability in $^{10}$Be fallout flux is strongly suppressed in the paraglacial section. This is a likely consequence of buffering of $^{10}$Be fallout by watershed processes in the absence of direct glacial meltwater input. As $^{10}$Be is known to accumulate in soils in nearly all environments (Pavich et al., 1984), in a paraglacial landscape, a substantial fraction of $^{10}$Be fallout deposited in the watershed is likely

adsorbed to developing soils during infiltration and therefore cannot be transported into the lake. Pollen records from this time period show that the landscape was heavily vegetated (Thompson et al., 2017), which highlights the likelihood of rapid soil development and $^{10}$Be accumulation in soils in the watershed. On the other hand, $^{10}$Be deposited on the surface of an ice sheet with negative mass balance, as presumably was the case for a large extent of the southern margin of the rapidly retreating Laurentide Ice Sheet at this time, cannot accumulate. Fallout $^{10}$Be deposited on an ablation area of the

ice sheet that drains into a glacial lake must all be flushed into the lake, either by supraglacial or subglacial drainage, during the summer melt season each year. Thus, if a large area of the ice sheet drains into the lake, fallout $^{10}$Be from a large area can be effectively concentrated and transported to the lake with a minimal time lag between fallout and delivery. We conclude from the striking contrast between glacial and paraglacial $^{10}$Be systematics at this site that glacial lake sediments very likely record $^{10}$Be fallout variations with a significantly higher amplitude and shorter time lag than

paraglacial sediments. Fallout variations in paraglacial lake sediments appear to be strongly suppressed by watershed buffering of $^{10}$Be fluxes.

Figure 22 compares the $^{10}$Be fallout flux estimate from Newbury to ice core $^{10}$Be flux records. Because the variability in $^{10}$Be fallout flux in the paraglacial section of the record is minimal and near the level of measurement uncertainty, this portion of the record has no effect on a correlation analysis and we do not consider it. However, the glacial portion of

the record prior to NAVC 7200 has significant centennial-scale variability. The important feature of the ice core records in this time interval is a group of prominent peaks in $^{10}$Be fallout, well in excess of the characteristic variability of the surrounding time period, between 13900-14500 GICC05 yr BP. If we accept the estimate of Adolphi et al. (2018) that the INTCAL13 and GICC05 time scales are equivalent in this interval, the best-fitting radiocarbon calibration of the NAVC Thompson et al. (2017) would imply a NAVC-GICC05 yr BP offset of 20820 yr. Event matching in the time range covered

by the Newbury glacial section by Ridge et al. (2012) suggested a NAVC-GICC05 yr BP offset of 20825 yr. However, these values fail to align peaks in the varve and ice core $^{10}$Be fallout records (Fig. 22) and, in fact, yield an inverse correlation between the two records. In contrast, a NAVC-GICC05 yr BP offset near 20950 years aligns significant peaks at NAVC 6950 and 6700 with corresponding peaks in the ice core record at 13975 and 14300 yr BP (Fig. 22). An offset of 20925 yields best alignment of the peaks at NAVC 6950/13975 GICC05 yr BP (the peak at NAVC 6950 dominates

a correlation metric; see Fig. 22), but a value of 20950 better aligns both peaks (as the Newbury record has 25-year spacing, the difference between 20920 and 20950 is not significant). Other matches are possible, because the Newbury record displays two major peaks and the ice core records display three major peaks and several adjacent minor ones in this time period. For example, an offset near 20675 would align the peak at NAVC 6700 with the peak at 13975 BP, but displays lower correlation overall.



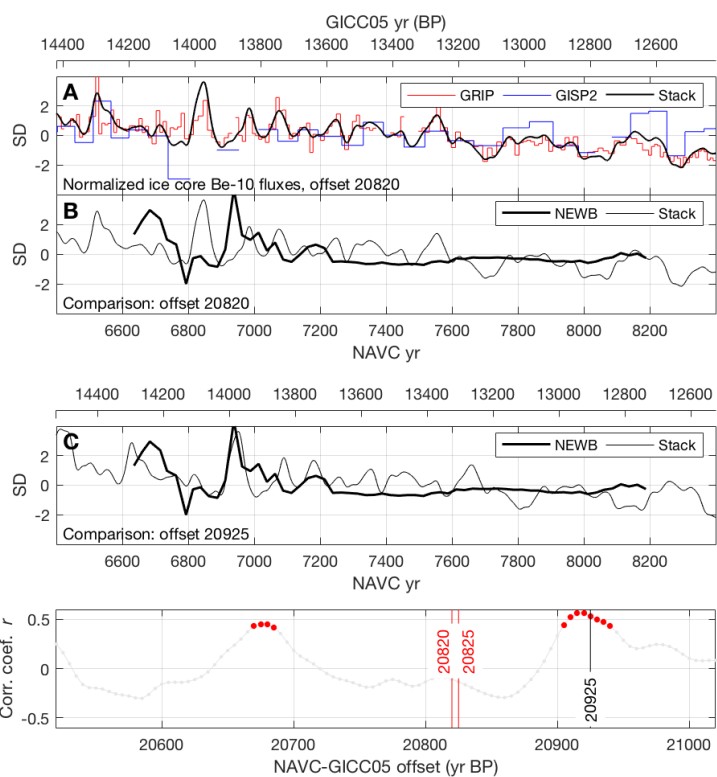

**Figure 22.** Comparison of the $^{10}$Be fallout flux estimate from Newbury with ice core records of $^{10}$Be fallout. Plot elements are the same as in Fig. 18. **A**, scaled and centered ice core $^{10}$Be flux records from Adolphi et al. (2018), offset to NAVC years with the value of the offset inferred from $^{14}$C data (Thompson et al., 2017) and the assumption that INTCAL13 and GICC05 timescales are equivalent in this interval. **B**, scaled and centered Newbury fallout flux estimate compared to the ice core stack for the same offset. **C**, same as B, only for an offset of 20925 yr that aligns peaks in varve and ice core $^{10}$Be (see text). **D**, correlation between varve and ice core $^{10}$Be records for different offset values. This considers only the glacial part of the section; the minimally variable paraglacial section does not affect the conclusions. Vertical lines highlight other suggested offsets inferred from radiocarbon calibration (20820) and event correlation (20825; see text).





Overall, the most prominent feature of the decadal-resolution [10]Be record at Newbury is the strong contrast between [10]Be systematics in glacial and paraglacial varves. This, as argued above, indicates that [10]Be fallout variability is likely to be enhanced in glacial varves by [10]Be runoff from the ice sheet, but suppressed in paraglacial varves by watershed buffering. The glacial part of the section displays significant centennial-scale [10]Be fallout variability, which has similar
magnitude to and can be matched with coeval ice core records.

### 4.2.3 Decadal series, summary results

Although the paraglacial section at Newbury does not show significant variability in [10]Be fallout flux, [10]Be fallout flux estimates from the glacial sections at Newbury, Kelsey-Ferguson (KF), and Glastonbury (GL) show variability that can be matched to global ice core [10]Be fallout records using NAVC-GICC05 yr BP offsets that are similar to independently
estimated values. Despite several unverified assumptions in our reconstruction of [10]Be fallout fluxes from Equation 1 and significant uncertainties in relating apparent magnitudes of [10]Be fallout variability to those expected from global records, the fact that NAVC-ice core matches with similar values of the offset can be made at both sites suggests that apparent [10]Be flux variations in NAVC sediments are, in fact, recording global fallout variations.

The NAVC-GICC05 yr BP offsets that produce the best matches at Newbury and KF/GL are different, in the range
20900-20950 for Newbury and 21000-21150 at KF/GL. The match with Newbury is likely more robust, because it is based on aligning anomalously large peaks in both records. Offsets between the NAVC and GICC05 timescales obtained for different times are not required to be the same, because the GICC05 time scale is not exactly linear in calendar years (Adolphi et al., 2018). If matches at Newbury and KF/GL were both valid, the difference would imply that GICC05 is overcounted by 150-200 years between 14,000-17,000 BP. However, other estimates of the GICC05 - calendar year
offset (Adolphi et al., 2018,  and references therein) show the opposite, that it is undercounted by ~175 years during this interval, so a GICC05-NAVC offset of 21000 for KF/GL should be equivalent to an offset of 21175 at Newbury, or, conversely, an offset of 20925 at Newbury would be equivalent to 20770 at KF/GL. Thus, nonlinearity in the GICC05 time scale does not resolve the difference in apparent matches. Another theoretical possibility that would allow both matches to be valid would be the presence of a significant time lag between [10]Be fallout variations onto the land surface
and deposition in NAVC sediments, with this time lag being 200-400 yr longer at KF/GL than at Newbury. However, a watershed residence time and associated fallout-deposition time lag of hundreds of years would not be consistent with the observation that 50-100 year peaks in [10]Be deposition at Newbury are recorded at similar resolution to those recorded in ice core records. Overall, these arguments tend to indicate that the match at 20925 at Newbury, the match at 21110 at KF/GL, or perhaps both, are inaccurate. In the next sections we discuss this in light of previous, independent approaches
to calibrating the NAVC time scale.




### 4.2.4 Comparison of NAVC-GICC05 yr BP offsets inferred from [10]Be flux matching with NAVC-cal yr BP calibration from radiocarbon dates

In previous work, Ridge et al. (2012) and Thompson et al. (2017) estimated the NAVC-calendar year BP offset by choosing the value of the offset that minimizes the weighted mean square of residuals between observed [14]C ages on plant macrofossils in individual varves and the INTCAL13 calibration curve, yielding an estimated NAVC-cal yr BP offset of 20820 yr. This procedure assumes that all [14]C ages are equally likely to be accurate, so it acts to choose a value for the offset such that the mean of the residuals is zero. However, Ridge et al. (2012) also observed that the residuals from this procedure were not normally distributed, and instead were characterized by a young-skewed distribution, so even though the mean of the residuals was zero, the mode was positive (see Fig. 10 in Ridge et al., 2012). They proposed that the skewed distribution was the result of small amounts of postdepositional contamination of samples by younger carbon. This would imply that the correct value of the NAVC-cal yr BP offset should be chosen so as to align the mode of the residuals, rather than their mean, with zero, which would increase the value of the offset by 100-200 years. As shown in Figure 23, if we accept that the GICC05 timescale is equal to true calendar years near 13,500 years BP Adolphi et al. (2018), then the NAVC-GICC05 yr BP offset of 20925 yr obtained from [10]Be flux matching at Newbury is consistent with this reasoning: [14]C residuals relative to INTCAL13 with this offset have a mode equal to zero and a young tail. In other words, with this offset, the INTCAL13 calibration curve is closely aligned with the oldest radiocarbon ages observed in clusters of data at several levels in the NAVC (Fig. 23; with the exception of four old outliers that are likely the result of contamination; see Ridge et al., 2012). Thus, the NAVC-GICC05 yr BP offset of 20925 yr inferred from [10]Be flux matching at Newbury is consistent with the [14]C data set, and in fact may better explain the [14]C data than the estimate of the offset previously obtained by aligning the mean of the [14]C residuals with zero.

On the other hand, applying the NAVC-GICC05 yr BP offset of 21110 obtained from [10]Be flux matching to the KF/GL records results in a poor alignment between INTCAL13 and the NAVC radiocarbon data set. Predicted [14]C ages with this offset are nearly all older than observed, resulting in both the mode and the mean of the residuals being less than zero (Fig. 23) and requiring that essentially all [14]C ages be spuriously young due to contamination. Adding the proposed ~175 yr undercount in the GICC05 timescale between ~13,000 and ~17,000 yr BP (Adolphi et al., 2018) worsens the agreement.

We conclude that (i) the NAVC-ice core [10]Be flux match at Newbury is consistent with the [14]C data set, but (ii) the [10]Be flux match to the KF/GL record results in a value of the offset that is too old to be consistent with the [14]C data.

### 4.2.5 Comparison of NAVC-GICC05 yr BP offsets inferred from [10]Be flux matching and climate event correlation

Ridge et al. (2012) and Thompson et al. (2017) proposed NAVC-GICC05 offsets based on correlation between abrupt climate events in Greenland recorded in ice core $\delta^{18}$O records and ice-marginal events recorded by the NAVC. The premise of this approach is that rapid climate changes likely affected central Greenland and the margin of the LIS in the north-



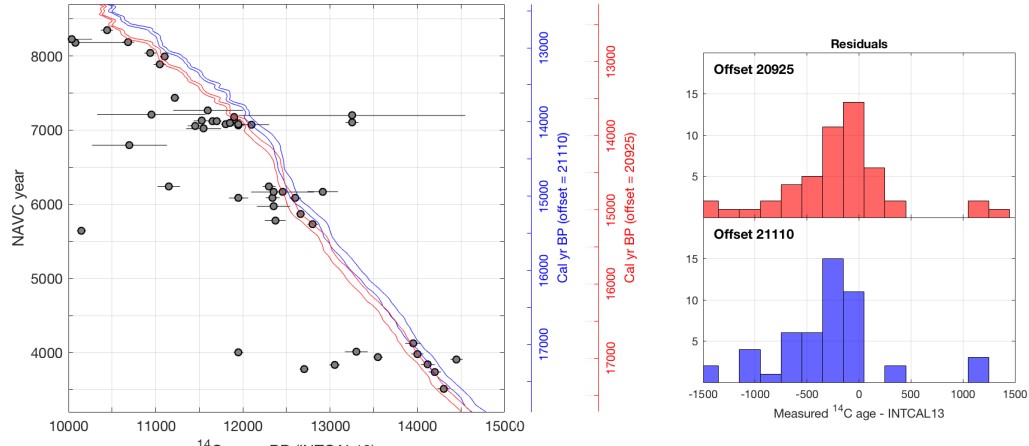

**Figure 23.** Alignment of radiocarbon ages of plant macrofossils in NAVC varves (Ridge et al., 2012; Thompson et al., 2017) with the INTCAL13 radiocarbon calibration curve, for values of the NAVC-GICC05 yr BP offset estimated from [10]Be flux matching (Figs. 18, 22) and the assumption that GICC05 years are equivalent to calendar years. The bands outlined in red and blue are the 68% uncertainty bounds for the INTCAL13 calibration assuming offsets of 20925 and 21110, respectively. The histograms in the right panel show residuals defined as the difference between the measured radiocarbon age of a sample from a particular varve year and the age predicted for that year by the INTCAL13 calibration and the selected offset.

eastern US similarly. Thus, cold periods in Greenland should be associated with reduced meltwater production, which is recorded by decreased varve thicknesses, and slower ice margin retreat rates, or potentially stillstands or readvances, at the LIS margin. Warm periods should be associated with increased meltwater production, increased varve thickness, and higher ice margin retreat rates.

5     *Events near 14,000 yr BP.* For the period near 14,000 cal yr BP, Thompson et al. (2017) showed that the NAVC-cal yr BP offset of 20825 yr derived from fitting [14]C data to INTCAL13 also yielded a close event correlation between Greenland ice core records and events in the NAVC (Figure 24A). This event correlation associates a decrease in varve thickness near NAVC 6800 and increase near NAVC 6920 (Fig. 24) at Newbury with a rapid cooling and warming in Greenland at the boundaries of stadial GI-1d at 14025 and 13905 GICC05 yr BP, respectively (Lowe et al., 2008; Rasmussen et al., 2014), 10  and aligns the beginning of the Littleton-Bethlehem Readvance, an ice margin readvance stratigraphically bracketed by NAVC varves 6836-6987, fairly closely with the beginning of GI-1d.

    The NAVC-GICC05 offset of 20925 yr inferred from [10]Be flux matching at Newbury (Fig. 22) implies that the Littleton-Bethlehem Readvance began in a precursory cold event prior to the start of GI-1d and ended at the GI-1d termination, which is stratigraphically permissible (Fig. 24B). On the other hand, this match fails to align appropriate varve thickness 15  changes with the boundaries of GI-1d, and is therefore inconsistent with the hypothesis that cold events in Greenland



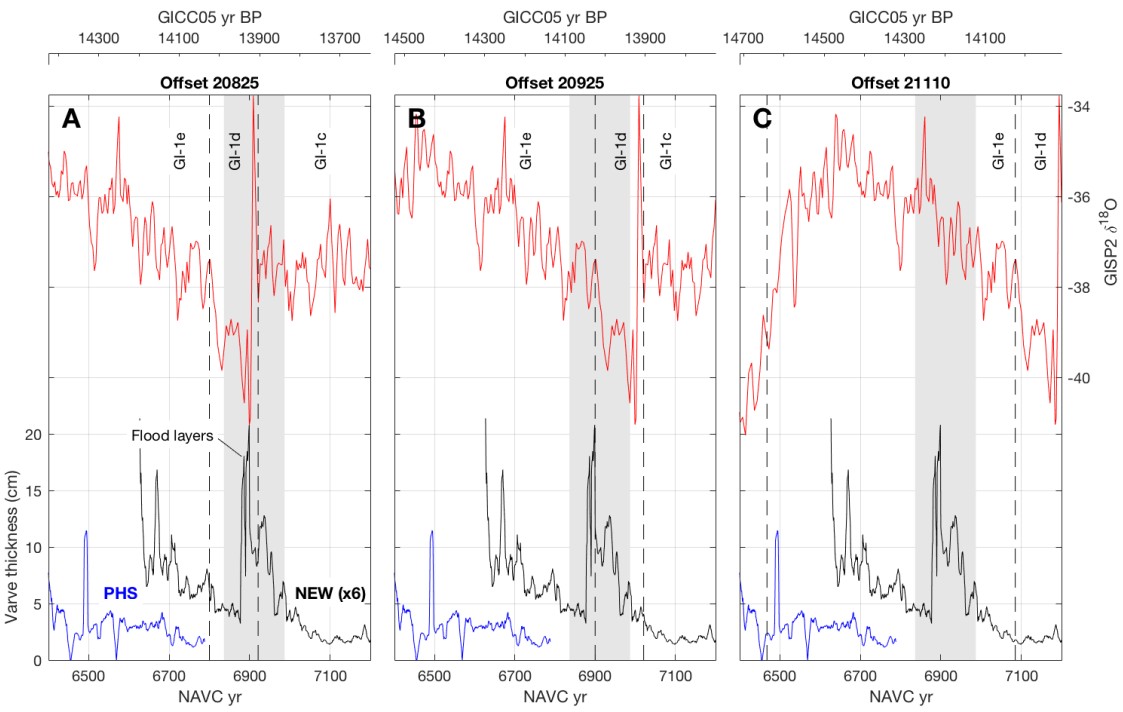

**Figure 24.** Event correlation between NAVC and GISP2 near 14,000 GICC05 yr BP. This duplicates Fig. 14 of Thompson et al. (2017), except that the x-axis is reversed for consistency with the rest of the figures in this paper. Records at bottom are 9-year running averages of varve thickness records from Newbury (NEWB) and Perry Hill Basin (PHS; see Ridge et al., 2012). The GISP2 $\delta^{18}$O record (Stuiver et al., 1995; Stuiver and Grootes, 2000) is a 3-sample running average. The gray band shows the minimum duration of the Littleton-Bethlehem Readvance inferred from varves directly underlying and overlying the readvance till (see Thompson et al., 2017). Panel **A** uses the offset proposed as the best event correlation by Ridge et al. (2012) and Thompson et al. (2017), which is the same as the offset derived from fitting to INTCAL13 by Thompson et al. (2017). **B** uses the offset that provides the best match between the $^{10}$Be flux at Newbury and the ice core stack (Fig. 22). **C** uses the offset that provides the best match between the $^{10}$Be flux at KF/GL and the ice core stack (Fig. 18). Note that the anomalously thick varves near NAVC 6900 (highlighted in A) are related to a lake outburst flood and therefore have no climate significance (see Ridge et al., 2012). The Greenland climate event boundaries are from Rasmussen et al. (2014)




should be associated with decreased meltwater production at the LIS margin. Therefore, accepting a NAVC-GICC05 yr BP offset of 20925 yr would require abandoning the assumption that the climates at the NAVC and central Greenland were similar, or revising our understanding of the significance of the varve thickness changes at Newbury.

The NAVC-GICC05 yr BP offset of 21110 inferred from [10]Be flux matching at KF/GL (Fig. 18) fails to match ice core
5    and NAVC events in this time range, instead placing the Littleton-Bethlehem readvance in a transitional phase of GI-1e (Fig. 24C) and yielding no obvious match between ice core climate and varve thickness records.

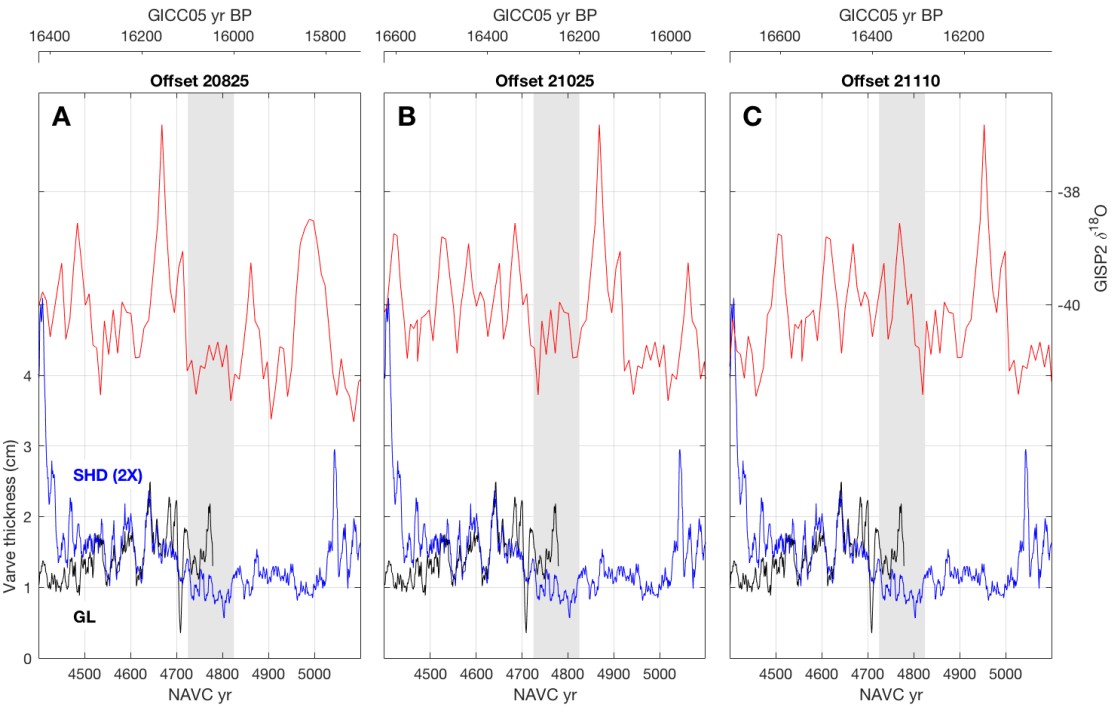

**Figure 25.** Event correlation between NAVC and GISP2 near 16,000 GICC05 yr BP. This duplicates Fig. 14A of Ridge et al. (2012), except that the x-axis is reversed for consistency with the rest of the figures in this paper. The plot elements are the same as in Figure 24. The gray band shows the Hatfield event, a period of slowed ice recession or a brief stillstand associated with thinner varves between NAVC ∼4725-4825.

*Events near 16,000 yr BP.* Ridge et al. (2012, ; their Figure 14A) proposed that a NAVC-GICC05 yr BP offset of 20825 yr also aligned ice core climate and NAVC events at the time of the Hatfield event, an ice margin stillstand between NAVC 4725-4825. This event is associated with an abruptly bounded period of thinner varves at South Hadley (Fig. 25),
10   and a 20820 offset aligns this period with an unnamed minor cold phase in the ice core record (Fig. 25A). Although the NAVC-GICC05 yr BP offset of 21110 instead aligns this event with a warm phase (Fig. 25B), a value of 21025, which is within the fairly broad range of acceptable matches at KF/GL, would align it with a different cold phase (Fig. 25C). It is




also true, however, that the offset near 20925 yr inferred from $^{10}$Be flux matching at Newbury, with the proposed ~150 yr undercount in GICC05 in the 13.5-16 ka range (Adolphi et al., 2018) would result in an offset near the originally proposed value of 20820. Thus, event correlation around the Hatfield event is inconclusive as to whether the $^{10}$Be flux matches at either Newbury or KF/GL are valid.

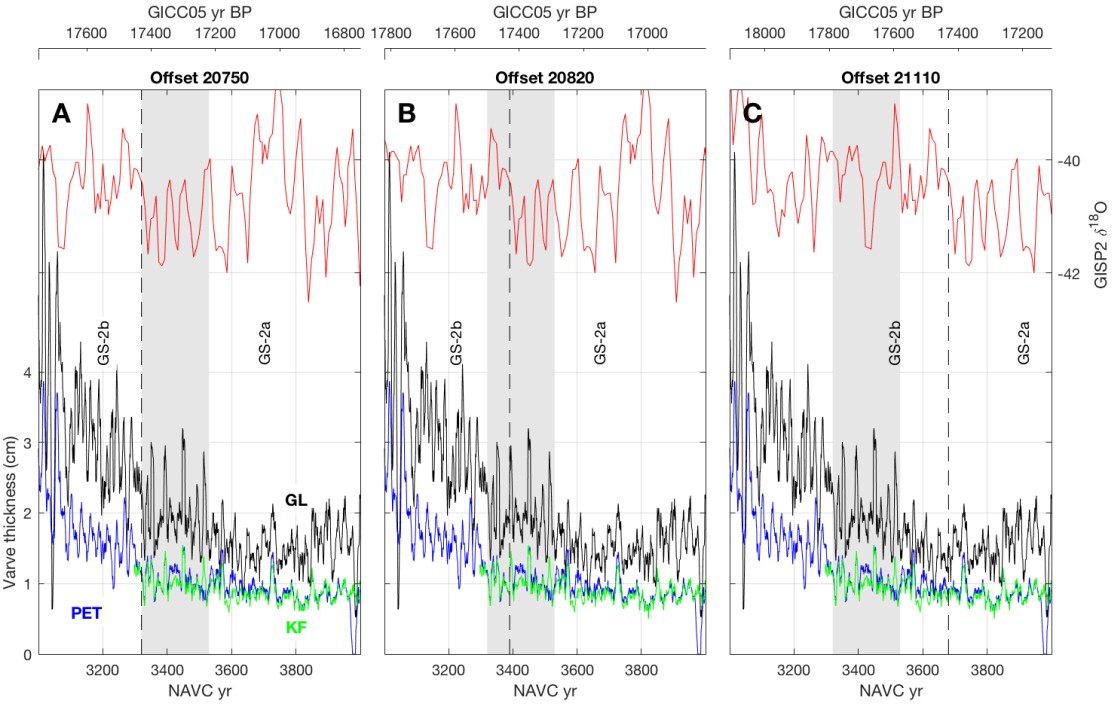

**Figure 26.** Event correlation between NAVC and GISP2 near 17,500 GICC05 yr BP. This duplicates Fig. 14B of Ridge et al. (2012), except that the x-axis is reversed for consistency with the rest of the figures in this paper. The plot elements are the same as in Figure 24. Varve thickness records shown are from Glastonbury (GL), Kelsey-Ferguson (KF), and and South Windsor, CT Ridge et al. (PET; see 2012). The gray band shows the period of the Chicopee Readvance, which is associated with a period of thin varves at NAVC 3320-3530.

5      *Events near 17,000 yr BP.* Ridge et al. (2012, their Figure 14A) proposed a NAVC-GICC05 yr BP offset of 20750 yr to align ice core climate and NAVC events at the time of the Chicopee Readvance, which is associated with a period of thin varves near NAVC 3320-3530. They noted that this value of the offset aligns century-scale peaks in varve thickness with corresponding ice core $\delta^{18}$O variations during an unnamed cold phase (Fig. 26A). The value of 20825 proposed as the best offset for event correlation around the Littleton-Bethlehem and Hatfield events fails to align these peaks and

10    does not correlate the readvance with the cold phase (Fig. 26B). The NAVC-GICC05 yr BP offset of 21110 yr inferred from $^{10}$Be flux matching in this time range at KF/GL fails to align the Chicopee Readvance with any cold phase (Fig.





26C). However, if we accept the NAVC-GICC05 yr BP match at 20925 from $^{10}$Be flux matching at Newbury and apply the estimate of Adolphi et al. (2018) that GICC05 is undercounted by ~175 yr between ~13,500 and ~17,000 BP, then a NAVC-GICC05 yr BP offset of 20925 at Newbury implies a NAVC-GICC05 yr BP offset near 20750 at the time of the Chicopee Readvance, which is the same as that proposed for event correlation by Ridge et al. (2012). Thus, the $^{10}$Be

flux match at Newbury is consistent with event correlation around the Chicopee Readvance, but the $^{10}$Be flux match at KF/GL is not.

## 5   Conclusions

### 5.1   Systematics of Be deposition in varved glacial lake sediments

We found that the majority of $^{10}$Be in glacial and paraglacial varves in the NAVC is inherited and is already adsorbed

to sediment when supplied to the lake. This means that the majority of variation in total $^{10}$Be flux to the lake reflects variations in the supply of inherited $^{10}$Be, which are primarily controlled by variations in sediment grain size, and not variations in fallout $^{10}$Be. By using $^9$Be as a normalizing isotope, we were able to identify variations in $^{10}$Be flux that are not explained by variations in inherited Be supply to the lake and which we therefore attribute to variations in atmospheric $^{10}$Be fallout. This is an important element of this study: if we had not measured $^9$Be concentrations, it might be difficult

or impossible to separate variations in inherited and fallout $^{10}$Be fluxes. Having quantified the relationship between grain size and Be concentrations in this study, however, it might be possible in future studies of the NAVC to extract variability in $^{10}$Be fallout without $^9$Be measurements by instead identifying variations in the total $^{10}$Be flux not explained by grain size, which would be similar to the approach of multivariate correlation with various sedimentological proxies applied in other studies (e.g.,  Czymzik et al., 2016). Regardless, using $^9$Be as a normalizing isotope appears simpler.

### 5.2   Variations in $^{10}$Be fallout flux

Our initial strategy in this study was to test the hypothesis that $^{10}$Be fallout variations are recorded in NAVC sediments by trying to identify the characteristic 11-year solar variability period in $^{10}$Be fallout variations. This strategy was not successful, most likely because of the predominance of inherited $^{10}$Be in relation to fallout $^{10}$Be in NAVC sediments. Expected fallout variability of 15-20% around the mean associated with the 11-year solar cycle translates into variability

in total $^{10}$Be flux that has similar magnitude to measurement uncertainty. Thus, we were not able to distinguish short-period solar forcing of $^{10}$Be fallout variations, if present, from measurement noise.

On the other hand, in 1000-year varve sequences sampled with biennial resolution at two sites, we did observe centennial-scale variability in $^{10}$Be fallout flux having similar magnitude and period to $^{10}$Be fallout variations observed in ice cores. In addition, these variations were reproducible in overlapping varve sequences from two different sites.

We conclude that $^{10}$Be concentrations in NAVC sediments do, in fact, record at least some prominent, high-amplitude, variations in $^{10}$Be fallout that can potentially be used to correlate NAVC and ice core records.



### 5.3 $^{10}$Be systematics in glacial and paraglacial varve sequences

An unexpected result of this study is that glacial varves are likely better suited to reconstructing $^{10}$Be fallout than paraglacial varves. Initially, we supposed that paraglacial varves would be more likely to yield a record of $^{10}$Be fallout simply because they are typically an order of magnitude thinner than glacial varves, so $^{10}$Be fallout would be less diluted

by higher sedimentation rates and associated larger inherited Be fluxes. This supposition was incorrect, and we found that variability in $^{10}$Be fallout was evident in glacial varve sequences, but nearly entirely suppressed in paraglacial varves.

This result is most likely explained by the different roles of the watersheds that contribute to glacial and paraglacial lakes. In a paraglacial environment where the lake is not receiving direct glacial meltwater, the lake watershed consists predominantly of a deglaciated landscape mantled by fresh glacial sediment. A significant fraction of $^{10}$Be fallout to the

watershed is therefore adsorbed to sediment in developing soils, which limits delivery of fallout $^{10}$Be to the lake and buffers variations in fallout rates.

In contrast, the contributing watershed of a proglacial lake in direct contact with the ice margin most likely includes large areas of the ablation zone of the ice sheet. Fallout $^{10}$Be cannot be stored on the ice in the ablation zone, so each year's accumulation of fallout must be flushed into the lake by supraglacial or englacial discharge during the summer melt

season. We hypothesize that this phenomenon can result in substantial focusing of fallout $^{10}$Be in glacial lake sediment as well as a minimal lag time between fallout variations and recording of those variations in glaciolacustrine sediment records. If true, this implies that varved glacial lake sediments, if deposited in a lake with a large supraglacial watershed, may be very effective recorders of atmospheric $^{10}$Be fallout. This hypothesis could easily be tested in modern proglacial lakes with large contributing areas on the ice surface, perhaps in west Greenland.

### 20 5.4 Correlation of the NAVC to ice core records by $^{10}$Be flux matching

We obtained possible matches between reconstructed variations in $^{10}$Be fallout flux and an ice core $^{10}$Be flux stack for two varve sequences: a sequence at Newbury deposited near 14,000 yr BP and overlapping sequences at Kelsey-Ferguson (KF) and Glastonbury (GL) deposited ca. 16,000 - 17,000 yr BP. The NAVC-GICC05 yr BP offsets derived from these matches are not the same for both sections, and also differ by 50-200 years from previously proposed values of the

NAVC-calendar year offset inferred from radiocarbon dating of NAVC sediments and event matching between NAVC varve thickness records and Greenland ice core climate records.

Matching ice core $^{10}$Be records with $^{10}$Be fallout flux variations inferred from the older KF and GL varve sequences implies a NAVC-GICC05 yr BP offset in the range 20965-21165 yr, with the best correlation at 21110 yr. However, the match between the two records mainly relies on aligning millenial-scale trends in the two records, rather than distinct

peaks associated with centennial-scale variations. As the KF/GL record is only 1000 yr long, this could lead to spurious results. In addition, centennial-scale variations in $^{10}$Be fallout flux in ice core records during this time period are relatively subtle. This match is not consistent with the NAVC-cal yr BP offset inferred from fitting NAVC $^{14}$C data to the INTCAL13 calibration curve, even when likely differences between the GICC05 ice core timescale and the true calendar





year timescale are taken into account. Although offsets in the range implied by this match yield an acceptable correlation between ice core climate records and NAVC varve thickness records for events near NAVC 4800 / 16,000 GICC05 yr BP (Fig. 25), they do not yield good event correlations for younger or older periods (Figs. 24, 26). Overall, the apparent match between KF/GL [10]Be fallout flux and ice core [10]Be does not appear consistent with most independent evidence.

Matching the [10]Be fallout record from the glacial section at Newbury with ice core [10]Be records, in contrast, relies on correlation of prominent large-magnitude, centennial-scale variations in both records and implies a NAVC-GICC05 offset in the range 20900-20940 (Fig. 22), with the best correlation at 20925 yr. This value of the offset is consistent with the [14]C data set and, if the likely undercounting of ice core years in the GICC05 timescale between ∼13,000-17000 yr BP is considered, also consistent with event correlations between the NAVC and Greenland ice core climate records for periods

surrounding the Hatfield event near NAVC 4800/16,000 GICC05 yr BP (Fig. 25) and the Chicopee readvance near NAVC 3400/17400 GICC05 yr BP (Fig. 26). This value of the offset yields an acceptable correlation between the stratigraphic boundaries of the Littleton-Bethlehem readvance and the GI-1d stadial (Fig. 24), but fails to align the associated period of decreased varve thicknesses, which is interpreted to reflect decreased meltwater production during a cold period, with the corresponding stadial boundaries. Thus, although the NAVC-ice core [10]Be fallout match at Newbury is consistent

with the majority of independent evidence, it is not consistent with one important constraint provided by the expected climatological relationship between climate in central Greenland and at the southern margin of the LIS. Overall, the 20925 yr NAVC-GICC05 yr BP offset inferred from [10]Be fallout matching at Newbury may be valid, but either an unrecognized time lag of several decades, or a reassessment of the climate significance of varve thickness variations associated with the Littleton-Bethlehem readvance, would be required to resolve all conflicts with independent data.

**5.5    Weaknesses and strategy for improvement**

The weakest part of our argument that the correlation between [10]Be fallout flux in NAVC sediments at Newbury and the ice core stack is valid is that the correlation is based on a short record, comprising only 600 glacial varves that display two prominent [10]Be fallout peaks. The ice core stack, on the other hand, includes three major fallout peaks between 14,000-15,000 GICC05 yr BP, as well as a number of relatively high-amplitude secondary features. In addition, we only

observed these [10]Be fallout peaks at one section, so cannot prove by replication between sections that they represent true fallout variations. Finally, the 25-year sampling resolution at Newbury is long relative to the size of the expected fallout peaks. For example, one of the most prominent peaks is only represented by a single sample, and this effect alone could potentially explain much of the apparent ∼100-year difference between the correlation based on [10]Be flux and that based on event matching. These weaknesses could be addressed by (i) higher-resolution sampling at the Newbury section to

better resolve apparent fallout peaks, as well as (ii) replication of the [10]Be fallout record from additional varve sequences that overlap with Newbury in the range NAVC 6000-7000. Several overlapping sections exist, including the Perry Hill Basin site shown in Fig. 24, that could be used for this purpose. Higher-resolution [10]Be measurements from older parts of the NAVC might be helpful in resolving whether or not the NAVC-ice core [10]Be match at KF/GL is or is not valid, but are less likely to be successful because [10]Be fallout variability in the ice core stack between 15,000 - 18000 GICC05 yr



BP has lower amplitude than that in the 13,000 - 15,000 GICC05 yr BP range. Overall, collecting higher-resolution [10]Be data on replicate sections of glacial varves from this younger time period, between approximately NAVC 5800 and the glacial-paraglacial transition near NAVC 7000, is the best strategy for resolving the remaining ambiguous results of this study.

5 *Data availability.* All data are in the supplementary files.

*Author contributions.* All authors contributed equally. Ridge had primary responsibility for correlation and sampling of varve sections, DeJong and Bierman had primary responsibility for sample preparation, Rood had primary responsibility for AMS measurements, and Balco had primary responsibility for data analysis and paper production.

*Competing interests.* Balco is an editor of *Geochronology.*

10 *Acknowledgements.* This work was supported by the U.S. National Science Foundation under grants EAR-1103381, EAR-1103532, EAR-1103399, EAR-1103037, and EAR-0639830. Balco was also supported in part by the Ann and Gordon Getty Foundation. We thank Kathyrn Lowe and Sarah Strand for help in the field and with grain size analysis, Sophie Greene for carrying out the [9]Be measurements, Florian Adolphi for supplying ice core and stacked [10]Be fluxes in an easily accessible form, and the staff of the AMS Laboratory at the Scottish Universities Environmental Research Centre (SUERC) for support for Be isotope ratio analyses.



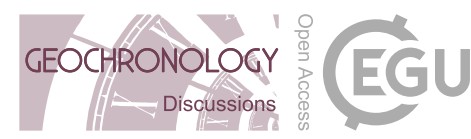

**Table 1.** Characteristics and purpose of varves and sets of varves sampled for Be measurement.

| Site identifier | Site name and location | Range of NAVC years sampled | Calendar age BP (offset: 20820 yr) | Varve type | Purpose | Sample format |
|---|---|---|---|---|---|---|
| GL | Glastonbury, Connecticut | 3805-4675 | 17015-16145 | Glacial, ice-distal | Long record, decadal resolution | 10 varves/sample, with 20-30 yr spacing |
| KF | Kelsey-Ferguson Plant of Redlands Brick Co., South Windsor, Connecticut | 3562 | 17258 | Glacial, ice-distal | Summer-winter analysis | Single season of single year |
| | | 3564 | 17256 | Glacial, ice-distal | Summer-winter analysis | Single season of single year |
| | | 3565-3644 | 17255-17176 | Glacial, ice-distal | Short record, biennial resolution | 2 varves/sample, contiguous samples |
| | | 3658-4175 | 17162-16645 | Glacial, ice-distal | Long record, decadal resolution | 10 varves/sample, with 25-yr spacing |
| NHT | North Hatfield, Massachusetts | Gray diamict beneath 4667 | >16153 | Ice-proximal diamicton | Glacial source material | |
| | | 4667-4669 | 16153-16155 | Glacial, ice-proximal | Glacial source material | 3 varves combined |
| | | 4673-4675 | 16147-16145 | Glacial, ice-proximal | Glacial source material | 2 varves combined |
| | | 5619 | 15201 | Glacial, ice-distal | Summer-winter analysis | Single season of single year |
| | | 5620 | 15200 | Glacial, ice-distal | Summer-winter analysis | Single season of single year |
| SHD | South Hadley, Massachusetts | Red diamict beneath 4019 | >16801 | Subglacial till | Glacial source material | |
| | | 4019-4021 | 16801-16803 | Glacial, ice-proximal | Glacial source material | 3 varves combined |
| | | 4026-4027 | 16792-16793 | Glacial, ice-proximal | Glacial source material | 2 varves combined |
| NEWB | Newbury, Vermont | 6631-8193 | 14189-12627 | Glacial transitioning to paraglacial | Long record, decadal resolution | 10-20 varves/sample, with 25-yr spacing |
| NHV | North Haverhill, New Hampshire | 7025 | 13795 | Paraglacial with minimal glacial input | Summer-winter analysis | Single season of single year |
| | | 7026 | 13794 | Paraglacial with minimal glacial input | Summer-winter analysis | Single season of single year |
| | | 7026-7109 | 13794-13711 | Paraglacial with minimal glacial input | Short record, biennial resolution | 2 varves/sample, contiguous samples |
| | | 7110 | 13710 | Paraglacial with minimal glacial input | Summer-winter analysis | Single season of single year |
| | | 7111 | 13709 | Paraglacial with minimal glacial input | Summer-winter analysis | Single season of single year |
| 930PN[1] | Eatonville, Mohawk Valley, New York (not correlated to NAVC; locally numbered) | (10) | ~17,000 | Glacial, ice-proximal | Summer-winter analysis | Single season of single year |
| | | (11) | ~17,000 | Glacial, ice-proximal | Summer-winter analysis | Single season of single year |

[1] Varves at 930PN are unusual in that they are derived from a limestone source and predominantly carbonate.





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
