# Peer review of "Atmospherically-produced beryllium-10 in annually laminated late-glacial sediments of the North American Varve Chronology"

_Geochronology, 2020_

## Author Comment (AC1) · 17 Jun 2020

This is to call readers' attention to a minor typographical error that may make it unnecessarily difficult to understand part of the methods.

On page 7, line 21, "$^{10}$Be" incorrectly appears in place of "$^9$Be".

This sentence should read:

"Our basic approach is to (i) use $^9$Be concentrations in sediment as a measure of

both the delivery of background Be to the lake and any environmental factors affecting Be scavenging efficiency in the lake, and then (ii) identify additional variation in the $^{10}$Be flux that cannot be accounted for by these effects and must therefore represent variability in the atmospheric fallout flux."

We apologize for the error.

---

## Referee Comment (RC1) · Anonymous Referee #1 · 15 Jul 2020

Balco et al. present new 10Be/9Be data from several varved sediment records which form part of the North American Varve Chronology (NAVC). Their aim is to test the applicability of these cosmogenic radionuclide records to synchronize the NAVC to Greenland ice core 10Be records. This would allow for detailed studies comparing the timing of climate changes recorded in both archives. To do this, they investigate a number of records addressing different questions: i) Do the sediments record the 11-year solar cycle? ii) Do the sediments record longer term solar variability? iii) How does the

measured variability compare to ice core records? Balco et al introduce a linear model which separates the measured 10Be into two components: one representing the 10Be from recent fallout (which is the crucial figure when wanting to reconstruct production rates in order to achieve synchronization to ice cores), and a second component derived from various sources ("old" 10Be present in the ice or adsorbed to the sediment) which they refer to as "inherited". By measuring 9Be and assuming that inherited Be has a constant 10Be/9Be, Balco et al. correct for the effect of variations in the delivery of inherited 10Be, isolating 10Be derived from fallout. Applying this model to the different records they find that i) The sediments do not record the 11-year solar cycle, ii) they may record longer term production rate variability, iii) the longer term 10Be variability shares features with the ice core records and may hence be synchronized. Subsequently, they derive possible timescale offsets between ice cores and sediment by lagged correlation analysis and discuss the results in context with previous timescales based on 14C and climate-matching.

Generally, I think this study is a very careful piece of work which provides great insight into 10Be records in complicated settings. Nevertheless, I have major reservations about the interpretation of the 10Be records the authors provide. Mainly, I do not think that the 10Be records reliably reflect production rate changes, but rather that they are dominated by variations in the 10Be/9Be of what the authors call "inherited" 10Be. I want to stress, that this does not affect the discussion and quality of this work, which is excellent. In fact, the authors point to the uncertainties of their records throughout the manuscript. Nevertheless, I think the sections dealing with the synchronization should possibly be left out of the manuscript as I consider them unreliable, but they might be used by future studies. Below, I will outline the reasons for my skepticism in more detail and propose additional tests.

Whenever attempting to use cosmogenic radionuclide records for synchronization, the key-assumption underlying all this, is that all records reflect production rates. This needs to be demonstrated before any further synchronization is attempted. Otherwise,

one may synchronize spurious signals. This may not be necessary when all records are very similar, but looking at figure 18 and 22 I'd argue that they actually do not share a lot of similarities, but that it is mainly single peaks that are being aligned.

The authors attempt to verify their 10Be records by looking at frequency spectra and cycles – none of which are giving a clear indication that we are indeed looking at production rate changes. One major argument that is not explored is the amplitude of changes. From production models we know which amplitudes we expect from production rate changes. For solar activity changes this is typically in a range of around 50% (R Muscheler & Heikkilä, 2011), while geomagnetic field changes may introduce larger changes (however, there are no large geomagnetic field changes within the study period). Through their regression model, the authors demonstrate that in their setting, meteoric fallout (∼production rate changes) only contributes to about 10% of the total 10Be, while 90% are inherited. Given this big reservoir, production rate changes will be strongly dampened. A production rate change of 50% would appear as a 5% change in the record, which is essentially equivalent to the estimated uncertainty of the data. More importantly, a variation of only 5% of the 10Be/9Be ratio of the inherited 10Be would have the same effect in the record: Can this really be excluded?

The authors argue that the reproducibility of 10Be variation at different sites demonstrates the influence of fallout 10Be on the sediment records. In my opinion, the records shown in figures 16/17 do not share a lot of variability and their disagreement should be used as an uncertainty estimate. The slight covariability between both sites is also seen in the mass accumulation rates, and hence, both records may simply share the same bias, introducing similarity.

In my opinion, the authors should discuss the amplitude of the changes in their data explicitly. In this light, also the comparison to ice core data done in zscores should be avoided, and instead relative amplitudes (relative to the mean) should be compared because they give additional information. From the regression model, the size of the reservoir (inherited 10Be) can be estimated, which can be used to rectify the amplitude dampening of the "fallout" component – is this amplitude physically reasonable, and significant beyond measurement uncertainty? Are the aligned wiggles of ice core and sediment 10Be of similar amplitude? Looking at figure 20 for example, and assuming a large contribution of inherited 10Be implies, that the glacial 10Be variations are in the order of a few hundred percent, happening within decades. From production rate models we can exclude that this type of variability can be caused by solar modulation. Large cosmic ray events on the other hand would be more short-lived, while geomagnetic field changes occur on longer timescales. And either way, the variations in the ice cores which this wiggle is linked to in figure 22 is significantly smaller, i.e., not the same feature.

Based on these revisions, the authors may or may not want to keep their section about the GICC05/NAVC comparison. I wonder, how much there is to be learned about the NAVC when 10Be records from ice and sediments actually don't really look alike. One will always find single peaks to align. The disagreement between ice core and sediment 10Be records even after alignment could at least be used to derive a posterior uncertainty estimate for the 10Be records (by e.g., RMSE), which could be factored into the uncertainty of the timescale-match. I imagine it will be so big, that all estimates (14C, climate, 10Be) are essentially indistinguishable within error.

As said in the beginning: I do not think that these points really affect the quality of the paper, which provides a great discussion of the factors influencing 10Be in lacustrine sediments. But I think, these points should rather be expanded by an amplitude-discussion, and instead the synchronization aspect should be shortened as it is (at least based on the present analysis) likely unreliable.

Minor Comments: Introduction: Please add a section to the introduction that highlights the main assumption behind the synchronization of radionuclide records (i.e., that the production rate changes dominate the signal) and why this is not a given for any archive (transport, deposition, reservoir, residence time, variable dilution,...) and thus a major complication for the aim of this (or any) study.

NAVC varve count error: In the entire discussion there is no mention of the varve counting (relative) error of the NAVC. Is this really zero? It should at least be noted in the appropriate sections that differences in the GICC05/NAVC offset may well in part be due to errors in the NAVC, not just GICC05.

9Be/10Be extraction: Typically, 9Be and 10Be are measured on the same leaching fraction of sediment. Here, the authors employ different extraction techniques for both isotopes. For this to work, the extraction efficiency of each method must be constant. Is this a problem? Maybe the authors could comment on whether this has the potential to introduce variations in 9/10Be.

P 1, L2: replace "calendar year timescale" with "ice core timescale"

P 1, L11-14: long, convoluted sentence. Consider dividing.

P 1, L20-24: difficult sentence. Similar but different?

P1, L20: "not consistent with independent evidence" – the previous sentence states the opposite?

P2 L15-16: Please indicate what this uncertainty refers to, i.e., how many sigma.

P2, L28-29: "generally globally synchronous". See my comment on an additional section for the introduction. This is only true under ideal conditions and when averaging sufficiently over meteorological variability. In many settings, meteorological influences will be dominant over the production rate changes.

P2, L30: Heikkilae et al. 2013 is not a good reference here, as they do not reconstruct solar variability. Please find more appropriate references such as: (Bard et al., 1997; Raimund Muscheler et al., 2007; Steinhilber et al., 2012).

P2, L32: Again, Heikkilae at al and Adolphi et al. do not contain new measurements. Please give credit to the authors that produced these long records: (Adolphi et al., 2014; Baumgartner et al., 1997; Finkel & Nishiizumi, 1997; Raimund Muscheler et al.,

2004; Vonmoos et al., 2006; Yiou et al., 1997)

P3, L22-24: If there is a transient transition over 100-200 years in the lake system: How can a synchronization of local climate records to Greenland be attempted? The result would depend on which varve record was chosen? Could you please comment? What does this mean for the achievable uncertainty of climate synchronizations?

P6, L1: See earlier: Could you please add a sentence on the counting uncertainty of NAVC? How valid is the approach of using a single value for the offset?

P6, L16-26: Recently a new radiocarbon calibration was released (IntCal20) which contains significant changes from IntCal13. I understand that it is beyond the scope of this study to redo all the calibrations, but maybe it is worthwhile mentioning that this adds extra uncertainty to the (by looking at the data, in my opinion very optimistic) 200 years.

P6, L29-33: Please do not use the term "climate events" here, when the event stratigraphy by the ice core community does not contain any events during this time. Generally, the climate wiggle matching in this section should be cross-checked by not only using 1 ice core, since these minor wiggles in d18O may be noise. They should be verified using GRIP, NGRIP and NEEM d18O.

P9, L15-16: Please add a citation to the figure for the fallout flux.

P9, L22: remove the "-" in the 0.6m.

P12, L1-3: This is only true if these inaccuracies are systematic. Are they?

P12, L9: Looking at figure 3 the +-0.05g/cm3 seem very optimistic. How would the regression in figure 3 change if the two summer varve measurements were excluded from the analysis?

P12, L20: replace "between" with "around"?

P14, L9: Is there a +1 missing from the right hand side of the equation? Otherwise, if

C/Cs equals 0.9, the right hand side becomes 0?

P16, L1-2: Earlier you write that S may be variable? Does this correction come with an uncertainty that is propagated throughout the study?

P18, Figure 9: Around 6700 NAVC years, there are large changes in MAR and 9Be, but not in 10Be. While it could of course be, that simultaneous and large production rate changes (however, of likely unphysical amplitude, see major comments) "counteract" this the changes in the delivery of inherited 10Be, this seems unlikely. Rather, this may highlight variable 10/9Be ratios in the inherited Be. Please add a few sentences discussing this feature.

P22, L8-9: "constant or normally distributed": But neither of these assumptions is true. It is obviously not constant (otherwise this study wouldn't be possible) and it is also not symmetric because i) if solar variability was normally distributed, the non-linear production rate relationship would still cause a skewed distribution of production rates, and ii) the transport and washout of aerosols from the atmosphere causes a logarithmic distribution of aerosol deposition even if the production rates were normally distributed. It is still ok, to use the model as is, but it should be highlighted, that these assumptions are not true, but sufficiently correct to not affect the validity of the results.

Figure 13: it would be interesting to see this figure in relative units, i.e., both 9Be and 10Be relative to their respective mean. If the assumption of a constant 10Be/9Be of the inherited Be was true, all data-points should scatter around the 1:1 line?

P26, L18: "factor of 2" it should be noted that also ice cores are affected by transport and deposition effects of 10Be, especially during periods of such variable climate. As mentioned earlier, there are physical constraints of what amplitude of changes we can expect.

Figure 18: "95% confidence" How is this determined? Do you take autocorrelation of the records into account?

P31, L3-4: I'd argue that the agreement is not that good? The amplitudes are different, and due to the similarity of both MAR records, the similarities in 10Be fluxes may well be cause by the flux calculation. Is it worthwhile discussing this option?

P31, L4-5: The general problem with matching these records is, that one will always find a match due to the periodicity of the signals. Hence, the amplitude discussion is important.

Figure 19: The doubling of 10Be/9Be between 6800 and 7200 NAVC cannot be production (unphysical amplitude). Please discuss.

Figure 21: See above

P34, L24-35: I'd argue that the sediment and ice core 10Be records simply don't look alike. Aligning a single peak can easily lead to erroneous results. Please find a measure for the similarity of the records that can be used to quantify the uncertainty in the match as well.

Figure 22: Please do not display the data as zscores. A lot of crucial information is lost that way, and it causes deviations between the ice core datasets that are merely due to resolution affecting the standard deviation of each record.

Figure 24: Please specify on which timescale the GISP2 d18O record is shown. GICC05 I suppose?

Figure 25: Consider plotting NGRIP, GRIP and NEEM d18O as well to get an idea about the robustness of those features.

P40, L7: According to the ice core event stratigraphy (Rasmussen et al., 2014) there are no "events" in the ice cores around this period.

Apologies for the lengthy review – I hope some of these comments are helpful to improve the manuscript and the efforts of dating the NAVC.

References:

Adolphi, F., Muscheler, R., Svensson, A., Aldahan, A., Possnert, G., Beer, J., Sjolte, J., Björck, S., Matthes, K., & Thiéblemont, R. (2014). Persistent link between solar activity and Greenland climate during the Last Glacial Maximum. Nature Geoscience, 7(9), 662–666. https://doi.org/10.1038/ngeo2225

Bard, E., Raisbeck, G. M., Yiou, F., & Jouzel, J. (1997). Solar modulation of cosmogenic nuclide production over the last millennium: comparison between 14C and 10Be records. Earth and Planetary Science Letters, 150(3–4), 453–462. https://doi.org/10.1016/s0012-821x(97)00082-4

Baumgartner, S., Beer, J., Wagner, G., Kubik, P., Suter, M., Raisbeck, G. M., & Yiou, F. (1997). 10Be and dust. Nuclear Instruments and Methods in Physics Research Section B: Beam Interactions with Materials and Atoms, 123(1–4), 296–301. https://doi.org/10.1016/s0168-583x(96)00751-3

Finkel, R. C., & Nishiizumi, K. (1997). Beryllium 10 concentrations in the Greenland Ice Sheet Project 2 ice core from 3–40 ka. Journal of Geophysical Research, 102(C12), 26699. https://doi.org/10.1029/97jc01282

Muscheler, R, & Heikkilä, U. (2011). Constraints on long-term changes in solar activity from the range of variability of cosmogenic radionuclide records. Astrophysics and Space Sciences Transactions, 7(3), 355–364. https://doi.org/10.5194/astra-7-355-2011

Muscheler, Raimund, Beer, J., Wagner, G., Laj, C., Kissel, C., Raisbeck, G. M., Yiou, F., & Kubik, P. W. (2004). Changes in the carbon cycle during the last deglaciation as indicated by the comparison of 10Be and 14C records. Earth and Planetary Science Letters, 219(3–4), 325–340. https://doi.org/10.1016/s0012-821x(03)00722-2

Muscheler, Raimund, Joos, F., Beer, J., Müller, S. A., Vonmoos, M., & Snowball, I. (2007). Solar activity during the last 1000yr inferred from radionuclide records. Quaternary Science Reviews, 26(1–2), 82–97.

https://doi.org/10.1016/j.quascirev.2006.07.012

Rasmussen, S. O., Bigler, M., Blockley, S. P., Blunier, T., Buchardt, S. L., Clausen, H. B., Cvijanovic, I., Dahl-Jensen, D., Johnsen, S. J., Fischer, H., Gkinis, V., Guillevic, M., Hoek, W. Z., Lowe, J. J., Pedro, J. B., Popp, T., Seierstad, I. K., Steffensen, J. P., Svensson, A. M., . . . Winstrup, M. (2014). A stratigraphic framework for abrupt climatic changes during the Last Glacial period based on three synchronized Greenland ice-core records: refining and extending the INTIMATE event stratigraphy. Quaternary Science Reviews, 106, 14–28. https://doi.org/http://dx.doi.org/10.1016/j.quascirev.2014.09.007

Steinhilber, F., Abreu, J. A., Beer, J., Brunner, I., Christl, M., Fischer, H., Heikkila, U., Kubik, P. W., Mann, M., McCracken, K. G., Miller, H., Miyahara, H., Oerter, H., & Wilhelms, F. (2012). 9,400 years of cosmic radiation and solar activity from ice cores and tree rings. Proc Natl Acad Sci U S A, 109(16), 5967–5971. https://doi.org/10.1073/pnas.1118965109

Vonmoos, M., Beer, J., & Muscheler, R. (2006). Large variations in Holocene solar activity: Constraints from10Be in the Greenland Ice Core Project ice core. Journal of Geophysical Research, 111(A10), A10105. https://doi.org/10.1029/2005ja011500

Yiou, F., Raisbeck, G. M., Baumgartner, S., Beer, J., Hammer, C., Johnsen, S., Jouzel, J., Kubik, P. W., Lestringuez, J., Stiévenard, M., Suter, M., & Yiou, P. (1997). Beryllium 10 in the Greenland Ice Core Project ice core at Summit, Greenland. Journal of Geophysical Research, 102(C12), 26783. https://doi.org/10.1029/97jc01265

---

## Referee Comment (RC2) · Anonymous Referee #2 · 17 Jul 2020

General comments:

The manuscript by Balco et al. titled 'Atmospherically produced beryllium-10 in annually laminated late-glacial sediments of the North American Varve Chronology' presents an important study of multiple 10Be time-series from pro and paraglacial varved sediments of the NAVC at seasonal to multi-decadal resolution. Major aim is to investigate the mechanisms of 10Be deposition in this, for this purpose, complicated archive and synchronize the new 10Be records to those from Greenland ice cores. The efforts undertaken are methodologically comprehensive and of high interest for the

geochronology community. The authors provide crucial information on the extraction and measurement of 10Be from sediment samples as well as aim to separate fallout (atmospheric) 10Be from inherited (redeposited) 10Be using a linear model. Therefore, I support publication of the paper in Geochronology. However, before publication one major point should be discussed and clarified.

Major point: Environmental influences on 10Be deposition in NAVC sediments:

All calculated bi-annual and decadal 10Be and 9Be flux time-series from the NAVC sediments strongly resemble changes in the mass accumulation rate (MAR) and not the variability in the originally measured 10Be and 9Be time-series. This is a common feature when 10Be flux time-series are calculated from sediment records (e.g. Berggren et al., 2013 J. Paleolim., Czymzik et al., 2015, EPSL). However, comparing the MAR with the measured 10Be and 9Be time-series indicates that major changes in MAR are most often not accompanied by comparable anti-phased changes in 10Be or 9Be (e.g. Figs. 11 and 16), suggesting that influences of changes in MAR on sedimentary 10Be or 9Be contents are assumedly rather small. Based on the above assumptions there might be a substantial environmental signal in the calculated 10Be flux and, consequently, 10Be fallout time-series. Please discuss this subject.

Considering the possible presence of a substantial environmental signal in the calculated 10Be fallout records questions the robustness of the synchronization of NAVC 10Be with Greenland ice core 10Be. Prerequisite for such studies is a reliable production rate signal in all applied 10Be records. This should be considered in the discussion of the synchronization and paleoclimate comparison. In addition, it would be also interesting to see how the originally measured 10Be concentration and 10Be/9Be-ratio time-series compare with Greenland ice core 10Be fluxes, e.g. to evaluate how much common variability is incorporated.

Specific comments:

(1) Calendar year time-scale: The Greenland ice core chronology includes its own

uncertainties (e.g. Muscheler et al., 2014, QSR). Therefore, in my opinion the NAVC is not synchronized to the calendar year time-scale, but to the Greenland ice core GICC05 time-scale.

(2) Page 2/28-31: I addition to the atmospheric fallout, please also mention weather and catchment effects on 10Be deposition.

(3) Pages 22/11 to 23/2: Please shortly discuss these two contradictory results, and their implications for the calculation 10Be fallout variability.

Detailed comments:

Page 1/Line 4: proglacial and 'paraglacial'? Page 3/15-16: Only one 'Ridge, 2012'? Page3/17-24: Please provide references. Page5/1: Please shortly described the composition of the 'glacial sediment'. Page5/3-8: Provide references. Page 8/2: Provide reference. Page 9/31: Provide reference. Page 18/1: Add 'and discussion' after 'results'. Figure 1: Caption: Define the abbreviations in the caption. Add information about the meaning of the long black line. Figure 14: Caption (line 2): Correct 'tallout'.

---

## Author Comment (AC2) · 10 Aug 2020

**Response to review 1 (anonymous)**

First, we thank this reviewer for a close reading and a comprehensive review. This was extremely helpful and we appreciate it.

The main text of this review focuses on the issue of the amplitude of variation in reconstructed Be-10 fallout fluxes. Ideally, if one wants to argue that these really are variations in fallout fluxes, it is not enough to show that the pattern of variability is the same as seen in other fallout records – it is also necessary to show that the magnitude of variability is compatible, or at least not physically impossible. This is important, and in this paper we have not shown that the relative magnitude of variability in our reconstructed fallout fluxes is the same as expected from independent records. This reviewer then goes on to propose that because of this, the correlations between NAVC and other fallout events are speculative.

Basically, we agree with this reasoning, with one important exception. The problem is that the relative amplitude of our reconstructed fallout variability is not reliable because of its dependence on the assumed inherited ratio Rs. Figures 1 and 2 below show this for the Glastonbury (GL) decadal record shown in Figures 16-17 in the paper. In Figure 1, we calculated the fallout flux $Q_{10,a}$ for three different assumed values of Rs, which shows that these different values yield a similar pattern of variability in the reconstructed fallout flux, but different values for the mean fallout flux. Similar absolute variability imposed on different mean values results in different relative magnitudes of variability. Figure 2 shows this effect: higher values of Rs predict higher relative variability in the reconstructed fallout.

[Figure]

Figure 1: Fallout fluxes for Glastonbury (GL) decadal record reconstructed with different values of $R_S$. Of course, the highest value shown is unphysical because it predicts some negative values, but it makes the point clear.

This effect is just a property of the linear model when applied to data sets whose intercept is close to the origin. As can be seen from, e.g., Fig. 13 in the paper, an uncertainty in the intercept (which should give the mean fallout flux under simplifying assumptions as described in the paper) that is small in absolute terms is large in relative terms just because it is close to zero.

[Figure]

Figure 2: Effect of different assumptions for $R_S$ on relative variability of the fallout flux for the Glastonbury (GL) decadal record.

This becomes a serious problem for us because for the decadal records, we really have no way to reliably estimate Rs, as we discussed on p. 30-31 in the paper. If we can't estimate Rs accurately, we cannot accurately estimate the mean fallout flux and therefore also cannot accurately estimate its relative variability. From the perspective of the reviewer's point, therefore, we cannot use the relative magnitude of our reconstructed fallout flux as a point of comparison with independent records of Be-10 fallout. In other words, we cannot disprove the hypothesis that the relative magnitude of variability in our reconstructed fallout records is similar to that in independent records. We also can't disprove the hypothesis that they're different.

So, because of this problem, in discussing potential matches between our records and independent fallout records, we have proceeded from the assumptions that our inability to estimate Rs accurately leads to a situation where we may have correctly reconstructed the pattern of variability, but we cannot assume that we have correctly reconstructed the amplitude of variability. This is why we subsequently rely only on scaled and centered records in the discussion of correlation between records.

To summarize, we agree with the reviewer that convincingly arguing that we have reconstructed fallout variability would require showing that both (i) the pattern and (ii) the amplitude of variability are consistent with independent records. We cannot determine whether the amplitude is consistent or inconsistent, so we have relied only on the pattern of variability. This is, in fact, an inherent weakness of our situation in which most of the Be-10 is inherited and very little is fallout.

To address this in a revised paper, we propose the following:

(1) Add additional discussion in the paper highlighting the difficulty of reconstructing the amplitude of fallout variability given a lack of knowledge of Rs. We did discuss this in section 4.1, and Figures 14 and 15 show examples of reconstructed fallout fluxes with different values of Rs that highlight the difference in amplitude but similarity in pattern between the resulting reconstructions. However, we did not repeat this discussion in section 4.2, where it is of course more directly relevant to the later attempts to correlate the

decadal records. We can add additional discussion of this.

(2) Add additional discussion of why we relied on scaled and centered records for combining the KF and GL decadal records and also in comparing them with ice cores. We agree that the existing paper doesn't give a clear justification for this.

(3) Add additional material to the "weaknesses" section of the conclusions emphasizing that the difficulty in reconstructing the amplitude of variability is a problem for evaluating whether correlations with other records are or are not valid.

It would be possible to add additional figures like Figs. 1 and 2 above in this review, but on the other hand an advantage of the open review system is that this review response remains available to readers. As there are already a lot of figures in the paper, we are inclined not to add more, but we would welcome editorial guidance on this question.

That completes our response to the major issue in this review. We now deal with the minor comments.

*NAVC varve count error: In the entire discussion there is no mention of the varve counting (relative) error of the NAVC. Is this really zero? It should at least be noted in the appropriate sections that differences in the GICC05/NAVC offset may well in part be due to errors in the NAVC, not just GICC05.*

*P6, L1: See earlier: Could you please add a sentence on the counting uncertainty of NAVC? How valid is the approach of using a single value for the offset?*

Basically, the NAVC is believed to have negligible counting uncertainty below approximately NAVC 7200, where nearly all parts of the sequence are replicated in multiple records. The counting uncertainty in stratigraphically higher paraglacial varves that are much thinner and occur only at the Newbury section is significant, and has been discussed in the Ridge (2012) and Ridge and Toll (1999) references. We agree this should have been mentioned in section 1.1 or 1.2, and we will add it. However, note that it's not relevant to the overall conclusions, because we don't consider any Be-10 data above NAVC 7200 for purposes of correlation.

*9Be/10Be extraction: Typically, 9Be and 10Be are measured on the same leaching fraction of sediment. Here, the authors employ different extraction techniques for both isotopes. For this to work, the extraction efficiency of each method must be constant. Is this a problem? Maybe the authors could comment on whether this has the potential to introduce variations in 9/10Be.*

Assuming that using two different methods does not bias the results relies on two assumptions. One, each method is equally efficient on all samples, i.e. that the leaching method for Be-9 extracts all Be-9 that is adsorbed and not mineral-bound, and that the fusion method extracts all Be-10, period. Both of these methods have been tested for complete extraction in a variety of step-leaching experiments for the Be-9 method (described in the Greene reference) and by a variety of tests for the Be-10 fusion method (described in the Stone reference). Thus, there is no indication that the efficiency of these methods is sample-dependent. The second assumption is that the total fusion method does not extract a significant amount of non-adsorbed Be-10 – the aim of the analysis is to determine the amount of adsorbed Be-10 and Be-9, so if we added Be-10 in minerals we might introduce a bias. The only possible source of mineral-bound Be-10 would be in-situ-produced cosmogenic Be-10 present in sediment that had been exposed to the surface cosmic-ray flux prior to glacial erosion and transport. Existing measurements of in-situ-produced Be-10 in subglacial sediment of the Laurentide Ice Sheet indicates in-situ-produced Be-10 concentrations on the order of 10,000 atoms/g,

which is four orders of magnitude smaller than observed total Be-10 concentrations in this study. The highest in-situ-produced Be-10 concentration observed anyhere on the present land surface in New England is slightly less than 1 Matom/g, which is still two orders of magnitude less than total Be-10 observed in this study. Thus, it is not possible for in-situ-produced Be-10 extracted by fusion to cause any significant bias in the present study.

From the perspective of paper revisions, we would prefer not to recapitulate internal reliability tests already described in the Greene and Stone papers. However, we can add a more explicit discussion of the minimal importance of in-situ-produced Be-10 to the methods section....this is already referred to briefly on page 7, but it would probably be good to remind the readers in section 2.5 in the methods as well.

*P 1, L2: replace ?calendar year timescale? with ?ice core timescale?*

Well, really what we are trying to do is synchronize it with both. We'll clarify this.

*P 1, L11-14: long, convoluted sentence. Consider dividing.*

*P 1, L20-24: difficult sentence. Similar but different?*

*P1, L20: ?not consistent with independent evidence? ? the previous sentence states the opposite?*

We'll try to clarify these areas of the abstract.

*P2 L15-16: Please indicate what this uncertainty refers to, i.e., how many sigma.*

As discussed in the Ridge (2012) and Thompson (2017) references, this uncertainty is probably not symmetric or normally distributed, so characterizing it as having a standard error $\sigma$ would be inappropriate. We'll clarify this.

*P2, L28-29: ?generally globally synchronous?. See my comment on an additional sec- tion for the introduction. This is only true under ideal conditions and when averaging sufficiently over meteorological variability. In many settings, meteorological influences will be dominant over the production rate changes.*

The purpose of this section is to give a simplified overview of the concept...that's why we said "generally globally synchronous" instead of "globally synchronous." We will add some clarification here.

*P2, L30: Heikkilae et al. 2013 is not a good reference here, as they do not reconstruct solar variability. Please find more appropriate references such as: (Bard et al., 1997; Raimund Muscheler et al., 2007; Steinhilber et al., 2012).*

Point taken. We will clarify this.

*P2, L32: Again, Heikkilae at al and Adolphi et al. do not contain new measurements. Please give credit to the authors that produced these long records: (Adolphi et al., 2014; Baumgartner et al., 1997; Finkel and Nishiizumi, 1997; Raimund Muscheler et al., 2004; Vonmoos et al., 2006; Yiou et al., 1997)*

Our aim is certainly not to disregard the contributions of the original researchers, but as a matter of scientific citation practice we think it is better to refer to these review papers, which provide extremely thorough and comprehensive reviews of past work, instead of attempting a weaker and less comprehensive review in this

paper. The purpose of the present paper is not to review past work, so we have predominantly cited the review papers.

*P3, L22-24: If there is a transient transition over 100-200 years in the lake system: How can a synchronization of local climate records to Greenland be attempted? The result would depend on which varve record was chosen? Could you please comment? What does this mean for the achievable uncertainty of climate synchronizations?*

Correct: this would indicate that it would be a bad idea to try to correlate a composite record that spans the glacial-interglacial transition at one or more sites. We haven't tried to do this – our attempted correlations involve records that are entirely glacial (KF/GL) or span the transition at a single site (Newbury). At Newbury, of course, we later show that there doesn't seem to be any fallout signal in the paraglacial varves, so data from within and after the transition do not seem to have any value for correlation. Thus, this turns out not to be an issue.

*P6, L16-26: Recently a new radiocarbon calibration was released (IntCal20) which contains significant changes from IntCal13. I understand that it is beyond the scope of this study to redo all the calibrations, but maybe it is worthwhile mentioning that this adds extra uncertainty to the (by looking at the data, in my opinion very optimistic) 200 years.*

Agreed. It is awkward that INTCAL20 came out just as we were working on this paper. As a practical matter, I am not sure we can do anything about this without incurring major delays.

*P6, L29-33: Please do not use the term ?climate events? here, when the event stratigra- phy by the ice core community does not contain any events during this time. Generally, the climate wiggle matching in this section should be cross-checked by not only using 1 ice core, since these minor wiggles in d18O may be noise. They should be verified using GRIP, NGRIP and NEEM d18O.*

Here we were using "climate events" in a general sense to refer to any climate variations recorded in ice cores, not with reference to any specific published "event" stratigraphy. We can easily use "climate changes" or "climate variations" instead.

Another important point, which is also discussed below, is that we are specifically not trying to redo or re-examine the previously proposed varve thickness-oxygen isotope correlations. We are only taking note of previously proposed correlations and comparing them to possible Be-10 fallout matches. This paper is already long enough with just the Be-10 data.

*P9, L15-16: Please add a citation to the figure for the fallout flux.*

Agreed.

*P9, L22: remove the ?-? in the 0.6m.*

Matter for copy-editing.

*P12, L1-3: This is only true if these inaccuracies are systematic. Are they?*

They are certainly systematic in the sense that they can only lead to an underestimate of the density, not an overestimate. However, the reviewer's statement is not strictly accurate. Suppose we measured the density on a number of samples and the results showed a broad distribution that was skewed to the low

side, indicating an underestimate of the density that had both random and systematic contributions. If we then took the average of all the density measurements and applied the average to convert concentrations to fluxes, we would, as is stated in the text, systematically underestimate the total Be flux. In other words, it would not be the distribution of the density measurements themselves, but the use of a simple average to summarize them, that would result in the systematic underestimate of Be fluxes. The point of collecting the data shown in Figure 3 was to to a better job than this.

*P12, L9: Looking at figure 3 the +-0.05g/cm3 seem very optimistic. How would the regression in figure 3 change if the two summer varve measurements were excluded from the analysis?*

The result would be similar, as long as the low-skewed residuals were also excluded. However, the purpose of making the summer and winter measurements was specifically to make this regression more accurate by spanning a wider range of grain sizes, so it is not clear why we would then exclude them.

*P12, L20: replace ?between? with ?around??*

We'll try to clarify this section.

*P14, L9: Is there a +1 missing from the right hand side of the equation? Otherwise, if C/Cs equals 0.9, the right hand side becomes 0?*

Yes, this is an error. Apologies. We will correct it.

*P16, L1-2: Earlier you write that S may be variable? Does this correction come with an uncertainty that is propagated throughout the study?*

We will clarify this. Overall we conclude from the comparison of data between AMS measurement periods that S is most likely constant. However, this is not actually relevant to the uncertainty estimate for the Be-10 concentrations, because we also have an internal criterion for how well the correction procedure performs that is provided by large numbers of replicate measurements. The uncertainty we eventually apply to the Be-10 concentrations is not derived from error propagation from the individual measurements, but instead from analysis of the corrected replicate data. Thus, it includes any uncertainty contributed by variable S.

*P18, Figure 9: Around 6700 NAVC years, there are large changes in MAR and 9Be, but not in 10Be. While it could of course be, that simultaneous and large production rate changes (however, of likely unphysical amplitude, see major comments) ?counteract? this the changes in the delivery of inherited 10Be, this seems unlikely. Rather, this may highlight variable 10/9Be ratios in the inherited Be. Please add a few sentences discussing this feature.*

Figure 9 shows grain size, not MAR. However, this is true and we have discussed it somewhat in section 4.2.2 that deals with the Newbury data that cover that period. This is an interesting point, though, because it is true that a large peak in reconstructed fallout flux around this time, which later becomes important in the correlation section, is associated with large variations in total deposition rate. As the reconstructed fallout peak is only represented by one sample, we agree this looks suspicious. However, this is exactly where we expect a large peak in Be-10 fallout based on ice core records and previous, independent, age calibration of the NAVC. There is no evidence that the Be-9 or Be-10 measurements in this area are inaccurate or anomalous, so we think we have to assume all the data are valid, apply the linear model to reconstruct the fallout rates, and then proceed with the results without second-guessing them. Disregarding a few data, no matter how suspicious they look, without evidence that they are spurious would be bad practice. So I think we are kind of stuck with this. We do highlight at the end of the paper that some of the apparent

correlations are based on very poorly characterized peaks and it would be very helpful to replicate them at higher resolution.

*P22, L8-9: ?constant or normally distributed?: But neither of these assumptions is true. It is obviously not constant (otherwise this study wouldn?t be possible) and it is also not symmetric because i) if solar variability was normally distributed, the non-linear production rate relationship would still cause a skewed distribution of production rates, and ii) the transport and washout of aerosols from the atmosphere causes a logarithmic distribution of aerosol deposition even if the production rates were normally distributed. It is still ok, to use the model as is, but it should be highlighted, that these assumptions are not true, but sufficiently correct to not affect the validity of the results.*

Correct. We can clarify this. Basically, what we are arguing here is that short-term periodic variations should be close enough to symmetric that we can make progress with simple assumptions.

*Figure 13: it would be interesting to see this figure in relative units, i.e., both 9Be and 10Be relative to their respective mean. If the assumption of a constant 10Be/9Be of the inherited Be was true, all data-points should scatter around the 1:1 line?*

Here they are in Fig. 3 below, normalized to standard deviations from the means. However, I think this argument is only strictly valid, though, if the data are really normally distributed? Anyway, they are pretty close.

[Figure]

Figure 3: Relationship of normalized $Q_{10}$ and $Q_9$ for NHV and KF biennial data. The dashed line is 1:1.

*P26, L18: ?factor of 2? it should be noted that also ice cores are affected by transport and deposition effects of 10Be, especially during periods of such variable climate. As mentioned earlier, there are physical constraints of what amplitude of changes we can expect.*

We agree.

*Figure 18: ?95% confidence? How is this determined? Do you take autocorrelation of the records into account?*

This is just a simple comparison against the expected distribution of the correlation coefficient for independent random data. Specifically, we used the MATLAB implementation described here:

https://www.mathworks.com/help/matlab/ref/corrcoef.html

*P31, L3-4: I?d argue that the agreement is not that good? The amplitudes are different, and due to the similarity of both MAR records, the similarities in 10Be fluxes may well be cause by the flux calculation. Is it worthwhile discussing this option?*

We agree that this is not established with extremely high confidence, but the two records have similarities not only in MAR but in the measured 10/9 ratio, which should theoretically be independent of MAR. It's the variations in the ratio that are most important to the flux estimate. If you compare MAR to the flux estimate in Figs. 16-17, it is evident that there are both (i) areas with fairly constant MAR but variable reconstructed flux (NAVC 3600-3800) and also (ii) areas with variable MAR but constant reconstructed flux (4300-4500). Given this, it seems unjustified to argue that only some variations in MAR create spurious flux variations. Having established how to do the flux estimate, we think we have to follow it through and not second-guess particular features later.

*P31, L4-5: The general problem with matching these records is, that one will always find a match due to the periodicity of the signals. Hence, the amplitude discussion is important.*

*Figure 19: The doubling of 10Be/9Be between 6800 and 7200 NAVC cannot be pro- duction (unphysical amplitude). Please discuss.*

*Figure 21: See above*

As discussed above at the beginning of the review, the amplitude of our reconstructed fluxes can't be taken to be reliable. Thus, although the reviewer is correct here, we aren't strictly able to conclude anything from the amplitudes.

*P34, L24-35: I?d argue that the sediment and ice core 10Be records simply don?t look alike. Aligning a single peak can easily lead to erroneous results. Please find a mea- sure for the similarity of the records that can be used to quantify the uncertainty in the match as well.*

Certainly we agree that aligning a single peak can lead to erroneous or at least non-unique results, and we said this in lines 30-35. We also said this in section 5.5 (which actually has the title "Weaknesses."). So we agree completely. Also, we think the correlation metric does show this, specifically in the lowest panel of Fig. 22 that clearly shows identical correlations between the records for different offsets of 20920 and 20675. Obviously, the uncertainty in this situation is not characterized by a continuous uncertainty distribution around some most likely value, but instead by a variety of distinct values that are equally possible, and we think Fig. 22 communicates this.

*Figure 22: Please do not display the data as zscores. A lot of crucial information is lost that way, and it causes deviations between the ice core datasets that are merely due to resolution affecting the standard deviation of each record.*

We think we disagree with this assessment. As we discussed at the beginning of the review, our recon- struction of the amplitude of fallout variability is not reliable and should not be used as a criterion for comparison. Thus, we specifically used a centering and scaling procedure to remove amplitude as an ele- ment in the comparisons. As noted above, we can add additional material clarifying this.

*Figure 24: Please specify on which timescale the GISP2 d18O record is shown. GICC05 I suppose?*

Correct. We can fix this omission.

*Figure 25: Consider plotting NGRIP, GRIP and NEEM d18O as well to get an idea about the robustness of those features.*

In writing this paper we were specifically trying not to break new ground in correlating varve thickness records with ice core climate records, because we are trying to focus on the Be-10 data. The proposed varve thickness-oxygen isotope correlations are reproduced exactly from the Ridge (2012) reference, and we have intentionally not added any new discussion or evaluation of the previously proposed correlations – our aim is just to benchmark the Be-10 data against previously proposed varve-ice core correlations, without reopening the subject of the validity of the previous correlations. We think this is the best approach, because the present paper is already very long, and although we agree that it would be interesting to bring in the GRIP and NGRIP data, we continue to think it is off topic from the perspective of this paper. Certainly we can add some text to clarify that we are not proposing or endorsing the event correlations in this paper, just presenting them as previous work.

*P40, L7: According to the ice core event stratigraphy (Rasmussen et al., 2014) there are no ?events? in the ice cores around this period.*

Again, in several areas of this paper we are using "climate events" in a more general sense to indicate any climate variations recorded in the ice cores. We can clarify this section to make it more clear whether we are talking about generic events or specific named ones.

*Apologies for the lengthy review ? I hope some of these comments are helpful to im- prove the manuscript and the efforts of dating the NAVC.*

Don't apologize. This review is extremely helpful. Frankly, this has been a new area of research for all of us and we are continuing to learn a lot as we put together this paper.

---

## Author Comment (AC3) · 13 Aug 2020

**Response to review 2 (anonymous)**

We thank this reviewer for a careful review. As with the first review, this review was clear and helpful, and we appreciate this reviewer's attention to the paper. Overall, this reviewer was supportive of the paper but had one major concern and a number of minor suggestions.

The major concern has to do with environmental influences on Be-10 deposition in NAVC sediments:

*All calculated bi-annual and decadal 10Be and 9Be flux time-series from the NAVC sediments strongly resemble changes in the mass accumulation rate (MAR) and not the variability in the originally measured 10Be and 9Be time-series. This is a common feature when 10Be flux time-series are calculated from sediment records (e.g. Berggren et al., 2013 J. Paleolim., Czymzik et al., 2015, EPSL). However, comparing the MAR with the measured 10Be and 9Be time-series indicates that major changes in MAR are most often not accompanied by comparable anti-phased changes in 10Be or 9Be (e.g. Figs. 11 and 16), suggesting that influences of changes in MAR on sedimentary 10Be or 9Be contents are assumedly rather small. Based on the above assumptions there might be a substantial environmental signal in the calculated 10Be flux and, consequently, 10Be fallout time-series. Please discuss this subject.*

This comment is quite dense so we will try to clarify it as we understand it before proceeding.

First, because total Be-9 or Be-10 flux (atoms/cm2/yr) is calculated by multiplying measured concentration (atoms/g) and MAR (g/cm2/yr), the only way that flux and MAR could be *uncorrelated* would be if there was a systematic *anticorrelation* between concentration and MAR.

This could be the case in a situation such as an ice core: if one hypothesizes a constant fallout flux of Be-10, variable snow accumulation rates, and no other sources of Be-10 (i.e., no dust input), then one would observe a perfect anticorrelation between MAR (in this case MAR is the snow accumulation rate) and Be-10 concentration, and equivalently zero correlation between MAR and total flux (because total flux is the same as fallout flux and is constant).

In lacustrine sedimentary records, however, as we discuss at length in the paper and the reviewer points out in this comment, this is not possible unless the sediment is delivered to the lake with zero Be-10. In our situation, the sediment is delivered to the lake with a high concentration of Be-10, so the total flux of "inherited" sediment-bound Be-10 is much greater than the Be-10 fallout flux to the lake. We found that the Be concentration in sediments strongly depends on grain size, but if we assume variable MAR and constant grain size, then multiplying an approximately constant Be-10 concentration by variable MAR always results in a total Be-10 flux that is highly correlated with MAR.

Thus, in a sedimentary system where the majority of Be-10 is already adsorbed to sediment when it is delivered to the lake and only a minority is supplied by subsequent fallout, we expect strong correlations between MAR and total Be-10 flux. The purpose of our linear model for unmixing inherited and fallout Be-10 flux is to remove the effect of variations in MAR (and, more importantly, grain size in our case) from the record of total flux, so as to identify the component of the flux that is not affected by environmental factors and therefore may represent fallout.

So, to specifically address the comments in this section of the review:

*All calculated bi-annual and decadal 10Be and 9Be flux time-series from the NAVC sediments strongly resemble changes in the mass accumulation rate (MAR) and not the variability in the originally measured 10Be and 9Be time-series.*

This is correct, and is simply a consequence of the fact that the majority of Be-10 delivered to the lake is adsorbed to sediment and is not from fallout into the lake. Thus, changes in the sedimentation rate are equivalent to changes in the total Be-10 flux.

*However, comparing the MAR with the measured 10Be and 9Be time-series indicates that major changes in MAR are most often not accompanied by comparable anti-phased changes in 10Be or 9Be (e.g. Figs. 11 and 16), suggesting that influences of changes in MAR on sedimentary 10Be or 9Be contents are assumedly rather small.*

Assuming that by "contents" the reviewer means "concentrations" (atoms/g), this is correct. As we note above, observing that MAR and total Be flux are positively, not negatively, correlated indicates that the majority of Be-10 present is inherited and not from fallout. However, a secondary issue is that directly comparing measured concentrations with MAR is oversimplified for our data because, as shown in Fig. 10, Be-10 concentrations in NAVC sediments are strongly dependent on grain size. Thus, the reasoning here would only be strictly valid if MAR was changing but grain size was held constant. Potentially, if MAR and grain size variations were correlated, then changes in MAR would be accompanied by changes in concentration due to the grain size dependence. Leaving aside this complication, however, this statement is essentially saying that because the majority of Be-10 present is inherited, variations in Be-10 fallout have a small effect on measured concentrations. This is true, and we have highlighted this throughout the paper as the major obstacle to reconstructing Be-10 fallout from NAVC sediments.

*Based on the above assumptions there might be a substantial environmental signal in the calculated 10Be flux and, consequently, 10Be fallout time-series. Please discuss this subject.*

Assuming that "calculated 10Be flux" here refers to the reconstructed *fallout* flux, it is certainly true in a general sense that there might be an environmental signal in the fallout reconstruction. However, we disagree somewhat with the reasoning here. The presence of a dominant environmental signal in the *total* Be-10 flux does not by itself imply an environmental signal in the reconstructed *fallout* flux. The purpose of our linear model based on Be-9 concentrations is to identify variations in the total flux that are independent of environmental factors and therefore may represent the fallout flux. Regardless, an environmental signal in the reconstructed fallout flux might arise for two reasons. First, if we incorrectly applied the model (for example, by incorrectly assuming constant $R_S$, or using the wrong value) we might infer spurious variations in the fallout flux that were correlated with environmental parameters. Second, there could be true variations in the flux of fallout Be-10 to the sediment, regardless of whether or not we can reconstruct them correctly, that were caused by environmental effects.

We could test for both of these possibilities by looking for correlations between environmental parameters (e.g., MAR, grain size, or total Be flux, which includes both of these effects) and reconstructed *fallout* fluxes. However, if we observed such correlations, we would not be able to determine whether they were the result of (i) incorrect application of the linear model (in which case the correlation would be an artifact of the error), (ii) environmental effects on the Be-10 fallout flux to the lake that were independent of the fallout rate (for example, if precipitation changes varied the efficiency of delivery of fallout Be-10; this could create apparent variations in fallout that did not reflect true changes in the fallout rate from the atmosphere), or (iii) simultaneous solar forcing of fallout rate and climatic parameters (which could increase or decrease the magnitude of recorded fallout variations, but not create spurious variations). However, because we cannot distinguish between these options without additional data, the presence or absence of a correlation between reconstructed *fallout* flux and some environmental parameter cannot be used to prove that we have or have not correctly identified fallout variations.

Because looking for correlations between reconstructed fallout fluxes and environmental parameters, therefore, has limited usefulness in diagnosing the accuracy of our reconstructed fallout fluxes, we did not focus

on this in the paper. We did, however, do these calculations, and we observed quite variable results that did not lead to any systematic conclusion about the validity of our fallout reconstructions. One interesting example comes from the Newbury biennial record (Figs. 19-21 in the paper). Figure 1 below compares total Be-9 flux to the sediment, which can be considered a measure of all environmental factors affecting Be delivery to the lake, with reconstructed Be-10 fallout flux using our linear model, for the glacial and paraglacial parts of the Newbury sequence.

[Figure]

Figure 1: *Reconstructed Be-10 fallout fluxes for glacial and paraglacial sections of Newbury decadal record (section 4.2.2 in paper) compared to total Be-9 fallout.*

For the glacial part of the section (blue symbols), reconstructed Be-10 fallout flux shows weak and not highly significant correlation with environmental effects as represented by Be-9 flux (the correlation coefficient is 0.27; p = 0.21). This is consistent with the hypothesis that our unmixing model has successfully removed environmental effects from the reconstructed fallout flux (although, as noted above, it does not prove it). Therefore, this analysis does not provide any evidence that the reconstructed fallout flux is unsuitable for correlation with ice core records. This is important in the overall context of the reviewer's comment that environmental influences may obstruct or invalidate correlations with other records, because much of our discussion about correlating NAVC and ice core Be-10 records focuses on the Newbury glacial section, and this analysis shows that it is not possible to prove that environmental influences invalidate the correlation (of course, there are lots of other reasons that the correlation may be invalid that are not addressed by this analysis at all).

On the other hand, reconstructed fallout flux for the paraglacial part of the section (green symbols) is significantly correlated (c = 0.7; p < 0.001) with total Be delivery to the lake, which agrees with our argument on p. 34-36 and elsewhere in the paper that Be-10 delivery to the lake under paraglacial conditions is strongly buffered by watershed processes and is unlikely to record true fallout variations.

To summarize our response to this section of the review, the reviewer is correct that the presence of a strong environmental signal in *total* Be-10 delivery to the lake makes it quite difficult to prove conclusively whether or not our reconstructed *fallout* fluxes are uncontaminated by environmental effects. This is one of the key motivations for our approach of defining a systematic procedure for reconstructing fallout fluxes using a

linear unmixing model, and then, having done so, proceeding to ask whether the fallout reconstruction resulting from this approach is or is not consistent with expectations for ice core records. To address this in a revised paper, we propose the following:

(1) Add some discussion of this issue to the description of our unmixing model in the methods section on pp. 8-9. Although the purpose of the unmixing model is to separate environmental from fallout variability, we did not make that clear in those words. We can make this clearer for readers.

(2) Add explicit discussion of this issue to the conclusions. Although we allude in various places to the issue of environmental vs. fallout variability, the submitted draft did not include an explicit discussion of whether or not the reconstructed fallout estimates are or are not contaminated by environmental factors. We can add this to section 5.

In the original submission we decided not to include figures like Figure 1 above in this review, because, as discussed above, their diagnostic value is somewhat ambiguous, and they do not lead to any clear conclusion. As this paper is already quite long with many figures, we would like to avoid adding additional ones. We note that because of the open review procedure, this discussion will be available to readers, and in addition we have provided all the data plotted in Figs. 11-15, 16-17, and 19-21 as simple supplementary tables, so that readers can easily make similar calculations in a spreadsheet.

*Considering the possible presence of a substantial environmental signal in the calculated 10Be fallout records questions the robustness of the synchronization of NAVC 10Be with Greenland ice core 10Be. Prerequisite for such studies is a reliable production rate signal in all applied 10Be records. This should be considered in the discussion of the synchronization and paleoclimate comparison.*

As discussed above, we agree. However, as also discussed above, this is a particularly difficult aspect of this study because with the data we collected, combined with the inherent imprecision of applying our unmixing model in this situation where Be-10 is dominantly inherited and fallout variations have only a small effect on the observed data, there is no unambiguous way to prove that reconstructed fallout variations are entirely independent of environmental factors. Now that we know more about Be-10 systematics in NAVC sediments, it is possible to think of some ideas for how to do a better job of this, for example by focusing on sections of the NAVC where we expect to observe anomalously large peaks in Be-10 fallout that are represented in the ice core records. We have discussed some of these ideas in the conclusions of the paper.

*In addition, it would be also interesting to see how the originally measured 10Be concentration and 10Be/9Be-ratio time-series compare with Greenland ice core 10Be fluxes, e.g. to evaluate how much common variability is incorporated.*

We agree that this might be interesting, but it is asking a completely different question. Our results clearly show that the total Be-10 (and Be-9) flux to NAVC sediments is mainly controlled by environmental factors, presumably mainly forced by climate. Thus, comparing total Be-10 flux records to Greenland ice core Be-10 fallout records would be equivalent to asking whether solar variability had an effect on local climate. Of course, this question would be better posed by comparing Greenland Be-10 records directly with properties of the NAVC that are more clearly linked to climate, for example varve thickness records, rather than indirectly through measured Be-10 concentrations. With regard to the measured 10/9 ratios, what we are doing in this paper by using a linear model to separate fallout and environmental effects could alternatively be described as simply comparing our measured 10/9 ratios to Greenland ice core Be-10 fluxes – all we have done is apply a linear transformation to the ratio.

That finishes with the main substantive issue in this review. We now address several additional minor substantive comments.

*(1) Calendar year time-scale: The Greenland ice core chronology includes its own uncertainties (e.g. Muscheler et al., 2014, QSR). Therefore, in my opinion the NAVC is not synchronized to the calendar year time-scale, but to the Greenland ice core GICC05 time-scale.*

Reviewer 1 also pointed this out. Of course, the broader objective is to synchronize the NAVC to both the ice core and calendar year timescales. We will clarify this section of the abstract.

*(2) Page 2/28-31: I addition to the atmospheric fallout, please also mention weather and catchment effects on 10Be deposition.*

We will clarify this section.

*(3) Pages 22/11 to 23/2: Please shortly discuss these two contradictory results, and their implications for the calculation 10Be fallout variability.*

A closer look at this section of the text indicates that the submitted version of the paper is misleading in suggesting a conflict here. The purpose of mentioning the Rittenour study was simply to highlight that the only relevant previous work did not show any evidence for the presence of solar-cycle variability in varve thickness variations. Regardless of the Rittenour result, however, the spectral analysis of biennial records in our paper identified spectral power in the 11-year band at 90% confidence only in one out of ten data sets from the NHV and KF sites (5 data sets from each site as shown in Figs. 11 and 12), which is exactly as expected for the null hypothesis in which there is no systematic variability in any of the data sets in this frequency band. Thus, our statement in the original draft that "...spectral analysis of grain size variations...may suggest otherwise" in this section of the text is not, in fact, supported by the analysis. We can clarify this section of the text to make clear that our results are not in conflict with the Rittenour study.

Finally, this reviewer made a number of minor corrections and stylistic suggestions, as follows:

*Page 1/Line 4: proglacial and ?paraglacial??*

*Page 3/15-16: Only one ?Ridge, 2012??*

*Page3/17-24: Please provide references.*

*Page5/1: Please shortly described the com- position of the ?glacial sediment?.*

*Page5/3-8: Provide references.*

*Page 8/2: Provide reference.*

*Page 9/31: Provide reference.*

*Page 18/1: Add ?and discussion? after ?re- sults?.*

*Figure 1: Caption: Define the abbreviations in the caption. Add information about the meaning of the long black line.*

*Figure 14: Caption (line 2): Correct ?tallout?.*

We thank the reviewer for these suggestions and corrections. These areas can be clarified in the revised manuscript.

---

## Author Response (AR1)

Dr. Dietze:

Thanks very much for your attention to this paper and the online discussion. This accompanies a revised manuscript that takes account of the review comments.

In addition to dealing with the specifics of each review, we also changed the general emphasis of several parts of the paper to be more skeptical, in line with the thinking of both reviewers. We think the revised version is fair, informative, and, especially when taken with the open online discussion, contains enough information and discussion for readers to be able to thoroughly evaluate whether or not our conclusions are correct.

Finally, we consolidated citations in several sections of the text to improve clarity and readability. For example, in section 1.1 describing the overall stratigraphy and sedimentology of the NAVC, we cite source papers once at the beginning of the section and indicate that the entire section summarizes material from these sources, instead of citing the same papers after every sentence.

This document indexes the main points made in the reviews to the appropriate sections of the revised text. We identify or repeat each major point of discussion from our posted responses to the two anonymous reviews, and then note the location of changes to the revised text that relate to each point.

Per Copernicus instructions, a latexdiff markup of original and revised text is attached to this document. However, the page and line numbers listed here relate to the revised manuscript, NOT the marked up version.

Note that to keep this document manageable in size, it only discusses the major substantive comments by the reviewers. In addition to these substantive comments, both reviewers corrected a number of errors and omissions and requested minor clarifications in many sections of the text. All of these minor corrections and clarifications that are not specifically indexed here have been corrected in the revised text.

> – Greg Balco (on behalf of all co-authors).
* * *
**Response to review 1 (anonymous)**
* * *
Regarding the main point of this review focusing on the amplitude of variation in reconstructed Be-10 fallout fluxes...

To summarize, we agree with the reviewer that convincingly arguing that we have reconstructed fallout variability would require showing that both (i) the pattern and (ii) the amplitude of variability are consistent with independent records. We cannot determine whether the amplitude is consistent or inconsistent, so we have relied only on the pattern of variability. This is, in fact, an inherent weakness of our situation in which most of the Be-10 is inherited and very little is fallout.

To address this in a revised paper, we propose the following:

(1) Add additional discussion in the paper highlighting the difficulty of reconstructing the amplitude of fallout variability given a lack of knowledge of Rs. We did discuss this in section 4.1, and Figures 14 and 15 show examples of reconstructed fallout fluxes with different values of Rs that highlight the difference in

amplitude but similarity in pattern between the resulting reconstructions. However, we did not repeat this discussion in section 4.2, where it is of course more directly relevant to the later attempts to correlate the decadal records. We can add additional discussion of this.

(2) Add additional discussion of why we relied on scaled and centered records for combining the KF and GL decadal records and also in comparing them with ice cores. We agree that the existing paper doesn't give a clear justification for this.

(3) Add additional material to the "weaknesses" section of the conclusions emphasizing that the difficulty in reconstructing the amplitude of variability is a problem for evaluating whether correlations with other records are or are not valid.

It would be possible to add additional figures like Figs. 1 and 2 above in this review, but on the other hand an advantage of the open review system is that this review response remains available to readers. As there are already a lot of figures in the paper, we are inclined not to add more, but we would welcome editorial guidance on this question.

We added or expanded discussion in section 4.1, p. 25, lines 5-10 through p. 26 lines 1-7; section 4.2.1, page 30, lines 20-30; section 4.2.2, p. 34, lines 10-15; section 5.5, page 46, lines 20-25.
* * *
*NAVC varve count error: In the entire discussion there is no mention of the varve counting (relative) error of the NAVC. Is this really zero? It should at least be noted in the appropriate sections that differences in the GICC05/NAVC offset may well in part be due to errors in the NAVC, not just GICC05.*

*P6, L1: See earlier: Could you please add a sentence on the counting uncertainty of NAVC? How valid is the approach of using a single value for the offset?*

Basically, the NAVC is believed to have negligible counting uncertainty below approximately NAVC 7200, where nearly all parts of the sequence are replicated in multiple records. The counting uncertainty in stratigraphically higher paraglacial varves that are much thinner and occur only at the Newbury section is significant, and has been discussed in the Ridge (2012) and Ridge and Toll (1999) references. We agree this should have been mentioned in section 1.1 or 1.2, and we will add it. However, note that it's not relevant to the overall conclusions, because we don't consider any Be-10 data above NAVC 7200 for purposes of correlation.

We added this to section 1.2, page 6 near line 5.
* * *
*9Be/10Be extraction: Typically, 9Be and 10Be are measured on the same leaching fraction of sediment. Here, the authors employ different extraction techniques for both isotopes. For this to work, the extraction efficiency of each method must be constant. Is this a problem? Maybe the authors could comment on whether this has the potential to introduce variations in 9/10Be.*

Assuming that using two different methods does not bias the results relies on two assumptions. One, each method is equally efficient on all samples, i.e. that the leaching method for Be-9 extracts all Be-9 that is adsorbed and not mineral-bound, and that the fusion method extracts all Be-10, period. Both of these methods have been tested for complete extraction in a variety of step-leaching experiments for the Be-9 method (described in the Greene reference) and by a variety of tests for the Be-10 fusion method (described in the Stone reference). Thus, there is no indication that the efficiency of these methods is sample-dependent. The

second assumption is that the total fusion method does not extract a significant amount of non-adsorbed Be-10 – the aim of the analysis is to determine the amount of adsorbed Be-10 and Be-9, so if we added Be-10 in minerals we might introduce a bias. The only possible source of mineral-bound Be-10 would be in-situ-produced cosmogenic Be-10 present in sediment that had been exposed to the surface cosmic-ray flux prior to glacial erosion and transport. Existing measurements of in-situ-produced Be-10 in subglacial sediment of the Laurentide Ice Sheet indicates in-situ-produced Be-10 concentrations on the order of 10,000 atoms/g, which is four orders of magnitude smaller than observed total Be-10 concentrations in this study. The highest in-situ-produced Be-10 concentration observed anyhere on the present land surface in New England is slightly less than 1 Matom/g, which is still two orders of magnitude less than total Be-10 observed in this study. Thus, it is not possible for in-situ-produced Be-10 extracted by fusion to cause any significant bias in the present study.

From the perspective of paper revisions, we would prefer not to recapitulate internal reliability tests already described in the Greene and Stone papers. However, we can add a more explicit discussion of the minimal importance of in-situ-produced Be-10 to the methods section....this is already referred to briefly on page 7, but it would probably be good to remind the readers in section 2.5 in the methods as well.

We revised the methods section to be more clear about the differences between the two types of measurement. Section 2.4, page 1, lines 15-20, and section 2.5, p. 13, lines 10-20.
* * *
*P2 L15-16: Please indicate what this uncertainty refers to, i.e., how many sigma.*

As discussed in the Ridge (2012) and Thompson (2017) references, this uncertainty is probably not symmetric or normally distributed, so characterizing it as having a standard error $\sigma$ would be inappropriate. We'll clarify this.

We clarified this on page 6 around line 25.
* * *
*P2, L32: Again, Heikkilae at al and Adolphi et al. do not contain new measurements. Please give credit to the authors that produced these long records: (Adolphi et al., 2014; Baumgartner et al., 1997; Finkel and Nishiizumi, 1997; Raimund Muscheler et al., 2004; Vonmoos et al., 2006; Yiou et al., 1997)*

Our aim is certainly not to disregard the contributions of the original researchers, but as a matter of scientific citation practice we think it is better to refer to these review papers, which provide extremely thorough and comprehensive reviews of past work, instead of attempting a weaker and less comprehensive review in this paper. The purpose of the present paper is not to review past work, so we have predominantly cited the review papers.

As noted in the review response, we cited review papers rather than attempting to cite all primary sources for [10]Be measurements in ice cores.
* * *
*P3, L22-24: If there is a transient transition over 100-200 years in the lake system: How can a synchronization of local climate records to Greenland be attempted? The result would depend on which varve record was chosen? Could you please comment? What does this mean for the achievable uncertainty of climate synchronizations?*

Correct: this would indicate that it would be a bad idea to try to correlate a composite record that spans the glacial-interglacial transition at one or more sites. We haven't tried to do this – our attempted correlations

involve records that are entirely glacial (KF/GL) or span the transition at a single site (Newbury). At Newbury, of course, we later show that there doesn't seem to be any fallout signal in the paraglacial varves, so data from within and after the transition do not seem to have any value for correlation. Thus, this turns out not to be an issue.

This issue did not appear to require any changes to the text.
* * *
*P6, L16-26: Recently a new radiocarbon calibration was released (IntCal20) which contains significant changes from IntCal13. I understand that it is beyond the scope of this study to redo all the calibrations, but maybe it is worthwhile mentioning that this adds extra uncertainty to the (by looking at the data, in my opinion very optimistic) 200 years.*

Agreed. It is awkward that INTCAL20 came out just as we were working on this paper. As a practical matter, I am not sure we can do anything about this without incurring major delays.

As noted, we did not attempt to update all radiocarbon calibrations in this paper to INTCAL20.
* * *
*P12, L1-3: This is only true if these inaccuracies are systematic. Are they?*

They are certainly systematic in the sense that they can only lead to an underestimate of the density, not an overestimate. However, the reviewer's statement is not strictly accurate. Suppose we measured the density on a number of samples and the results showed a broad distribution that was skewed to the low side, indicating an underestimate of the density that had both random and systematic contributions. If we then took the average of all the density measurements and applied the average to convert concentrations to fluxes, we would, as is stated in the text, systematically underestimate the total Be flux. In other words, it would not be the distribution of the density measurements themselves, but the use of a simple average to summarize them, that would result in the systematic underestimate of Be fluxes. The point of collecting the data shown in Figure 3 was to to a better job than this.

*P12, L9: Looking at figure 3 the +-0.05g/cm3 seem very optimistic. How would the regression in figure 3 change if the two summer varve measurements were excluded from the analysis?*

The result would be similar, as long as the low-skewed residuals were also excluded. However, the purpose of making the summer and winter measurements was specifically to make this regression more accurate by spanning a wider range of grain sizes, so it is not clear why we would then exclude them.

These issues did not appear to require additional text.
* * *
*P16, L1-2: Earlier you write that S may be variable? Does this correction come with an uncertainty that is propagated throughout the study?*

We will clarify this. Overall we conclude from the comparison of data between AMS measurement periods that S is most likely constant. However, this is not actually relevant to the uncertainty estimate for the Be-10 concentrations, because we also have an internal criterion for how well the correction procedure performs that is provided by large numbers of replicate measurements. The uncertainty we eventually apply to the Be-10 concentrations is not derived from error propagation from the individual measurements, but instead from analysis of the corrected replicate data. Thus, it includes any uncertainty contributed by variable S.

We clarified this issue on p. 17, near line 5 and also in the next paragraph near line 10.
* * *
*P18, Figure 9: Around 6700 NAVC years, there are large changes in MAR and 9Be, but not in 10Be. While it could of course be, that simultaneous and large production rate changes (however, of likely unphysical amplitude, see major comments) ?counteract? this the changes in the delivery of inherited 10Be, this seems unlikely. Rather, this may highlight variable 10/9Be ratios in the inherited Be. Please add a few sentences discussing this feature.*

Figure 9 shows grain size, not MAR. However, this is true and we have discussed it somewhat in section 4.2.2 that deals with the Newbury data that cover that period. This is an interesting point, though, because it is true that a large peak in reconstructed fallout flux around this time, which later becomes important in the correlation section, is associated with large variations in total deposition rate. As the reconstructed fallout peak is only represented by one sample, we agree this looks suspicious. However, this is exactly where we expect a large peak in Be-10 fallout based on ice core records and previous, independent, age calibration of the NAVC. There is no evidence that the Be-9 or Be-10 measurements in this area are inaccurate or anomalous, so we think we have to assume all the data are valid, apply the linear model to reconstruct the fallout rates, and then proceed with the results without second-guessing them. Disregarding a few data, no matter how suspicious they look, without evidence that they are spurious would be bad practice. So I think we are kind of stuck with this. We do highlight at the end of the paper that some of the apparent correlations are based on very poorly characterized peaks and it would be very helpful to replicate them at higher resolution.

As this issue was already discussed in the original version, it did not appear to require new text.
* * *
*P22, L8-9: ?constant or normally distributed?: But neither of these assumptions is true. It is obviously not constant (otherwise this study wouldn?t be possible) and it is also not symmetric because i) if solar variability was normally distributed, the non-linear production rate relationship would still cause a skewed distribution of production rates, and ii) the transport and washout of aerosols from the atmosphere causes a logarithmic distribution of aerosol deposition even if the production rates were normally distributed. It is still ok, to use the model as is, but it should be highlighted, that these assumptions are not true, but sufficiently correct to not affect the validity of the results.*

Correct. We can clarify this. Basically, what we are arguing here is that short-term periodic variations should be close enough to symmetric that we can make progress with simple assumptions.

Clarified on page 23, lines 1-10.
* * *
*P26, L18: ?factor of 2? it should be noted that also ice cores are affected by transport and deposition effects of 10Be, especially during periods of such variable climate. As mentioned earlier, there are physical constraints of what amplitude of changes we can expect.*

We agree.

We added a brief reference to this issue on p. 26, near line 5.
* * *
*Figure 18: ?95% confidence? How is this determined? Do you take autocorrelation of the records into account?*

This is just a simple comparison against the expected distribution of the correlation coefficient for independent random data. Specifically, we used the MATLAB implementation described here:

https://www.mathworks.com/help/matlab/ref/corrcoef.html

This did not appear to require any changes to the text, although we made sure to use standard statistics terminology throughout the paper to avoid confusion, and we added a more precise description of the confidence estimate to the caption for Fig. 18.
* * *
*P31, L3-4: I?d argue that the agreement is not that good? The amplitudes are different, and due to the similarity of both MAR records, the similarities in 10Be fluxes may well be cause by the flux calculation. Is it worthwhile discussing this option?*

We agree that this is not established with extremely high confidence, but the two records have similarities not only in MAR but in the measured 10/9 ratio, which should theoretically be independent of MAR. It's the variations in the ratio that are most important to the flux estimate. If you compare MAR to the flux estimate in Figs. 16-17, it is evident that there are both (i) areas with fairly constant MAR but variable reconstructed flux (NAVC 3600-3800) and also (ii) areas with variable MAR but constant reconstructed flux (4300-4500). Given this, it seems unjustified to argue that only some variations in MAR create spurious flux variations. Having established how to do the flux estimate, we think we have to follow it through and not second-guess particular features later.

This did not appear to require any specific changes. As described below in reference to the second reviewer's major concern, we did add some discussion of environmental vs. fallout variation in several places.
* * *
*P31, L4-5: The general problem with matching these records is, that one will always find a match due to the periodicity of the signals. Hence, the amplitude discussion is important.*

*Figure 19: The doubling of 10Be/9Be between 6800 and 7200 NAVC cannot be pro- duction (unphysical amplitude). Please discuss.*

*Figure 21: See above*

As discussed above at the beginning of the review, the amplitude of our reconstructed fluxes can't be taken to be reliable. Thus, although the reviewer is correct here, we aren't strictly able to conclude anything from the amplitudes.

As noted above in our response to the first major comment, we added discussion of this issue in several places.
* * *
*P34, L24-35: I?d argue that the sediment and ice core 10Be records simply don?t look alike. Aligning a single peak can easily lead to erroneous results. Please find a mea- sure for the similarity of the records that can be used to quantify the uncertainty in the match as well.*

Certainly we agree that aligning a single peak can lead to erroneous or at least non-unique results, and we said this in lines 30-35. We also said this in section 5.5 (which actually has the title "Weaknesses."). So we

agree completely. Also, we think the correlation metric does show this, specifically in the lowest panel of Fig. 22 that clearly shows identical correlations between the records for different offsets of 20920 and 20675. Obviously, the uncertainty in this situation is not characterized by a continuous uncertainty distribution around some most likely value, but instead by a variety of distinct values that are equally possible, and we think Fig. 22 communicates this.

This did not seem to require any changes to the text.
* * *
*Figure 22: Please do not display the data as zscores. A lot of crucial information is lost that way, and it causes deviations between the ice core datasets that are merely due to resolution affecting the standard deviation of each record.*

We think we disagree with this assessment. As we discussed at the beginning of the review, our reconstruction of the amplitude of fallout variability is not reliable and should not be used as a criterion for comparison. Thus, we specifically used a centering and scaling procedure to remove amplitude as an element in the comparisons. As noted above, we can add additional material clarifying this.

We added specific discussion of this in section 4.2.1 on page 30, lines 30-35.
* * *
*Figure 25: Consider plotting NGRIP, GRIP and NEEM d18O as well to get an idea about the robustness of those features.*

In writing this paper we were specifically trying not to break new ground in correlating varve thickness records with ice core climate records, because we are trying to focus on the Be-10 data. The proposed varve thickness-oxygen isotope correlations are reproduced exactly from the Ridge (2012) reference, and we have intentionally not added any new discussion or evaluation of the previously proposed correlations – our aim is just to benchmark the Be-10 data against previously proposed varve-ice core correlations, without reopening the subject of the validity of the previous correlations. We think this is the best approach, because the present paper is already very long, and although we agree that it would be interesting to bring in the GRIP and NGRIP data, we continue to think it is off topic from the perspective of this paper. Certainly we can add some text to clarify that we are not proposing or endorsing the event correlations in this paper, just presenting them as previous work.

We did not add new material evaluating previously proposed correlations with GISP2 also against GRIP and NGRIP. Honestly, we agree this would be interesting but it's just beyond the scope of the present paper.
* * *
**Response to review 2 (anonymous)**
* * *
Regarding the main point of this review focusing on environmental variability in reconstructed Be-10 fallout fluxes...

To summarize our response to this section of the review, the reviewer is correct that the presence of a strong environmental signal in *total* Be-10 delivery to the lake makes it quite difficult to prove conclusively whether or not our reconstructed *fallout* fluxes are uncontaminated by environmental effects. This is one of the key motivations for our approach of defining a systematic procedure for reconstructing fallout fluxes using a linear unmixing model, and then, having done so, proceeding to ask whether the fallout reconstruction

resulting from this approach is or is not consistent with expectations for ice core records. To address this in a revised paper, we propose the following:

(1) Add some discussion of this issue to the description of our unmixing model in the methods section on pp. 8-9. Although the purpose of the unmixing model is to separate environmental from fallout variability, we did not make that clear in those words. We can make this clearer for readers.

(2) Add explicit discussion of this issue to the conclusions. Although we allude in various places to the issue of environmental vs. fallout variability, the submitted draft did not include an explicit discussion of whether or not the reconstructed fallout estimates are or are not contaminated by environmental factors. We can add this to section 5.

In the original submission we decided not to include figures like Figure 1 above in this review, because, as discussed above, their diagnostic value is somewhat ambiguous, and they do not lead to any clear conclusion. As this paper is already quite long with many figures, we would like to avoid adding additional ones. We note that because of the open review procedure, this discussion will be available to readers, and in addition we have provided all the data plotted in Figs. 11-15, 16-17, and 19-21 as simple supplementary tables, so that readers can easily make similar calculations in a spreadsheet.

Added discussion in section 1.3, p. 9, lines 5-10; section 4.2, p. 29, lines 5-10 through p. 30 line 15; page 35 lines 10-15; page 44, lines 5-10.
* * *
*Considering the possible presence of a substantial environmental signal in the calculated 10Be fallout records questions the robustness of the synchronization of NAVC 10Be with Greenland ice core 10Be. Prerequisite for such studies is a reliable production rate signal in all applied 10Be records. This should be considered in the discussion of the synchronization and paleoclimate comparison.*

As discussed above, we agree. However, as also discussed above, this is a particularly difficult aspect of this study because with the data we collected, combined with the inherent imprecision of applying our unmixing model in this situation where Be-10 is dominantly inherited and fallout variations have only a small effect on the observed data, there is no unambiguous way to prove that reconstructed fallout variations are entirely independent of environmental factors. Now that we know more about Be-10 systematics in NAVC sediments, it is possible to think of some ideas for how to do a better job of this, for example by focusing on sections of the NAVC where we expect to observe anomalously large peaks in Be-10 fallout that are represented in the ice core records. We have discussed some of these ideas in the conclusions of the paper.

The added discussion noted above pertains to this comment as well.
* * *
*In addition, it would be also interesting to see how the originally measured 10Be concentration and 10Be/9Be-ratio time-series compare with Greenland ice core 10Be fluxes, e.g. to evaluate how much common variability is incorporated.*

We agree that this might be interesting, but it is asking a completely different question. Our results clearly show that the total Be-10 (and Be-9) flux to NAVC sediments is mainly controlled by environmental factors, presumably mainly forced by climate. Thus, comparing total Be-10 flux records to Greenland ice core Be-10 fallout records would be equivalent to asking whether solar variability had an effect on local climate. Of course, this question would be better posed by comparing Greenland Be-10 records directly with properties of the NAVC that are more clearly linked to climate, for example varve thickness records, rather than indirectly through measured Be-10 concentrations. With regard to the measured 10/9 ratios, what we are doing in this paper by using a linear model to separate fallout and environmental effects could alternatively be described as simply comparing our measured 10/9 ratios to Greenland ice core Be-10 fluxes – all we have done is apply a linear transformation to the ratio.

This did not appear to require any additional text.
* * *
*(3) Pages 22/11 to 23/2: Please shortly discuss these two contradictory results, and their implications for the calculation 10Be fallout variability.*

A closer look at this section of the text indicates that the submitted version of the paper is misleading in suggesting a conflict here. The purpose of mentioning the Rittenour study was simply to highlight that the only relevant previous work did not show any evidence for the presence of solar-cycle variability in varve thickness variations. Regardless of the Rittenour result, however, the spectral analysis of biennial records in our paper identified spectral power in the 11-year band at 90% confidence only in one out of ten data sets from the NHV and KF sites (5 data sets from each site as shown in Figs. 11 and 12), which is exactly as expected for the null hypothesis in which there is no systematic variability in any of the data sets in this frequency band. Thus, our statement in the original draft that "...spectral analysis of grain size variations...may suggest otherwise" in this section of the text is not, in fact, supported by the analysis. We can clarify this section of the text to make clear that our results are not in conflict with the Rittenour study.

We removed the incorrect statement. Page 23, near line 10.
* * *

[revised manuscript text omitted]

---

## Author Response (AR2)

Nov. 12, 2020

Dr. Dietze:

Thanks very much for your attention to this paper and the online discussion. This accompanies a second revised manuscript that includes minor changes in response to the second round of review comments.

As noted in your revision instructions, the second reviews only included one important comment, which is a request to amend Figures 18 and 22 such that readers can discern the relative amplitude of variations in both ice core [10]Be fallout records and our reconstructed [10]Be fallout fluxes from NAVC sediments. In the initial and revised version, these figures only showed centered records that were scaled by their respective standard deviations, or in other words 'z-scores,' thus obscuring information about the magnitude of variation relative to the mean of each record.

We take the reviewer's point and agree with this suggestion. As discussed in our email correspondence, we think the easiest way to correct the problem is to add additional y-axes to the relevant plots in Figs. 18 and 22, such that each y-axis is labeled both with 'z-score' units of standard deviations and also with units of variation relative to the mean. Although this adds to the information density in these already dense figures, it gives readers the ability to clearly appreciate the reviewer's concern that the relative amplitude of our reconstructed fallout flux variations is not the same as the relative amplitude of [10]Be flux variations in ice cores.

Thus, this revision of the paper includes only additions to Figs. 18 and 22 and their captions. Because there are no other changes, we have not included a separate latexdiff markup of the second and third versions of the paper. We hope these changes are sufficient for acceptance and publication.

Regards,

– Greg Balco (on behalf of all co-authors).

---

## Author Response (AR3)

Production files include the following changes from the accepted version ('gchron-2020-16-manuscript-version3.pdf'):

- Corrected several typographical errors and misspellings

- Changed aspect ratios and font sizes in several figures

- Updated author information

These changes do not affect the scientific content of the paper.

Regards,

    – Greg Balco (on behalf of all co-authors).